# Kinase KEY1 controls pyrenoid condensate size throughout the cell cycle by disrupting phase separation interactions

Shan He [1,2,8,11], Linnea M. Lemma[2,3,11], Alejandro Martinez-Calvo[4,5,6], Guanhua He[1], Jessica H. Hennacy[1], Lianyong Wang[1,9], Sabrina L. Ergun[1,2], Ashwani K. Rai[1], Colton Wang[1], Luke Bunday[1], Angelo Kayser-Browne[1], Quan Wang[7,10], Clifford P. Brangwynne [2,3,6], Ned S. Wingreen [1,4,7] ✉ & Martin C. Jonikas [1,2,3] ✉

Biomolecular condensates spatially organize cellular functions, but the regulation of their size, number, dissolution and re-condensation is poorly understood. The pyrenoid, an algal biomolecular condensate that mediates one-third of global $CO_2$ fixation, typically exists as one large condensate per chloroplast, but during cell division it transiently dissolves and reconfigures into multiple smaller condensates. Here, we identify a kinase, KEY1, in the model alga *Chlamydomonas reinhardtii* that regulates pyrenoid condensate size and number dynamics throughout the cell cycle and is necessary for normal pyrenoid function and growth. Unlike the wild type, *key1* mutant cells have multiple smaller condensates throughout the cell cycle that fail to dissolve during cell division. We show that KEY1 localizes to the condensates and promotes their dissolution by disrupting interactions between their core constituents, the $CO_2$-fixing enzyme Rubisco and its linker protein EPYC1, through EPYC1 phosphorylation. We develop a biophysical model that recapitulates KEY1-mediated condensate size and number regulation and suggests a mechanism for controlling condensate position. These data provide a foundation for the mechanistic understanding of the regulation of size, number, position and dissolution in pyrenoids and other biomolecular condensates.

Biomolecular condensates, such as the nucleolus[1], P granules[2] and purinosomes[3], enable the colocalization of biomolecules[2,4–6] to facilitate diverse functions[7–20] from ribosome biogenesis[21] to metabolic regulation[22,23]. Condensates form through liquid–liquid phase separation, a process by which proteins and/or nucleic acids assemble into droplets[4,14,24–27].

Condensate size impacts biological function, particularly evident in the context of metabolic compartmentalization, where enzyme-containing condensates show maximal reaction efficiency within a specific size range[28,29]. Aberrant condensate sizes are associated with disease, such as enlarged nucleoli in pancreatic cancer[30,31].

There has been substantial interest in understanding how condensate size is determined and regulated. The size distributions of some condensates seem to be passively governed by scaling laws[1,32], surface tension[33,34] or diffusion-limited dynamics[35]. Experimental studies show that the sizes of some condensates are regulated by nucleation[36] or building-block availability[37] and synthesis[38]. Furthermore, the one-time dissolution of multiple condensates is well established[39,40].

Theoretical mechanisms have been proposed for how cells could actively control condensate size by regulating the relative rates of dissolution and condensation[41–50]. While such active control of condensate size has been demonstrated in a synthetic system[51], it has not been established in living cells.

The pyrenoid is a singular biomolecular condensate[52] found in the chloroplast of eukaryotic algae[53,54], where it mediates approximately one-third of global $CO_2$ assimilation[55]. The pyrenoid condensate clusters the $CO_2$-fixing enzyme Rubisco[56] around a localized $CO_2$ source, enhancing Rubisco's activity[57]. In the model alga *Chlamydomonas reinhardtii* (*Chlamydomonas*), the pyrenoid condensate forms through multivalent interactions between Rubisco and the intrinsically disordered linker protein EPYC1 (Fig. 1a)[55,58,59]. Mutants that fail to form a singular pyrenoid condensate have growth defects under conditions that require efficient $CO_2$ delivery to Rubisco, including ambient air[55,59–62], making the pyrenoid one of the few condensates with a known functional significance of condensate formation. During cell division, the size, number, dissolution and re-condensation of pyrenoid condensates are highly dynamic and seem tightly regulated: the mother cell's single condensate rapidly dissolves and multiple smaller condensates appear before coarsening into one per descendant cell[52]. However, the mechanisms regulating these behaviours remain unknown.

Here, we use genetics, cell biology, in vitro reconstitution and mathematical modelling to elucidate mechanisms regulating pyrenoid condensate phase behaviours, including size and dissolution. We discover that a protein kinase, KEY1, regulates pyrenoid condensate size throughout the cell cycle and mediates condensate dissolution during cell division. KEY1 functions in the condensate and is targeted to it via a common Rubisco-binding motif. KEY1 phosphorylates EPYC1 on its Rubisco-binding sites, disrupting the Rubisco–EPYC1 interactions that drive phase separation. We show that the observed changes in pyrenoid size, number and dissolution can be recapitulated with a minimal mathematical model of condensate regulation by kinase-driven protein fluxes. Our findings identify a central molecular regulator of pyrenoid phase behaviours and elucidate a mechanism for regulating condensate size, number and dissolution.

## Results

### The candidate protein kinase KEY1 physically interacts with EPYC1

During *Chlamydomonas* cell division, pyrenoid condensate dissolution and condensation each occur within 5–10 min[52], a timescale commonly associated with post-translational regulation[63,64]. Moreover, several phosphopeptides of the Rubisco linker protein EPYC1 were previously identified through proteomics[65–68]. We therefore hypothesized that pyrenoid condensate phase behaviours are regulated through EPYC1 phosphorylation.

To test this hypothesis, we sought to identify the kinase that phosphorylates EPYC1. Our top candidate was Cre01.g008550 (Extended Data Fig. 1a), encoding a predicted dual-specificity protein kinase[69] that physically interacted with EPYC1 in a previous large-scale protein-protein interaction study[70]. Based on the results presented

below, we name this kinase 'kinase of EPYC1' (KEY1). We replicated the immunoprecipitation of KEY1 by EPYC1 (Fig. 1b and Source data) and also observed that EPYC1 co-precipitated with KEY1 when the latter was used as a bait (Fig. 1c and Source data), validating the physical interaction between KEY1 and EPYC1.

### KEY1 is necessary for normal pyrenoid size, number and function

To investigate whether KEY1 regulates phase behaviours of the pyrenoid condensate, we characterized two *key1* intron insertion mutant alleles, *key1-1* and *key1-2*, from the CLiP mutant library[71] (*Chlamydomonas* Resource Center IDs LMJ.RY0402.107748 and LMJ.RY0402.168949; Extended Data Fig. 1b–h, Supplementary Table 1 and Methods). *key1-1* had lower *KEY1* messenger RNA transcript abundance (~3% of wild type; Extended Data Fig. 1i) than *key1-2* (~70% of wild type). Using transmission electron microscopy, we observed more than one pyrenoid condensate in *key1-1* cells, whereas wild-type cells typically possessed a single condensate (Fig. 1d,e).

As transmission electron microscopy examines only a thin cell slice, we used confocal microscopy of Venus-tagged Rubisco (RBCS1–Venus) to quantify pyrenoid condensate size and number in wild-type and *key1-1* cells[55,72]. We observed that *key1-1* had multiple smaller pyrenoid condensates in contrast to the singular large pyrenoid condensate in wild-type cells (Fig. 1f,g and Supplementary Video 1). Reintroducing the *KEY1* gene under its endogenous promoter into *key1-1* restored the singular large pyrenoid (Fig. 1h and Supplementary Video 1), establishing that the multiple-small-condensate phenotype was due to a disruption of *KEY1*.

On average, *key1-1* had more condensates per cell than wild-type and rescued strains (Fig. 1i; $P < 10^{-6}$, *t*-test). The largest condensate was smaller on average in *key1-1* mutant cells than in wild-type or *key1-1;KEY1-SNAP*[73,74] rescued cells (Fig. 1j; $P < 0.002$, *t*-test), and the distribution of the size of the largest condensate was broader, suggesting a defect in size regulation. These results establish that KEY1 is essential for normal pyrenoid condensate size and number.

To determine whether KEY1 activity affected pyrenoid function in the algal $CO_2$-concentrating mechanism, we performed a spot test growth assay. *key1-1* and *key1-2* exhibited growth defects in Tris-phosphate medium under low $CO_2$ (0.04%, air level) and very low $CO_2$ (0.004%) conditions, where cells require a functional pyrenoid to grow, but not under high $CO_2$ or on Tris-acetate-phosphate medium in the dark, where a pyrenoid is not required for growth[55,75,76] (Fig. 1k). These growth defects were partially rescued by introducing Venus–3×Flag- or SNAP–3×Flag-tagged *KEY1* under its endogenous promoter. The incomplete rescue at very low $CO_2$ could be due to the presence of the tags or due to suboptimal regulation as the constructs insert at random sites in the genome. Together, these results indicate that KEY1 is necessary for normal pyrenoid size, number and function.

### Pyrenoid dissolution and re-condensation often occur via a multiple-small-condensate intermediate

To understand how KEY1 controls pyrenoid condensate size and number, we first investigated pyrenoid dynamics during cell division, when

**Fig. 1 | The candidate kinase KEY1 is necessary for normal pyrenoid size, number and function. a**, The pyrenoid matrix in *Chlamydomonas* is a biomolecular condensate that forms through phase separation of the $CO_2$-fixing enzyme Rubisco and the intrinsically disordered linker protein EPYC1. **b,c**, Spectral counts of proteins identified by mass spectrometry after immunoprecipitation of EPYC1–Venus–3×Flag (**b**) or KEY1–Venus–3×Flag (**c**), plotted against the spectral counts of the same proteins after immunoprecipitation of Venus–3×Flag. See also Source data. **d,e**, Transmission electron micrographs of a wild-type (WT) cell (**d**) and a *key1-1* mutant cell (**e**). P, pyrenoid. See also Extended Data Fig. 1. The experiment was performed twice independently with similar results. **f–h**, Representative confocal fluorescence images of WT (**f**), *key1-1* mutant (**g**) and the rescued strain *key1-1;KEY1-SNAP* (**h**), each expressing RBCS1–Venus to label Rubisco. Magenta shows the mid-plane of

chlorophyll autofluorescence. Green shows the maximum intensity projection of RBCS1–Venus. See also Supplementary Video 1. The experiment was performed three times independently with $n > 100$ cells for each strain with similar results. **i,j**, Violin plots of the number of condensates per cell (**i**) and the per cent of cell area that the largest condensate occupies (**j**) for Rubisco labelled by RBCS1–Venus in WT ($n = 191$ cells), *key1-1* mutant ($n = 208$ cells) and *key1-1;KEY1-SNAP* rescue ($n = 122$ cells). $P$ values calculated by a two-sided *t*-test; $P_1 = 0.27$, $P_2 = 10^{-67}$, $P_3 = 10^{-46}$, $P_4 = 10^{-11}$, $P_5 = 0.0012$ and $P_6 = 10^{-14}$. White circles indicate median of the distributions. Grey bars show first (thick) and third (thin) quartiles of the distributions. **k**, Spot growth assays of WT, two *key1* mutant alleles and two rescued strains of *key1-1* on Tris-phosphate (TP) medium in low $CO_2$ (air, 0.04%), very low $CO_2$ (0.004%), high $CO_2$ (3%) in light at 200 µmol photons $m^{-2}$ $s^{-1}$, and on Tris-acetate-phosphate (TAP) medium in the dark in air.

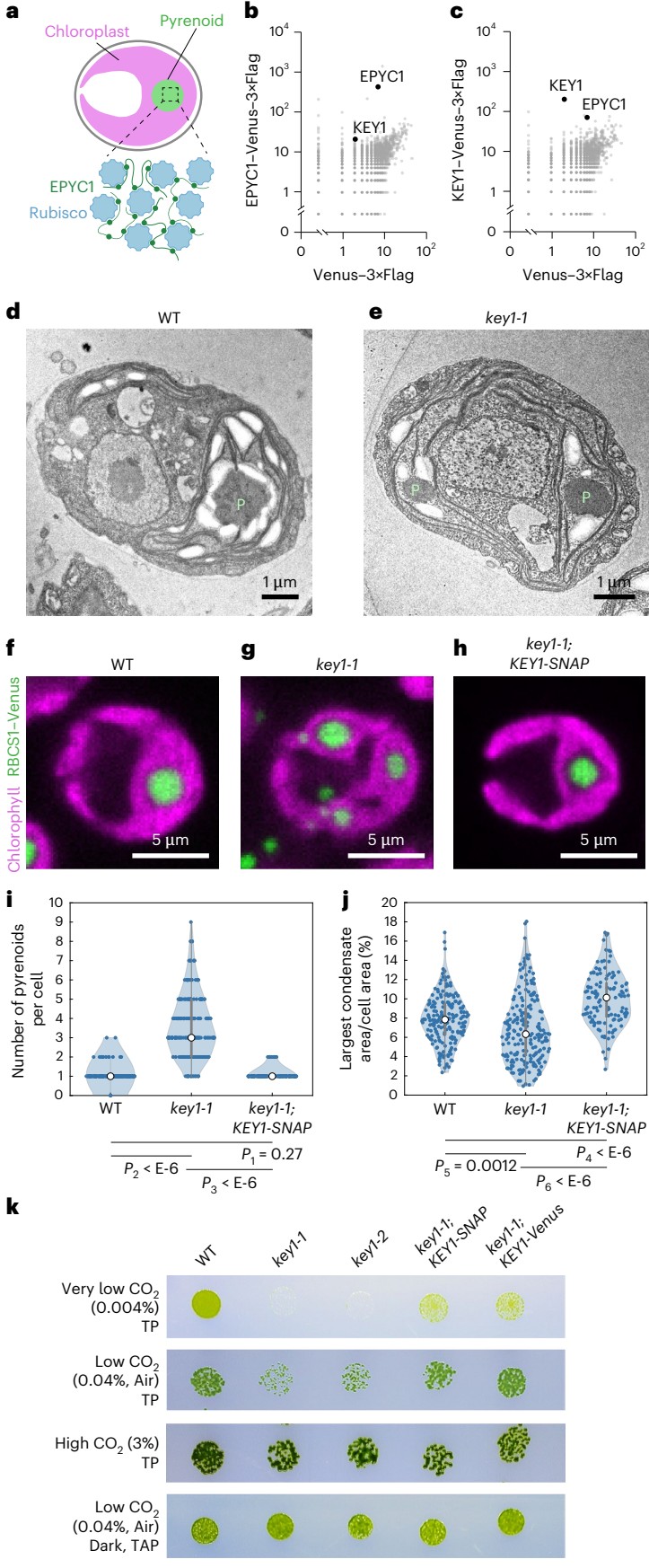

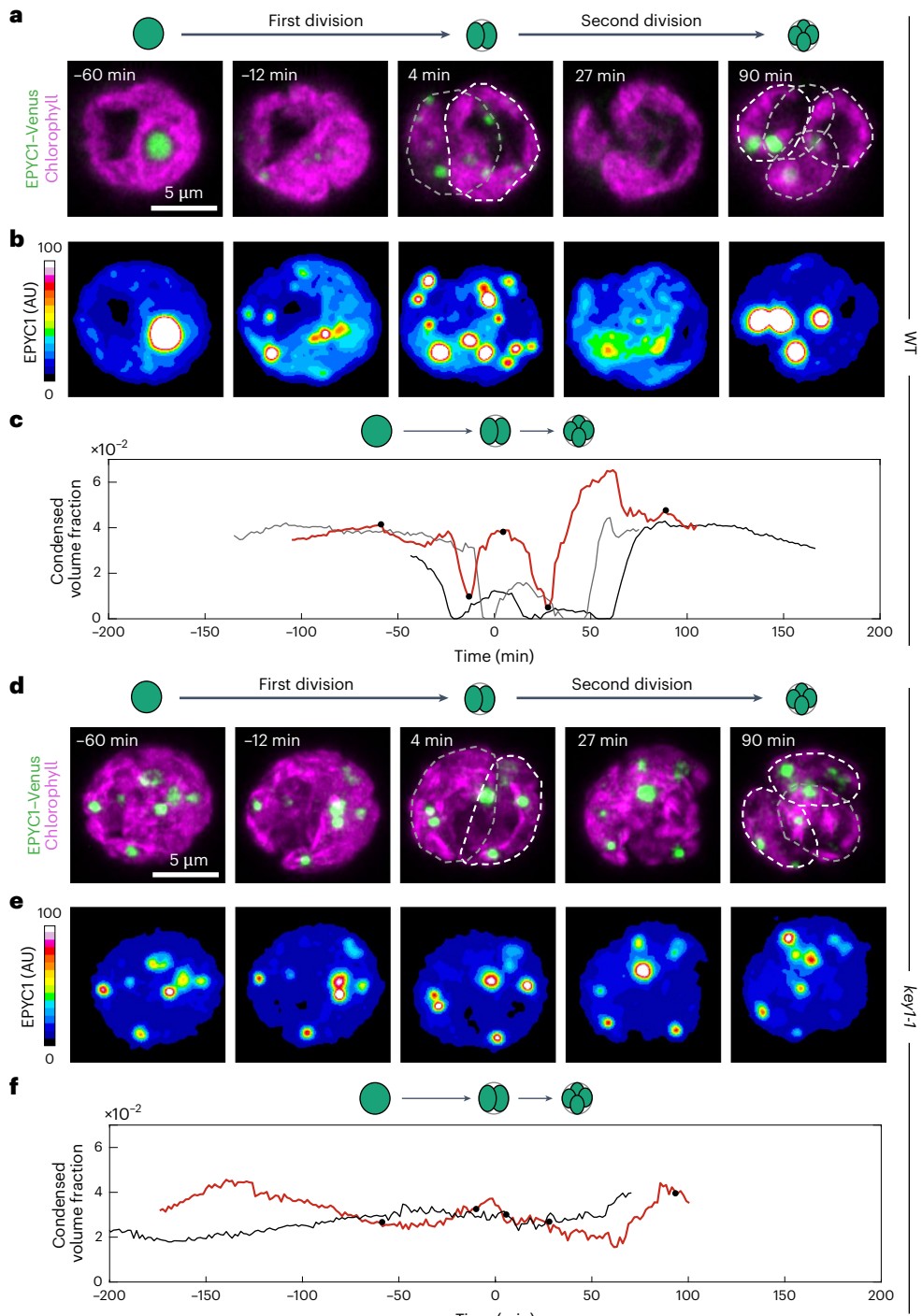

**Fig. 2 | KEY1 is necessary for pyrenoid condensate size, number and dissolution dynamics during cell division. a,b**, Timelapse microscopy of a dividing WT cell where the pyrenoid condensate is labelled by EPYC1–Venus (green, maximum z-projection) and the chloroplast is visualized through chlorophyll autofluorescence (magenta, mid-plane z) (**a**). The first cell division was completed at 0 min, and the second division was completed at 44 min, ending with four descendant cells. A heatmap allows visualization of EPYC1–Venus dissolution during cell division (**b**). **c**, The condensed volume fraction ($V_{densephase}/V_{chloroplast(s)}$) of EPYC1–Venus in WT cells throughout cell division for three representative parent cells. The red curve shows the cell in **a,b** with the time points marked as black dots. The first cell division occurs at 0 min for each cell. Protein concentration was assumed to be constant across the acquisition.

**d,e**, Timelapse microscopy of a dividing *key1-1* mutant cell with the pyrenoid condensate labelled by EPYC1–Venus (green, maximum z-projection) and the chloroplast visualized through chlorophyll autofluorescence (magenta, maximum z-projection) (**d**). The first cell division was completed at 0 min, and the second division was completed at 48 min, ending with four descendant cells. The heatmap shows EPYC1–Venus concentration (**e**). **f**, The condensed volume fraction of EPYC1–Venus in *key1-1* mutant cells throughout cell division for two representative cells. The red curve shows the cell in **d,e** with the time points marked as black dots. The first cell division occurs at 0 min for each cell. Protein concentration was assumed to be constant across the acquisition. See also Extended Data Figs. 2 and 3 and Supplementary Videos 2–5.

the pyrenoid dissolves and re-condenses, undergoing changes in size and number[52]. We observed cell division in wild-type cells whose cell cycles we synchronized using a diurnal light cycle (Extended Data Fig. 2a and Methods)[77–79]. Under our growth conditions, at the transition from light to dark, each wild-type mother cell typically divided twice in rapid succession to produce four descendant cells. During this process, as we reported previously[52], the pyrenoid condensate of a given mother cell typically underwent two sequential dissolution and condensation cycles, dissolving before each chloroplast division and condensing shortly after (Fig. 2a–c, Extended Data Fig. 2b–d and Supplementary Videos 2 and 3). Fluorescently tagged EPYC1 and Rubisco colocalized throughout pyrenoid dissolution dynamics in wild-type cells (Supplementary Video 4).

Notably, we frequently observed that during dissolution of the major condensate, new, smaller condensates appeared elsewhere in the chloroplast and grew (Fig. 2b and Supplementary Video 2), which we propose is a manifestation of an active biophysical system that regulates condensate size (see 'Modelling' section). After full dissolution, as previously described[52], we observed the appearance of multiple small condensates that gradually ripened to one condensate per chloroplast (Fig. 2a,b and Supplementary Video 2).

### KEY1 is necessary for pyrenoid size, number and dissolution dynamics during cell division

During cell division, key1-1 cells did not show any of the size, number or dissolution dynamics we observed in wild-type cells. The condensate sizes and numbers in key1-1 did not change appreciably over cell division (Fig. 2d–f and Supplementary Video 2). Moreover, key1-1 condensates failed to dissolve during cell division, as evidenced by the lack of diffuse material in the stroma outside the condensates when they were visualized by either EPYC1–Venus (Fig. 2d–f and Supplementary Video 2) or Rubisco–Venus (Extended Data Fig. 2e–g and Supplementary Video 3). Pyrenoid condensate size, number and dissolution dynamics were partially recovered in the key1-1;KEY1-SNAP rescue strain (Extended Data Fig. 2h–k); incomplete recovery could be due to non-native regulation of KEY1 or the presence of the SNAP tag and may explain the residual growth defect of key1-1;KEY1-SNAP at very low levels of $CO_2$ (Fig. 1k). These results indicate that KEY1 is necessary for normal pyrenoid condensate size, number and dissolution dynamics during cell division.

### KEY1 suppresses the appearance of ectopic condensates during growth

Pre-division key1-1 chloroplasts typically contained more than four condensates, which were distributed among the descendant cells so that, on average, the number of condensates per cell decreased during cell division (Fig. 2d,e). This indicates that the extra condensates observed in key1-1 must be produced during a different part of the diurnal growth cycle.

Indeed, we observed that small ectopic pyrenoid condensates formed in the chloroplast of the key1-1 cells during the light portion of the diurnal cycle and grew thereafter (Extended Data Fig. 3a,d,e and Supplementary Video 5). By contrast, in wild-type cells, a singular condensate per cell grew during this period[80] (Extended Data Fig. 3b,c,f,g). Thus, our results indicate that ectopic condensates in the key1-1 arise during cell growth and KEY1 inhibits their formation in wild-type cells by either preventing nucleation or dissolving them while they are too small to be detected by microscopy.

### KEY1 is necessary for EPYC1 phosphorylation

To understand how KEY1 regulates condensate size and suppresses ectopic condensates, we sought to characterize its molecular activity. We hypothesized that KEY1 phosphorylates EPYC1, based on KEY1's annotation as a protein kinase[69], its co-precipitation with EPYC1 (Fig. 1b,c), and the previous observations of EPYC1 phosphopeptides[65–68].

We first investigated the phosphorylation state of EPYC1 in key1 mutants. EPYC1 from the cell lysates of key1-1 and key1-2 showed higher mobility than EPYC1 from wild-type cell lysate in a Phos-tag gel-based western blot, which separates proteins based on their extent of phosphorylation[81,82] (Fig. 3a). Reintroducing KEY1 fused to a Venus or SNAP tag in key1-1 restored the lower-mobility EPYC1 bands observed in the wild type (Fig. 3a). Phosphatase treatment caused the EPYC1 band from wild-type lysates to run with higher mobility similar to that in key1-1, confirming that the observed lower mobility of EPYC1 in wild type is due to phosphorylation (Fig. 3b). By contrast, phosphatase treatment of key1-1 lysate led to no observable shift in mobility, establishing that EPYC1 has no detectable phosphorylation in key1-1 (Fig. 3b). Together, these observations indicate that KEY1 is necessary for EPYC1 phosphorylation in vivo.

We sought to determine whether the phenotypes of key1-1 are due to deficient phosphorylation of EPYC1 specifically, as KEY1 may also phosphorylate other proteins. We transformed an epyc1 mutant strain, which still had wild-type KEY1, with a construct encoding a mutant EPYC1 where all the serines and threonines were changed to alanines (Extended Data Fig. 4a). In this epyc1;EPYC1$^{phosphonull}$-Venus line, wild-type KEY1 could still phosphorylate other potential targets, but it cannot phosphorylate the mutant EPYC1. We observed that this strain exhibited multiple condensates (Extended Data Fig. 4b–d) similar to the pyrenoid condensates of the key1-1;EPYC1–Venus strain (Fig. 1g), indicating that the multiple-pyrenoid phenotype in key1-1 cells is due to defects in EPYC1 phosphorylation. These results indicate that KEY1 regulates pyrenoid size, number and dynamics by directly or indirectly phosphorylating EPYC1.

### KEY1 directly phosphorylates specific sites on EPYC1

To characterize KEY1's activity in vitro, we used purified KEY1 from Escherichia coli (Extended Data Fig. 5a,b). This purified KEY1 was active,

---

**Fig. 3 | KEY1 directly phosphorylates specific sites on EPYC1. a**, Anti-EPYC1 western blot based on Phos-tag gel of lysates of unsynchronized WT, key1 mutants and key1-1 rescued strains. The experiment was performed three times with similar results. **b**, Anti-EPYC1 western blot based on Phos-tag gel of lysates of unsynchronized WT and key1-1, with or without the addition of Lambda phosphatase or E. coli-expressed KEY1. The experiment was performed once and was consistent with results in a,c. **c**, Coomassie-stained Phos-tag gel of purified EPYC1–GFP incubated with or without purified KEY1. Both proteins were expressed in E. coli. The experiment was performed three times with similar results. **d**, Phosphorylation pattern of EPYC1 from unsynchronized WT Chlamydomonas cells analysed by mass spectrometry. Residues with yellow backgrounds represent the identified phosphorylation sites. The grey block highlights previously identified Rubisco-binding regions. **e–g** Phosphorylation pattern of EPYC1–GFP purified from E. coli treated with a large amount (3 μM) (**e**) or small amount (115 nM) (**f**) of KEY1 or untreated (**g**). Residues with yellow backgrounds represent phosphorylation sites identified

in at least one experiment; bolded residues indicate sites identified in both replicate experiments. Untreated EPYC1–GFP from E. coli contained several phosphorylation sites not observed in EPYC1 from Chlamydomonas. **h,i**, Spectral counts of unphosphorylated and phosphorylated versions of two EPYC1 peptides identified by mass spectrometry from EPYC1–GFP purified from E. coli with different treatments. The positions of these peptides are indicated in **d–g** with straight underlines (**h**) or wavy underlines (**i**). **j**, Chlamydomonas cells were synchronized in a diurnal cycle, during which they grow in the light cycle and divide twice in rapid succession upon the shift from light to dark each day. **k**, Cell lysates from a synchronized culture of WT cells were run on a Phos-tag gel and probed for EPYC1. Lower-mobility bands correspond to a more highly phosphorylated form of EPYC1. The experiment was performed three times with similar results. **l**, The mRNA expression level of KEY1 during a day, based on transcriptome data reported in Strenkert et al.[79] The time points in **j** to **l** are aligned. See also Extended Data Figs. 4 and 5.

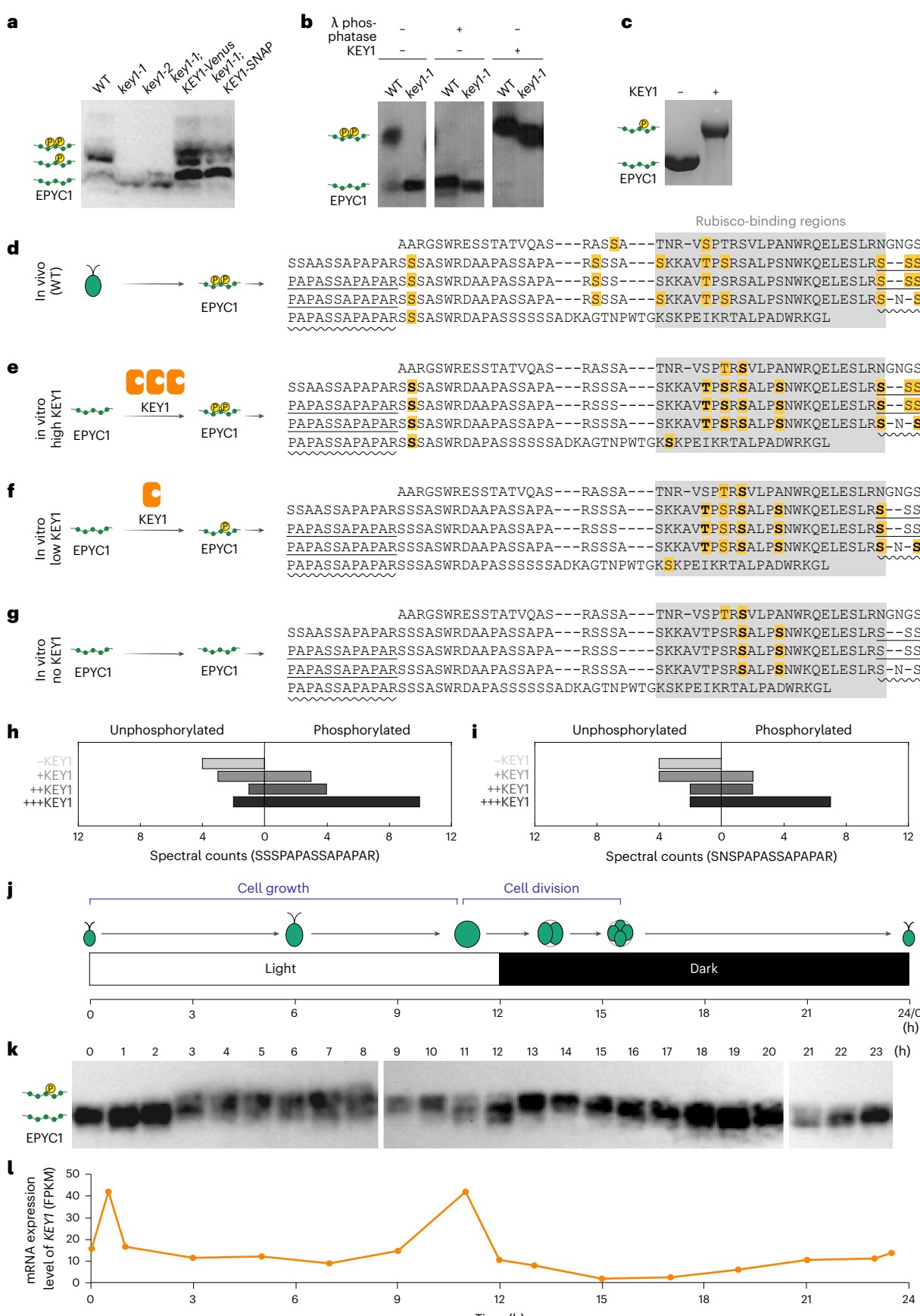

as treating the *key1-1* cell lysate with it caused EPYC1 to run at lower mobility similar to that observed in wild-type cells (Fig. 3b). To determine whether KEY1 can directly phosphorylate EPYC1, we incubated this purified KEY1 with purified tagged EPYC1 from *E. coli* in the presence of ATP[58,83]. The KEY1-treated EPYC1 migrated more slowly on a Phos-tag gel than mock-treated EPYC1 (Fig. 3c and Extended Data Fig. 5c,d), indicating that KEY1 directly phosphorylates EPYC1 in vitro.

As the EPYC1–Rubisco-binding interface is known[59], we wondered whether KEY1 phosphorylation impacts this interface. We used mass spectrometry to determine the phosphorylation sites on EPYC1–Venus–3×Flag purified from unsynchronized cells with a wild-type background (Fig. 3d). The results identified 13 distinct phosphorylated EPYC1 peptides. Because EPYC1 is a repeat protein, some of these peptides align onto multiple sites of the EPYC1 sequence; thus, our data indicate that up to 24 serines or threonines on EPYC1 were phosphorylated. Thirteen of the phosphorylated EPYC1 residues were on or directly adjacent to the Rubisco-binding regions (Fig. 3d)[59], suggesting that EPYC1 phosphorylation could impact EPYC1–Rubisco binding.

When we phosphorylated purified EPYC1–GFP in vitro with 3 µM purified KEY1 (Extended Data Fig. 5d), we observed a similar pattern of phosphorylation (Fig. 3e,g) to what we had observed on EPYC1 phosphorylated in vivo (Fig. 3d), indicating that KEY1 can directly phosphorylate nearly all the sites observed to be phosphorylated in vivo. When we treated EPYC1–GFP in vitro with a lower (115 nM) concentration of KEY1 (Extended Data Fig. 5d), we observed preferential phosphorylation at sites in the Rubisco-binding region (Fig. 3f,g). For peptides where we observed both unphosphorylated and phosphorylated variants, the proportion of phosphorylated variants increased with the concentration of KEY1 (Fig. 3h,i). Together with the observed lack of EPYC1 phosphorylation in the *key1* mutants (Fig. 3a,b), these results establish that KEY1 is the primary kinase of EPYC1 and suggest that KEY1 preferentially phosphorylates the Rubisco-binding regions of EPYC1.

### The extent of EPYC1 phosphorylation changes diurnally

If the KEY1-mediated EPYC1 phosphorylation underlies the dynamic diurnal phase behaviours of the pyrenoid condensate, we would expect the extent of EPYC1 phosphorylation to change over the course of the diurnal cycle. Indeed, we observed diurnal changes in the extent of EPYC1 phosphorylation in cells grown in a 12-h light–dark cycle (Fig. 3j,k). EPYC1 was minimally phosphorylated from the middle of the night through the early morning (hours 18 to 2 of the cycle), with a single high-mobility band. During the day from mid-morning to dusk (hours 3 to 12) EPYC1 was phosphorylated at intermediate levels, with both high- and low-mobility species. At the beginning of the night (hours 13 to 14 of the cycle), the phosphorylation level reached its maximum, with a single low-mobility band. We speculate that these diurnal changes in the extent of EPYC1 phosphorylation reflect a lower level of phosphorylation needed to dissolve ectopic condensates during growth during the day (hours 3 to 12; Fig. 3k) and a higher level of phosphorylation needed to dissolve the pyrenoid condensate during cell division (hours 13 to 14; Fig. 3k).

The two diurnal increases in the extent of EPYC1 phosphorylation (at hours 3 and 13) were each preceded by a peak in *KEY1* mRNA levels measured in a previous study: one in the early morning (hour 0.5) and the other at the end of the day (hour 11; Fig. 3l)[79]. This observation suggests that transient increases in *KEY1* mRNA may result in increases in total KEY1 activity, which could contribute to the observed changes in the EPYC1 phosphorylation extent.

### KEY1-phosphorylated EPYC1 promotes the dissolution of Rubisco–EPYC1 condensates in vitro

The requirement of KEY1 for pyrenoid condensate dissolution during cell division (Fig. 2 and Extended Data Fig. 2) and the observation that KEY1 is the primary kinase of EPYC1 (Fig. 3a–i) suggested

that phosphorylated EPYC1 promotes pyrenoid condensate disassembly. To test this idea, we used a previously established assay[58,83] to measure the in vitro phase diagrams of Rubisco combined with mock-treated or KEY1-phosphorylated EPYC1–GFP (Fig. 4a–d). Our mock-treated EPYC1–GFP was minimally phosphorylated (Fig. 3c,g and Extended Data Fig. 5d) and formed condensates in agreement with our previous study (Fig. 4c)[83]. By contrast, EPYC1–GFP treated with KEY1 did not phase separate at any tested concentration (Fig. 4d). We conclude that phosphorylation of EPYC1 by KEY1 promotes the dissolution of Rubisco–EPYC1 condensates.

### Phosphorylation of EPYC1 by KEY1 disrupts EPYC1 binding to Rubisco

We next sought to determine how EPYC1 phosphorylation promotes dissolution of the Rubisco–EPYC1 condensate. Our previous study determined that small complexes can form between an individual Rubisco and one or more EPYC1 molecules in the dilute phase (Fig. 4e)[83]. Thus, one possibility is that EPYC1 phosphorylation could promote the formation of such small Rubisco–EPYC1 complexes (Fig. 4e; hypothesis 1), which would inhibit the interaction network required for condensation[52,83,84]. An alternative hypothesis is that EPYC1 phosphorylation could favour dissolution of the condensate by disrupting binding between EPYC1 and Rubisco (Fig. 4e; hypothesis 2).

To distinguish between these two hypotheses, we studied the effect of phosphorylation on EPYC1 binding to Rubisco in the dilute phase using Fluorescence Correlation Spectroscopy[83] (Methods). In this assay, the binding of EPYC1–GFP (60 kDa) to Rubisco (550 kDa) can be detected as a decrease in the diffusion coefficient of EPYC1–GFP[83]. The diffusion coefficient of unphosphorylated EPYC1–GFP decreased from 60 µm² s⁻¹ to 41 µm² s⁻¹ when Rubisco was added (Fig. 4f), indicating that EPYC1 and Rubisco bound and formed small Rubisco–EPYC1 complexes, as expected from previous findings[83]. However, when we repeated the experiment with KEY1-phosphorylated EPYC1–GFP, we observed no change in the diffusion coefficient of EPYC1 when Rubisco was added, indicating no formation of small Rubisco–EPYC1 complexes (Fig. 4f). These findings show that KEY1 phosphorylation of EPYC1 directly disrupts the binding between Rubisco and EPYC1 to favour condensate dissolution (Fig. 4e; hypothesis 2).

### Phosphorylated EPYC1 is enriched in the dilute phase in vivo

Further consistent with our observations, we found that in vivo, unphosphorylated EPYC1 is enriched in the pyrenoid condensate, while phosphorylated EPYC1 is enriched in the dilute phase. We mechanically lysed unsynchronized wild-type cells and centrifuged the cell lysate to separate the pyrenoid condensate in the pellet from the dilute phase in the supernatant (Fig. 4g)[55,58,70]. We observed an enrichment of unphosphorylated EPYC1 in the pellet and an enrichment of phosphorylated EPYC1 in the soluble supernatant (Fig. 4g). This result suggests that KEY1-phosphorylated EPYC1 partitions into the dilute phase.

### KEY1 localizes to the pyrenoid condensates throughout the cell cycle

Depending on where KEY1 localizes, the kinase could influence the size, number and dissolution of the pyrenoid condensates via different biophysical mechanisms: (1) KEY1 could localize to the dilute phase (throughout the chloroplast stroma outside the pyrenoid condensate) to control directly the amount of phosphorylated EPYC1; (2) KEY1 could relocalize from the dilute phase to the condensate specifically during cell division, to induce dissolution at that time; and (3) KEY1 could localize to the condensate at all times and induce dissolution through changes in concentration or activity. To distinguish these possibilities, we investigated the dynamic subcellular localization of KEY1. In unsynchronized cultures, KEY1–SNAP localized to the pyrenoid and sometimes also to its periphery (Fig. 5a). In synchronized cultures, fluorescently tagged KEY1 localized to the pyrenoid condensate throughout

the cell cycle (Extended Data Fig. 6). Its persistent localization to the condensate suggests that KEY1 regulates pyrenoid dissolution and re-condensation not through changes in its localization but instead through temporal changes in its concentration and/or activity.

### A Rubisco-binding motif localizes KEY1 to condensates and is necessary for function

We hypothesized that KEY1 is targeted to the condensate by its predicted Rubisco-binding motif (Fig. 5b,c and Supplementary Video 6)[85]. Many pyrenoid-localized proteins, including EPYC1, possess a common Rubisco-binding motif[85]. This motif was previously found to be necessary for targeting a pyrenoid-resident protein to the pyrenoid and was sufficient for enriching a chloroplast stromal protein in the pyrenoid[85]. Using surface plasmon resonance, we detected that a peptide containing KEY1's predicted Rubisco-binding motif could bind to Rubisco, indicating that KEY1 contains a bona fide Rubisco-binding motif that could contribute to KEY1's localization to the pyrenoid condensate (Fig. 5d and Extended Data Fig. 7).

To test whether the motif was necessary for KEY1's localization, we expressed KEY1–SNAP with a mutated Rubisco-binding motif in the *key1-1* strain expressing RBCS1–Venus (Fig. 5a). In this *key1-1;RBCS1-Venus;KEY1^ΔRBM-SNAP* strain, KEY1^ΔRBM signal did not partition into the pyrenoid condensates, indicating that the Rubisco-binding motif is necessary for KEY1 localization to the pyrenoid condensates (Fig. 5a and Extended Data Fig. 8a–e).

To determine whether the localization of KEY1 affects its kinase activity on EPYC1, we analysed EPYC1 phosphorylation in the *key1-1;RBCS1-Venus;KEY1^ΔRBM-SNAP* strain, and observed decreased phosphorylation of EPYC1 relative to the control expressing KEY1 with a wild-type Rubisco-binding motif (Fig. 5e). Consistent with the decreased phosphorylation of EPYC1, the multiple-pyrenoid phenotype was not rescued in this strain (Fig. 5a and Extended Data Fig. 8f). To verify that the mutation did not substantially impact KEY1's kinase activity, we confirmed that KEY1^ΔRBM is still able to phosphorylate EPYC1 (Extended Data Fig. 8g). These results indicate that correct localization of KEY1 is needed for its normal activity in phosphorylating EPYC1 and maintaining a singular pyrenoid condensate.

Taking sum of our experimental data, we propose a model for KEY1's regulation of EPYC1 phosphorylation and its partitioning between the condensed and dilute phases (Fig. 5f). Unphosphorylated EPYC1 forms condensates with Rubisco. A Rubisco-binding motif localizes KEY1 to the pyrenoid condensate throughout the cell cycle, where it phosphorylates EPYC1, decreasing its binding affinity to Rubisco and promoting its partitioning to the dilute phase. We speculate that at least

one unidentified phosphatase acts on EPYC1 in the dilute phase, promoting its partitioning into the condensate. While the identification of such a phosphatase is beyond the scope of the present study, evidence supporting its existence includes the presence of low-phosphorylation EPYC1 species at various time points during the diurnal cycle (Fig. 3k) and the rapid re-condensation of the pyrenoid condensate after cell division (Fig. 2a–c and Supplementary Videos 2 and 3).

### Modelling suggests potential mechanisms underpinning KEY1 regulation of pyrenoid size, positioning and phase dynamics

We sought to understand how the local molecular interactions of EPYC1, KEY1 and the putative EPYC1 phosphatase could produce the observed pyrenoid condensate size and number dynamics. We hypothesized that the molecular interactions between these components could produce an active system with the emergent property of size control through the previously theorized enrichment-inhibition mechanism[41]. In this

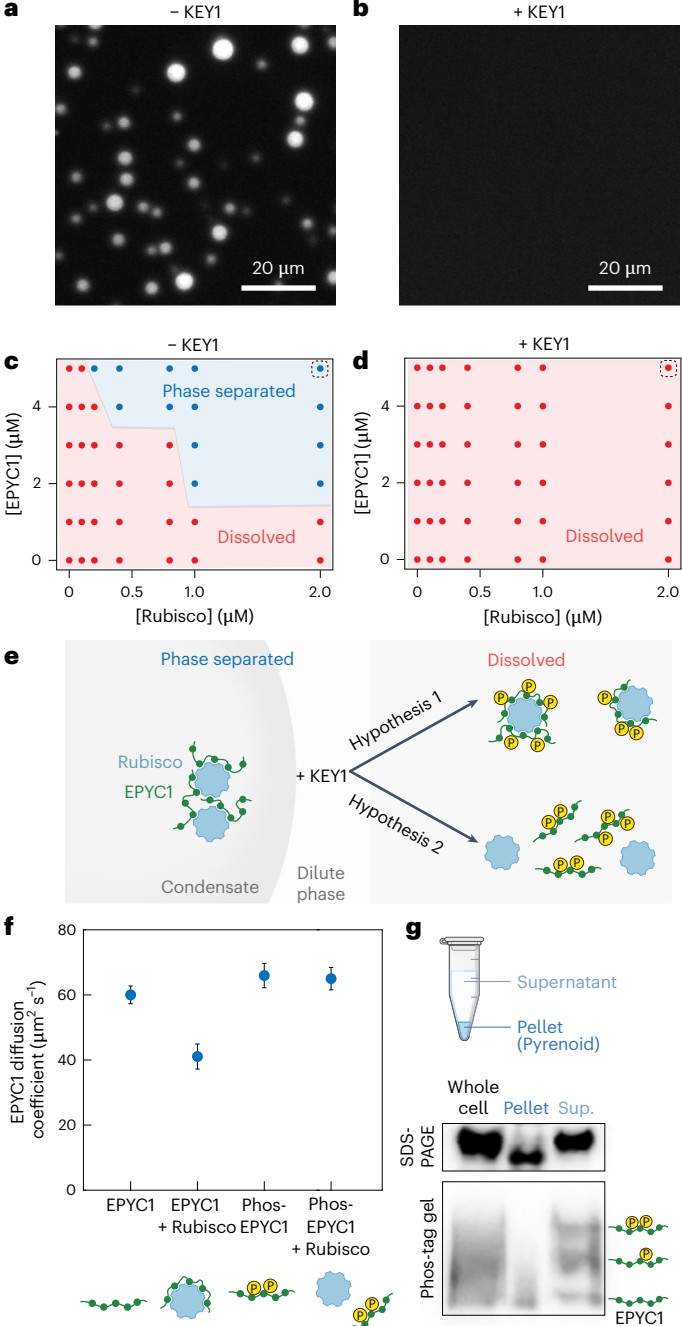

**Fig. 4 | EPYC1 phosphorylation promotes the dissolution of Rubisco–EPYC1 condensates by disrupting Rubisco–EPYC1 binding. a,b,** 9 μM *E. coli*-purified EPYC1–GFP was untreated (**a**) or treated with 3.6 μM KEY1 (**b**) before being mixed with 2 μM Rubisco with a final concentration of 5 μM EPYC1–GFP and imaged. **c,d,** Concentration-dependent phase diagram of unphosphorylated EPYC1–GFP (**c**) or EPYC1–GFP phosphorylated by 3.6 μM KEY1 (**d**) along with Rubisco. Dashed boxes indicate samples shown in **a,b. e,** Schematic of the two possible mechanisms for KEY1-mediated EPYC1 phosphorylation that promote dissolution of the EPYC1–Rubisco condensate. In hypothesis 1, phosphorylation of EPYC1 promotes the formation of small clusters of a single Rubisco and several EPYC1 molecules. In hypothesis 2, phosphorylation of EPYC1 disrupts the interactions between EPYC1 and Rubisco. **f,** The diffusion coefficient of unphosphorylated or phosphorylated full-length EPYC1 tagged with GFP in the presence of Rubisco was measured using fluorescence correlation spectroscopy. Blue circles represent the mean diffusion coefficient from two experimental replicates in which three fluorescence correlation spectroscopy measurements were made. The error bars represent the s.d. The cartoon models underneath the *x* axis show our interpretation of the results. **g,** Whole-cell lysate was fractionated into condensed phase (pyrenoid pellet) and the dilute phase (supernatant; sup.). Fractions were separated by SDS–PAGE (top) and Phos-tag (bottom) gels and probed for EPYC1 by western blot.

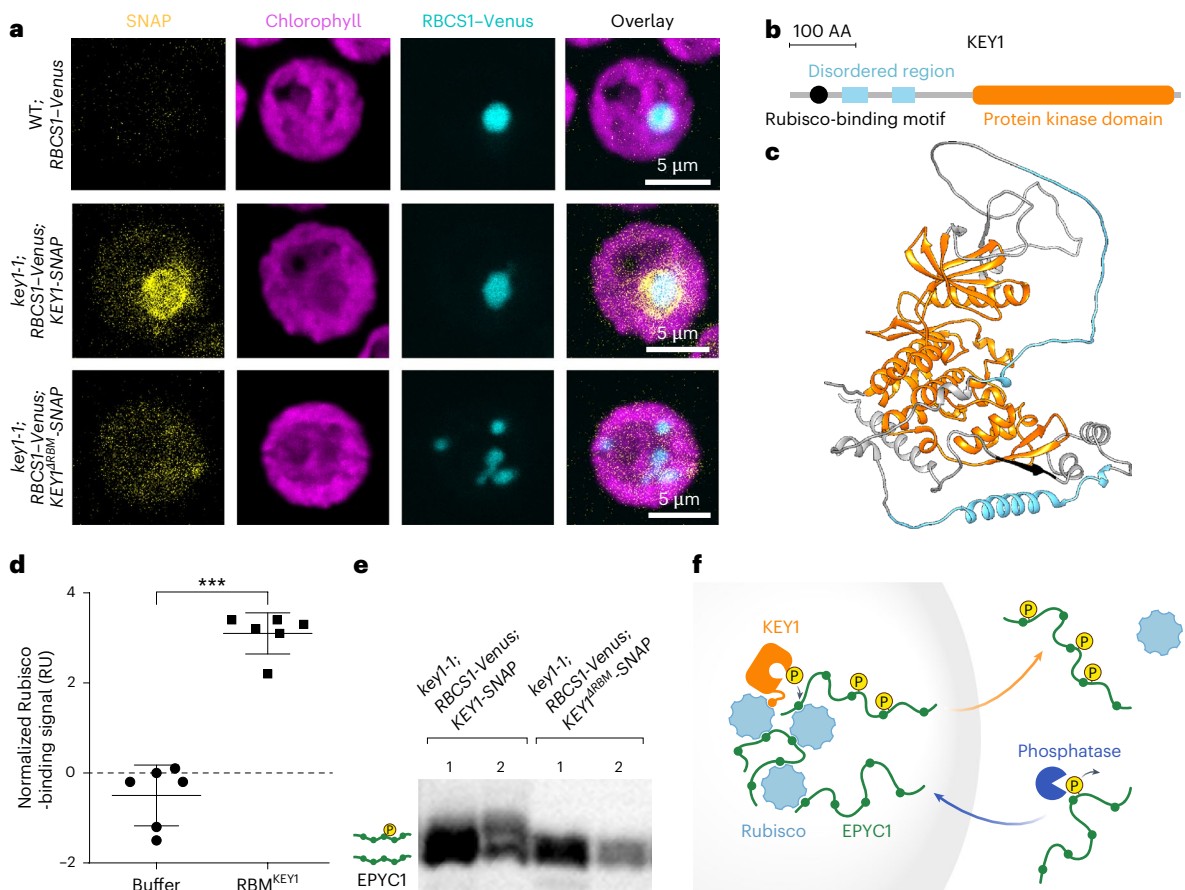

**Fig. 5 | A Rubisco-binding motif localizes KEY1 to the pyrenoid condensate to mediate function. a**, Confocal fluorescence images of cells from unsynchronized cultures of WT, showing the background signal of SNAP dye staining (top row); *key1-1* with RBCS1–Venus rescued by KEY1-SNAP, showing representative localization of KEY1 (middle row); and *key1-1* with RBCS1–Venus expressing KEY1ᴬᴿᴮᴹ–SNAP, showing representative localization of KEY1 with a five-amino-acid mutation disrupting the Rubisco-binding motif (bottom row). Magenta shows chlorophyll autofluorescence, cyan shows RBCS1–Venus and yellow shows SNAP. See also Extended Data Fig. 6. **b**, Predicted KEY1 domains (based on the UniProt dataset[92]). The Rubisco-binding motif is in black, the disordered regions are shown in cyan, and the protein kinase domain is shown in Orange. **c**, KEY1 three-dimensional structure predicted by AlphaFold. The colour key is the same as in **b**. See also Supplementary Video 6. **d**, The binding to Rubisco of a peptide representing KEY1's Rubisco-binding motif (RBMᴷᴱʸ¹) was assayed

using surface plasmon resonance in comparison to a buffer-only control. Each dot indicates a separate surface (Methods), *n* = 6 independent channels containing experimental and reference surfaces. Centre line indicates mean; error bars indicate s.d. \*\*\**P* = 0.00022, paired *t*-test. **e**, EPYC1 phosphorylation was assayed using anti-EPYC1 western blot of lysates from *key1-1;KEY1-SNAP* and *key1-1;KEY1ᴬᴿᴮᴹ-SNAP* run on a Phos-tag gel. **f**, Cartoon of the proposed molecular mechanism of KEY1. In the pyrenoid condensate, KEY1 phosphorylates EPYC1, disrupting interactions between EPYC1 and Rubisco and allowing phosphorylated EPYC1 to diffuse out of the condensate. We speculate that a phosphatase dephosphorylates EPYC1 in the dilute phase outside the pyrenoid condensate, promoting its interaction with Rubisco and favouring the entry of unphosphorylated EPYC1 into the condensate. See also Extended Data Figs. 6–8 and Supplementary Video 6.

mechanism, a kinase localizes to the condensate and converts its building blocks into a dissolution-promoting form (in our case, phosphorylated EPYC1), while a phosphatase localized to the dilute phase converts the building blocks into a condensation-promoting form (in our case, unphosphorylated EPYC1), resulting in a stable condensate size determined by the ratio of kinase to phosphatase activities.

To investigate this hypothesis, we developed a minimal continuum mathematical model of pyrenoid condensate dynamics. For simplicity, we represented the EPYC1–Rubisco system through EPYC1 only, capturing the propensity of unphosphorylated EPYC1 to cluster via interaction with Rubisco as an effective self-interaction (Methods). Specifically, we considered a suspension of EPYC1 in a solvent, where EPYC1 can exist in two states: (1) unphosphorylated EPYC1, which we denote as 'sticky' and (2) phosphorylated EPYC1, which we denote as 'non-sticky' (Fig. 6a). Sticky EPYC1 tends to condense via self-attractive interactions and form clusters. Non-sticky EPYC1 does not condense and thus tends to dissolve the clusters, favouring a configuration where EPYC1 is uniformly dispersed. The diffusive fluxes of sticky and

non-sticky EPYC1 are driven by gradients of their chemical potentials, which we obtained from Flory–Huggins free energies (Methods).

To model the phosphorylation of EPYC1 by KEY1, we introduced a rate of switching from sticky to non-sticky EPYC1. To model the activity of the unidentified phosphatase, we also included a rate of switching from non-sticky to sticky (Fig. 6a). We assumed that these switching rates are spatially uniform and depend only on time, for example, to model changes of KEY1 levels during the cell cycle. Thus, our model is simpler than the enrichment-inhibition mechanism[41] in which switching rates vary in space. We also assumed that the diffusion coefficients of both sticky and non-sticky EPYC1 are equal and constant (Methods). Thus, in this minimal model, for a fixed amount of EPYC1 with a fixed self-attractive interaction strength, the dimensionless parameters controlling the phase behaviours of EPYC1 are the ratio between kinase and phosphatase activities and the ratio between the switching times and the time for EPYC1 to diffuse over the chloroplast (Methods).

We performed numerical simulations in a two-dimensional geometry with no-flux boundaries, mimicking the shape of the

*Chlamydomonas* chloroplast (Supplementary Videos 7–13 and Methods). Additionally, we conducted simulations using a square geometry with periodic conditions and obtained similar results (Supplementary Video 14 and Methods).

**Dissolution, re-condensation and size control during cell division.** We tested if this simple model could recapitulate the observed phase behaviours of the pyrenoid condensate during cell division. From the peak in *KEY1* mRNA expression immediately preceding cell division (Fig. 3l)[79] and EPYC1 phosphorylation during cell division (Fig. 3k; hours 13 to 14), we speculated that KEY1 activity increases leading up to cell division and declines thereafter. Thus, in the model we varied the kinase activity, the switching rate from sticky to non-sticky, following this inferred pattern (Fig. 6b). We observed that an initially stable large condensate fully dissolved, and eventually re-condensed again into a singular condensate (Fig. 6c and Supplementary Video 7), recapitulating what we observed in vivo (Fig. 2a–c). These results support the idea that the observed dissolution and condensation during cell division could be driven by changes in the EPYC1 phosphorylation rate.

Notably, as observed in vivo (Fig. 2a,b), the dissolution and re-condensation each proceeded via intermediates that contained multiple smaller condensates (Fig. 6c(ii),(iv),(v) and Supplementary Video 7), which we propose is the manifestation of the EPYC1–KEY1–phosphatase system acting as a condensate size regulation mechanism. A state of multiple stable condensates has been observed theoretically in previous models of active condensates and experimentally in reactive mixtures, where the presence of nonequilibrium chemical reactions between system components can suppress coarsening[41–47,51,86]. In our system, competing kinase and phosphatase activities establish a preferred condensate radius where the flux of non-sticky EPYC1 out of each condensate ($\sim R^3$) balances the flux of sticky EPYC1 into each condensate ($\sim R$) (Fig. 6d, Supplementary Video 8 and Methods)[41]. The exact value of this preferred radius is determined by EPYC1 diffusivity and kinase/phosphatase activity ratio (Methods). Leading up to cell division, the kinase/phosphatase activity ratio increases, progressively decreasing the preferred condensate radius and favouring a state with multiple smaller condensates before EPYC1 is fully dissolved (Fig. 6c(iv),d(ii) and Supplementary Video 7). Specifically, as the initial single condensate shrinks toward the smaller preferred radius, the efflux of EPYC1 from this condensate leads to the buildup of a high concentration of EPYC1 away from the condensate, which is converted to the sticky form by the phosphatase. Once the concentration of sticky EPYC1 in the dilute phase is sufficiently high, a new condensate forms by spinodal decomposition. Following cell division, as KEY1 activity decreases, the preferred condensate radius progressively increases, ultimately causing the system to coarsen into a single condensate via Ostwald ripening

(Extended Data Fig. 9). As a consequence of kinase and phosphatase activities, the distribution of sticky EPYC1 both inside and outside the condensates is spatially nonuniform––a signature of active systems (Fig. 6d(i),(ii))[46]. Thus, our findings suggest that pyrenoid dynamics and size control are consequences of the system being active, with energy injected via phosphorylation/dephosphorylation reactions.

**Self-centring and inter-condensate repulsion.** In our model, as a consequence of the active mechanism described above, we found that the fluxes of molecules due to kinase and phosphatase activity can cause the condensate to self-centre within the boundaries of the simulation (Fig. 6c(vi) and Supplementary Video 7). To see this phenomenon clearly, we initialized a singular condensate in the arm of the chloroplast (Fig. 6e and Supplementary Video 9). Over time, the condensate moved to the centre of the chloroplast. In contrast, without kinase and phosphatase activity, the condensate remained in the arm of the chloroplast over the same time scale (Supplementary Video 10). Of note, the position favoured by the active model matches the in vivo position of the pyrenoid condensate in the chloroplast (Figs. 1d,f and 6e). This result suggests that KEY1-driven phosphorylation of EPYC1 may contribute to proper pyrenoid localization within the chloroplast.

We also observed that our simulated condensates seemed to repel each other. Indeed, when we initialized two condensates close together with a finite phosphorylation rate, we saw them move away from each other such that they were eventually evenly spaced within the chloroplast geometry (Fig. 6f and Supplementary Video 11). In the model, the self-centring and repulsive phenomena are due to the same underlying mechanism: phosphorylated, non-sticky EPYC1 departs a condensate symmetrically, but after diffusion and dephosphorylation within the confining geometry, the influx of sticky EPYC1 is highest from the side of the condensate farthest from the chloroplast boundaries or from other condensates. This mechanism for condensate interactions may extend to other regulated biomolecular condensates.

**Dissolution of ectopic condensates.** Finally, we also explored whether our minimal model could recapitulate the observed suppression of small ectopic clusters mediated by KEY1 activity, compared with mutant cells lacking KEY1, which cannot dissolve them (Extended Data Fig. 3b–e). To this end, we considered that the absence of KEY1 not only implies negligible switching of EPYC1 to a non-sticky state, but also leads to anomalously low EPYC1 phosphorylation, thus increasing the EPYC1 self-attraction strength (Methods). The presence of KEY1 at low activity leads to the dissolution of a small ectopic cluster (Fig. 6g and Supplementary Video 12), whereas in the absence of KEY1, EPYC1 self-interactions are too strong to allow the dissolution of the

**Fig. 6 | Modelling supports the role of KEY1 in regulating pyrenoid condensate size and number and reveals a centring mechanism. a**, Overview of our model: we represent the EPYC1–Rubisco system through EPYC1 only, with EPYC1 existing in two states: unphosphorylated (sticky) and phosphorylated (non-sticky). The rates of switching between these states are mediated by a kinase (KEY1) and a phosphatase. For simplicity, we assume both the switching rates to be spatially uniform, and the dephosphorylation rate to be constant. **b**, To recapitulate the hypothesized temporal changes in KEY1 activity during cell division, the EPYC1 phosphorylation rate is varied over time. **c**, The computed spatiotemporal phase behaviours are shown as snapshots (times indicated by dots in **b**). The colour bar indicates EPYC1 concentration for **c**–**g**. At a low phosphorylation rate, a single EPYC1 cluster forms (i). At intermediate phosphorylation rate, EPYC1 forms small stable clusters that do not coarsen (ii). At a very high phosphorylation rate, the stable cluster dissolves as most EPYC1s are non-sticky (iii). As the phosphorylation rate is reduced, multiple stable small clusters re-appear (iv) and coarsen into a single cluster (v and vi). **d**, Due to KEY1 activity, the system exhibits a size control mechanism at intermediate phosphorylation rates (rate same as in **c**(iv)). The model is initialized with two EPYC1 clusters of different sizes (i). After a

finite time, both clusters equilibrate, reaching the same stable size (ii). The sticky and non-sticky EPYC1 volume fractions (top) and radially directed influx and outflux (bottom) across one of the condensates are plotted for time points i and ii. The clusters reach a stable size when the inflow and outflow of sticky and non-sticky EPYC1, respectively, become balanced (ii; Methods). The grey horizontal line indicates the maximum volume fraction reached by non-sticky EPYC1 in the dense phase. **e**, The model is initialized with an off-centre condensate, which self-centres into the canonical position farthest from the boundaries (KEY1 level lower than in **d** and **f**). **f**, The model is initialized with two condensates that are initially placed too close to each other. They move apart over time, exhibiting repulsive behaviour. **g**, With no KEY1 activity, EPYC1 is modelled as overly sticky, which prevents coarsening over time through Ostwald ripening (top). Low KEY1 activity, in addition to mediating switching, is modelled as reducing the magnitude of sticky self-attractive interactions, which allows ectopic clusters to dissolve (bottom). **h**, Summary of our understanding of how KEY1 activity levels impact the phase behaviours and size of pyrenoid condensates. See also Extended Data Fig. 9 and Supplementary Videos 7–14.

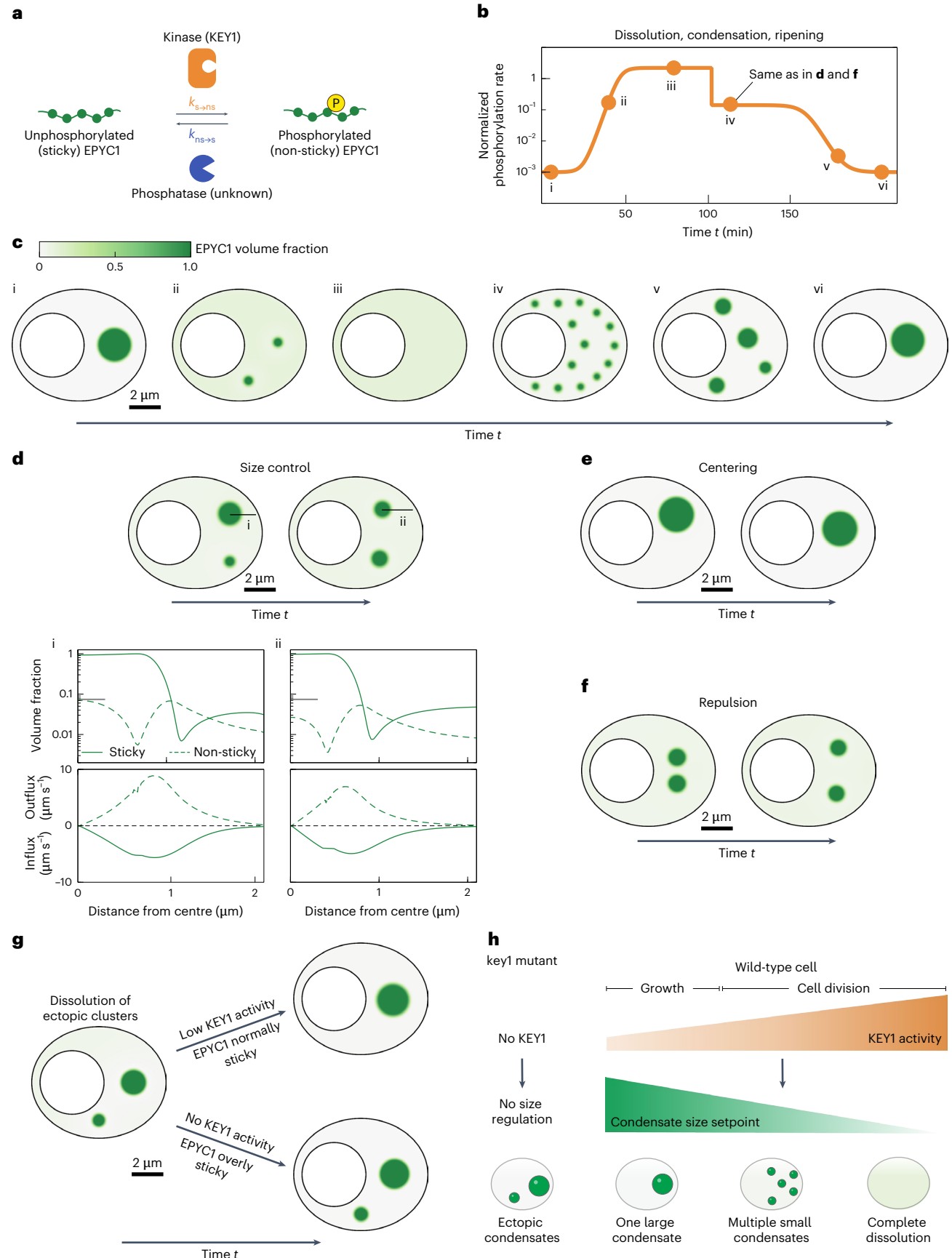

ectopic cluster within the same time window (Extended Data Fig. 9 and Supplementary Video 13). These results support the idea that a low level of KEY1 activity could mediate the dissolution of ectopic condensates during cell growth to maintain a single pyrenoid and can explain the origin of the ectopic condensates that lack size control in the *key1-1* mutant (Extended Data Fig. 3b–e).

Thus, while the multiple condensates exhibited by the *key1-1* mutant resemble the multiple condensates seen during cell division in wild-type cells, they arise by distinct mechanisms. The multiple condensates exhibited by the *key1-1* mutant result from the effective absence of Ostwald ripening as new material is added to the system, whereas the multiple condensates seen during cell division in wild-type cells result from an active size control mechanism mediated by KEY1 (Fig. 6h).

## Discussion

We identified a protein kinase, KEY1, which regulates the size and dissolution of the pyrenoid condensate. KEY1 phosphorylates the Rubisco linker protein EPYC1, inhibiting EPYC1's interaction with Rubisco and promoting the dissolution of the pyrenoid condensate. KEY1 localizes to the condensate throughout the cell cycle via a Rubisco-binding motif, which is required for KEY1's localization and function.

Our results suggest mechanisms by which KEY1 regulates condensate size, number and dissolution. Our data are consistent with KEY1 being a central player in an active condensate size control system where the condensate size setpoint is inversely related to the ratio of KEY1 activity to phosphatase activity (Fig. 6h). In wild-type cells during cell growth, low KEY1 activity establishes a 'large' size setpoint, favouring a single large condensate and suppressing ectopic condensates. During cell division, an increase in KEY1 activity transiently decreases the condensate size setpoint, favouring a progression to multiple small condensates and complete dissolution. After cell division, a return to lower KEY1 activity again favours a single large condensate. In the absence of KEY1, cells lack an effective pyrenoid condensate size regulation mechanism, resulting in the appearance and persistence of ectopic condensates, aberrant condensate size and the failure to dissolve the pyrenoid condensate during cell division.

While kinases are known to promote the dissolution of various biomolecular condensates[87–91], the molecular mechanisms connecting phosphorylation to dissolution remain poorly understood. Our identification of KEY1, characterization of its physical interactions with Rubisco and the sites it phosphorylates on EPYC1, and analysis of how EPYC1 phosphorylation disrupts phase separation contribute a system where the molecular basis of kinase-driven condensate dissolution is known.

Our work also advances the broader understanding of the regulation of condensate size, number and position. The examples previously proposed to potentially represent active regulation of condensate size[41] (MBK-2 in P granules[87], DYRK3 in stress granules[88], the CLK kinase in nuclear speckles[89], Cdk2 in Cajal bodies[90] and CaMKII in synapsin condensates[91]) could all be cases where the kinase acts at one time to disassemble the condensate rather than participating in active size regulation. Here we show that the pyrenoid condensate system exhibits all the previously established hallmarks of genuine active size regulation[41]: (1) the pyrenoid condensate linker protein, EPYC1, is phosphorylated by a kinase; (2) kinase activity dissolves the condensate; and (3) the kinase is localized to the condensate throughout the cell cycle. The pyrenoid condensate exhibits two additional properties that we propose are also hallmarks of active size control: (4) there is a basal level of EPYC1 phosphorylation during steady-state growth; and (5) increases in kinase activity lead to a decrease in condensate size concomitant with an increase in condensate number. Unexpectedly, the absence of regulation by the kinase led to the appearance of ectopic mislocalized condensates, suggesting that the same kinase-based condensate size regulation is leveraged to suppress ectopic condensates during

cell growth. Finally, our modelling and experimental observations suggest that the same kinase and phosphatase activity can contribute to the centring of condensates within cellular compartments and inter-condensate repulsion. Together, our results advance the understanding of the regulation of condensate size, number and position and establish the pyrenoid condensate as a promising experimental model for further studies of these behaviours, including the possibility of their eventual in vitro reconstitution.

## Online content

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

[1]Department of Molecular Biology, Princeton University, Princeton, NJ, USA. [2]Howard Hughes Medical Institute, Princeton University, Princeton, NJ, USA. [3]Omenn-Darling Bioengineering Institute, Princeton University, Princeton, NJ, USA. [4]Princeton Center for Theoretical Science, Princeton University, Princeton, NJ, USA. [5]Department of Physics, Princeton University, Princeton, NJ, USA. [6]Department of Chemical and Biological Engineering, Princeton University, Princeton, NJ, USA. [7]Lewis-Sigler Institute for Integrative Genomics, Princeton University, Princeton, NJ, USA. [8]Present address: Department of Botany, University of Wisconsin-Madison, Madison, WI, USA. [9]Present address: Institute for Plant-Human Interface, Northeastern University, Boston, MA, USA. [10]Present address: Laboratory of Chemical Physics, National Institute of Diabetes and Digestive and Kidney Diseases, National Institutes of Health, Bethesda, MD, USA. [11]These authors contributed equally: Shan He, Linnea M. Lemma. ✉e-mail: wingreen@princeton.edu; mjonikas@princeton.edu

## Methods

### Strains and culture conditions

The *Chlamydomonas reinhardtii* strain cMJ030 (CC-4533) was the wild type (WT) for all experiments (hereafter WT). The *key1-1* and *key1-2* mutant strains were obtained from the CLiP mutant collection[71] and can be found at the *Chlamydomonas* Resource Center (ID LMJ. RY0402.107748 and LMJ.RY0402.168949), which is funded by the US National Science Foundation. All strains were maintained at 19 °C in the dark or low light (~10 μmol photons m$^{-2}$ s$^{-1}$) on 1.5% agar plates containing Tris-acetate-phosphate (TAP) medium (pH 7.4) with revised trace elements[93].

For unsynchronized liquid cultures, the TAP medium was primed with a loopful of cells and was grown to ~4 × 10$^6$ cells per ml at 22 °C, shaking at 200 rpm under ~200 μmol photons m$^{-2}$ s$^{-1}$ white light in the air in an orbital incubator-shaker (Infors).

For synchronized liquid cultures, cells were first cultivated in the unsynchronized liquid cultures, as described above. Then, cells were inoculated with ~2 × 10$^4$ cells per ml in bottles with Tris-phosphate (TP) or TAP medium, aerated with air and mixed using a conventional magnetic stirrer at 200 rpm in growth chambers with a diurnal cycle for 5–7 days for synchronization before sample collections. The diurnal cycle was set to a 12-h light–dark cycle with a temperature at 28 °C under ~200 μmol photons m$^{-2}$ s$^{-1}$ and a 12-h light–dark cycle with a temperature at 18 °C in the dark[79].

### Co-immunoprecipitation and mass spectrometry to verify kinase KEY1 and EPYC1 interaction

Immunoprecipitation and mass spectrometry of Venus–3×Flag–KEY1, Venus–3×Flag–EPYC1 or Venus–3×Flag was performed as described previously[94] with the following modification: a 40-cm long chromatography column was used. The column temperature was set at 45 °C and a 2-h gradient method with 300 nl min$^{-1}$ flow was used. The mass spectrometer was operated in a data-dependent mode with a 120,000 resolution MS1 scan (positive mode, profile data type, AGC gain of 4e5, maximum injection time of 54 s and mass range of 375–1,500 m/z) in the Orbitrap followed by HCD fragmentation in the Orbitrap with a 30,000 resolution MS2 scan and 30% collision energy.

### Mutant genotype analysis

The cassette insertion site of the *key1-1* mutant strain (LMJ.RY0402.107748) was validated by PCR amplifications performed with the Phusion High-Fidelity DNA Polymerase (New England BioLabs) and primer pairs 3676_F and CIB_5′_R, and 3673_F and CIB_5′_R for testing the 5′ end of the insertion, and primer pairs 3785_R and CIB_3′_F, and 3809_R and CIB_3′_F for testing the 3′ end of the insertion. The specific insertion site was detected and validated by Sanger sequencing (GENEWIZ) and whole-genome sequencing as described by Kafri et al.[95].

The cassette insertion site of the *key1-2* mutant strain (LMJ. RY0402.168949) was validated by PCR amplification performed with the Phusion High-Fidelity DNA Polymerase (New England BioLabs) and primer pairs 949_CLiP_R and CIB_5′_R, and CIB_3′_F and 949_CLiP_F, and 4859-5018_F and 4859-5018_R. The specific insertion site was detected and validated by Sanger sequencing (GENEWIZ).

The presence of the insertion cassette in the rescued strains of *key1-1* was validated by PCR with the primers 3673_F and CIB_5′_R and the Phusion High-Fidelity DNA Polymerase (New England BioLabs).

### Transmission electron microscopy

The transmission electron microscopy experiment was performed as described by Hennacy et al.[75]. Specifically, the samples for electron microscopy were prepared at room temperature and were nutated in 1-ml volumes during chemical treatments and washes unless otherwise noted. After the initial centrifugation for collection, all pelleting was carried out at 3,000*g* for 1 min. Approximately 50 × 10$^6$ cells were collected at 1,000*g* for 5 min and fixed in 2.5% glutaraldehyde in TP

medium (pH 7.4) for 1 h. After three 5-min washes in MilliQ water, the samples were treated with a freshly prepared solution of 1% OsO$_4$, 1.5% K$_3$Fe(CN)$_6$ and 2 mM CaCl$_2$. After four 5-min washes in MilliQ water, the samples were serially dehydrated (5-min incubations in 50%, 75%, 95% and 100% ethanol, followed by two 10-min incubations in 100% acetonitrile). The samples were then suspended in 50% acetonitrile, 17.5% Quetol 651, 22.5% nonenyl succinic anhydride and 10% methyl-5-norbornene-2,3-dicarboxylic anhydride and were left uncapped and stationary overnight in a fume hood to allow for the evaporation of the acetonitrile. The samples were then embedded in epoxy resin containing 34% Quetol 651, 44% nonenyl succinic anhydride, 20% methyl-5-norbornene-2,3-dicarboxylic anhydride and 2% catalyst dimethylbenzylamine. The resin mixture was refreshed daily for four subsequent days. After the final resin refresh, the pellets were resuspended in 300 μl of the resin mixture and centrifuged at 30 °C for 20 min at 10,970*g* (10,500 rpm) in an Eppendorf S-24-11-AT swinging bucket rotor for microfuge tubes. They were then cured at 65 °C for 48 h. Subsequently, ultramicrotomy was performed using a DiaTome diamond knife on a Leica UCT Ultramicrotome at the Imaging and Analysis Center, Princeton University and imaging was performed on a CM100 transmission electron microscope (Philips) at 80 kV or CM200 at 200 kV.

### Mating

The *key1-1*;RBCS1–Venus strain was generated by mating the strains *key1-1* (mt$^-$) and RBCS1–Venus (mt$^+$), which was generated by mating the strain RBCS1–Venus (mt$^-$) in a cMJ030 background with the WT strain CC-1690 (mt$^+$). The mating protocol was adapted from Jiang and Stern[96], except that zygotes were grown to colonies instead of dissecting tetrads, and then streaked to single colonies. These single colonies were first screened by a Typhoon scanner (GE Healthcare) for RBCS1–Venus fluorescent and then by PCR for the correct genotype of *key1-1* with the primers 3676_F and 3785_R.

### Plasmid construction and cloning

The open reading frame of *KEY1* was cloned by PCR with the primers KEY1_473-491_adaptor_F and KEY1_5967-5984_adaptor_R and the KOD Xtreme Hot Start DNA Polymerase (TOYOBO) with the genomic DNA of WT strain (cMJ030) as a template. The plasmid pRAM118-KEY1 was generated using the In-fusion Snap Assembly Master Mix (Takara) with endonuclease HpaI-linearized pRAM118 backbone and the open reading frame of *KEY1*. The sequence of the open reading frame of *KEY1* amplified was verified by Sanger sequencing (GENEWIZ) in the pRAM118-KEY1 plasmid. This generated pRAM118-KEY1 plasmid has a sequence encoding a Venus protein followed by a 3×Flag tag on the backbone, which follows the *KEY1* gene with a short linker fragment in between.

The native promoter of *KEY1* was cloned by PCR with the primers KEY1_US-2141_BstBI_F and KEY1_584_AgeI_R and the Phusion High-Fidelity DNA Polymerase (New England BioLabs) with the genomic DNA of WT strain (cMJ030) as a template. The plasmid pRAM118-pro+KEY1 was generated using the In-fusion Snap Assembly Master Mix (Takara) with endonucleases BstBI and AgeI linearized pRAM118-KEY1 backbone and the native promoter of KEY1 amplified. The sequence of the native promoter of KEY1 amplified was verified by Sanger sequencing (GENEWIZ) in the pRAM118-pro+KEY1 plasmid. This generated pRAM118-pro+KEY1 plasmid has a sequence encoding a Venus protein followed by a 3×Flag tag on the backbone, which follows the *KEY1* gene with a short linker fragment in between.

The plasmid pRAM118-pro+KEY1-SNAP was generated by GenScript Biotech by replacing the Venus-tag encoding sequence in the plasmid pRAM118-pro+KEY1 with codon-optimized SNAP-tag encoding sequence[73]. This generated pRAM118-pro+KEY1-SNAP plasmid has a sequence encoding a 3×Flag on the backbone, which follows the SNAP-tag encoding gene with a short linker fragment in between. The SNAP-tag-encoding gene was codon-optimized for *Chlamydomonas*.

The plasmid pRAM118-pro+KEY1_ΔRBM-SNAP was generated by GenScript Biotech by replacing the candidate Rubisco-binding motif WRVDI encoding sequence (TGGCGGGTAGACATC) to AAVDD encoding sequence (GCGGCGGTAGACGAC). This generated pRAM118-pro+KEY1_ΔRBM-SNAP plasmid has a sequence encoding a 3×Flag on the backbone, which follows the SNAP-tag encoding gene with a short linker fragment in between. The SNAP-tag-encoding gene was codon-optimized for *Chlamydomonas*.

The plasmid pRAM118–proEPYC1–EPYC1–Venus was generated by GenScript Biotech by replacing the PsaD promoter in the plasmid pRAM118 with the synthesized EPYC1 native promoter (2,000 bp upstream of ATG). This generated pRAM118–proEPYC1–EPYC1–Venus plasmid has a sequence encoding a Venus protein followed by a 3×Flag tag on the backbone, which follows the *EPYC1* gene with a short linker fragment in between.

The plasmid pRAM118–proEPYC1–EPYC1–Astring–Venus was generated by GenScript Biotech by changing all the Serine- or Threonine-encoding sequences to Alanine-encoded sequences (after the predicted transit peptide encoding sequences). In detail, it is changing 'GGCAGCTGGCGCGAGTCTTCCACTGCCACCGTGCA GGCCAGGTGAGCACACTTCTGCAGCTATGAGATGCATCTGGGT CCAGCTTAAAGCGGCTCGCGTTGTGTGGCGCGCCGCGATCCC TTATCCGCTCGCCTGCCAGCCGGGCCTTTTCGCACTTGTTTCCTA AGTCAAGTTCGAACCTGCAGCTGGCTGTGCATATCTTGCTAAGTG ATAGCGCGGTTGTACGCGGTTTGAGTACGCTGCTCAACTGGTGTAC TGACACGTTTGCTTGCCGTTTCCCCTGGTGCCCCTTCGCCCCTGCAG CCGCGCCTCGTCGGCCACCAACCGCGTGAGCCCCACCCGCTCCGT CCTGCCCGCCAACTGGCGCCAGGAGCTGGAGAGCCTGCGCAAC GGCAACGGCTCCTCCTCGGCTGCCTCGTCGGCCCCCGCCCCGGC CCGCTCCTCGTCGGCCAGCTGGCGCGACGCCGCCCCGGCCTCGTC GGCCCCTGCCCGCTCCAGCTCTGCCTCCAAGAAGGCCGTGACC CCGTCGCGCAGCGCCCTGCCCTCCAACTGGAAGCAGGAGCTGG AGAGCCTGCGCAGCAGCTCCCCCGCCCCCGCCTCGTCGGCCCCCG CCCCCGGCCCGCTCCTCGTCGGCCAGCTGGCGTGATGCCGCCCC GGCCTCGTCGGCCCCCGCCCCGCTCCAGCTCCTCCAAGAAGGCTG TGACCCCGTCGCGCAGCGCCCTGCCCTCCAACTGGAAGCAGGAGCT GGAGAGCCTGCGCAGCAGCTCCCCCGCCCCCGCCTCGTCGGCC CCTGCCCCGGCCCGCTCCTCGTCGGCCAGCTGGCGTGACGCCG CCCCGGCCTCGTCGGCCCCTGCCCGCTCCAGCTCTGCCTCCAAGA AGGCCGTGACCCCGTCGCGCAGCGCCCTGCCCTCCAACTGGAAGCA GGAGCTGGAGAGCCTGCGCAGCAACTCCCCTGCCCCCGCCTCGTC GGCCCCTGCCCCGGCCCGCTCCTCGTCGGCCAGCTGGCGTGACGC CCCCGCCTCGAGCTCCAGCTCGAGCGCCGACAAGGCCGGCACCAAC CCCTGGACTGGCAAGTCCAAGCCCGAGATCAAGCGCACCGCCC TGCCC' to 'GGCGCCTGGCGCGAGGCCGCGGCCGCCGCCGTGCA GGCCGCGTGAGCACACTTCTGCAGCTATGAGATGCATCTGGGT CCAGCTTAAAGCGGCTCGCGTTGTGTGGCGCGCCGCGATCCCTTAT CCGCTCGCCTGCCAGCCGGGCCTTTTCGCACTTGTTTCCTAAGTCA AGTTCGAACCTGCAGCTGGCTGTGCATATCTTGCTAAGTGATAGCGC GGTTGTACGCGGTTTGAGTACGCTGCTCAACTGGTGTAC TGACACGTTTGCTTGCCGTTTCCCCTGGTGCCCCTTCGCCCCTGCAG CCGCGCCGCGGCGGCCGCCAACGCGTGGCCCCCGCCCGCGCCGT CCTGCCCGCCAACTGGCGCCAGGAGCTGGAGGCCCTGCGCAAC GGCAACGGCGCCGCCGCGGCTGCCGCGGCGGCCCCCGCCCCGGC CCGCGCCGCGGCGGCCGCCTGGCGCGACGCCGCCCCGGCCGCGGC GGCCCCTGCCCGCGCCGCCGCGCCGCCAAGAAGGCCGTGGCCCC GGCGCGCGCCGCCCTGCCCGCCAACTGGAAGCAGGAGCTGGAGG CCCTGCGCGCCGCCGCCCCCGCCCCCGCCGCGGCTGCCCCCGC CCCGGCCCGCGCCGCGGCGGCCGCCTGGCGCGACGCCGCCCCGG CCGCGGCGGCCCCCGCCCCGCCGCCGCCGCCGCCAAGAAGGCTGT GGCCCCGGCGCGCGCCGCCCTGCCCGCCAACTGGAAGCAGG

AGCTGGAGGCCCTGCGCGCCAACGCCCCTGCCCCCGCCGCGGCGGCC CCTGCCCCGGCCCGCGCCGCGGCGGCCGCCGCCTGGCGTGACGCCCCC GCCGCGGCGGCCGCCGCGGCCGCCGACAAGGCCGGCGCCAACCCC TGGGCCGGCAAGGCCAAGCCCGAGATCAAGCGCGCCGCCCTGCCC'. This generated pRAM118–proEPYC1–EPYC1–Astring–Venus plasmid has a sequence encoding a Venus protein followed by a 3×Flag tag on the backbone, which follows the mutated *EPYC1* gene with a short linker fragment in between.

The plasmids generated for this study have been submitted to the *Chlamydomonas* Resource Center (www.chlamycollection.org). The antibiotic resistances and other information of these plasmids are available in Supplementary Table 2.

All cloning of *KEY1* described in this study was based on the sequence of Cre01.g008550 in the *C. reinhardtii* v.5.6 genome, before the release of the *C. reinhardtii* CC-4532 v.6.1 genome. Specifically, there is a shift in the annotation of the start codon of the *KEY1* gene, which introduces an additional 58 amino acids at the N terminus of the KEY1 protein in the v.6.1 genome compared with the v.5.6 genome. Of note, in all our *KEY1* plasmid constructs used for in vivo experiments, expression was driven by the default native *KEY1* promoter, defined as the 2,000 bp genomic region upstream of the start codon in the v.5.6 genome. Therefore, we expect that the translations were initiated correctly in our in vivo experiments. For the in vitro experiments, the KEY1 and KEY1ΔRBM proteins we expressed in *E. coli* both lacked the 58 N-terminal amino acids annotated in the v.6.1 genome. However, we expect that some or all these amino acids encode the chloroplast transit peptide, and thus we do not expect a major impact on the protein's function in vitro. Thus, the reannotation does not affect the expression, localization or function of any of the KEY1 proteins analysed in this study.

### Transformation of *Chlamydomonas reinhardtii*

The strains *key1-1;EPYC1-Venus*, *key1-1;KEY1-Venus* and *key1-1;KEY1-SNAP* were generated by transforming the plasmids pLM005-EPYC1 (together with a DNA fragment carrying a hygromycin-resistance gene), pRAM118-pro+KEY1, and pRAM118-pro+KEY1–SNAP into *key1-1* strain, respectively. The strains *key1-1;RBCS1-Venus;KEY1-SNAP* and *key1-1;RBC S1-Venus;KEY1ΔRBM-SNAP* were generated by transforming the plasmids pRAM118-pro+KEY1–SNAP and pRAM118-pro+KEY1_ΔRBM–SNAP into the *key1-1;RBCS1-Venus* strain, respectively. The *epyc1;EPYC1-Venus* and *epyc1;EPYC1phosphonull-Venus* strains were generated by transforming the plasmid pRAM118–proEPYC1–EPYC1–Venus and pRAM118–proEPYC1–EPYC1–Astring–Venus into the *epyc1* mutant strain, respectively. *Chlamydomonas* transformations were performed as described by Wang et al.[94]. pRAM118-pro+KEY1–SNAP was linearized by ScaI, and the other plasmids were linearized by EcoRV. The transformants were plated on the TAP agar medium with 20 µg ml⁻¹ hygromycin. The single colonies of the transformants were screened by PCR.

The *Chlamydomonas* strains generated for this study have been submitted to the *Chlamydomonas* Resource Center (www.chlamycollection.org). The accession numbers and the antibiotic resistances of these strains are available in Supplementary Table 3.

### RNA extraction and RT–qPCR

Cells of WT, *key1-1*, *key1-2*, *key1-1;-KEY1–Venus*, and *key1-1;KEY1–SNAP* were grown and synchronized in diurnal cycles as described in previous section. Cell samples were collected at +11 h time point in the light cycle. Total RNA was extracted using TRIzol reagent (Invitrogen) following the manufacturer's protocol. For RT–qPCR, 100 ng of total RNA from each sample was used with the Luna Universal One-Step RT–qPCR Kit (New England Biolabs) according to the manufacturer's instructions, and reactions were run on a QuantStudio 6 Real-Time PCR System (Applied Biosystems). G protein β subunit-like polypeptide (*CBLP*) was used as the reference gene for all RT–qPCR analyses. cDNA of *KEY1* was amplified by primer pair Key1_P1Fw and Key1_p1Rv, whereas the

cDNA of *CBLP* was amplified by primer pair CBLP_FW and CBLP_RV[97] (Supplementary Table 1). The relative expression level of each gene was calculated with the cycle threshold (CT) $2^{-\Delta\Delta CT}$ method[98].

### Confocal microscopy for living cells

The WT strains expressing fluorescent proteins (RBCS1–Venus, EPYC1–Venus) were generated by Mackinder et al.[55] using the same method as described above. The strains can be obtained through the *Chlamydomonas* Resource Center under accession numbers CC-5357 (RBCS1–Venus) and CC-5359 (EPYC1–Venus). The *key1-1;EPYC1-Venus, key1-1;RBCS1-Venus, key1-1;RBCS1-Venus;KEY1-SNAP, key1-1;RBCS1-Venus;KEY1^{ΔRBM}-SNAP, epyc1;EPYC1-Venus* and *epyc1;EPYC1^{phosphonull}-Venus* strains were generated as described above.

For observing condensate size and number, cells were grown in unsynchronized liquid culture conditions as described above and transferred to TP medium 6 h before imaging.

For observing pyrenoid phase behaviours during cell division and at the start of the day, cells were grown in diurnal growth conditions in TAP medium as described above for 5 days or longer. The cultures were maintained at ~2 × 10^6 cells per ml through periodic dilution. On the day of imaging, 200 µl of culture was removed from the incubator 1–4 h before darkness.

For observing the subcellular localization of KEY1 or KEY1^{ΔRBM}, we used TMR-STAR SNAP-tag dye (New England BioLabs) to label KEY1–SNAP and KEY1^{ΔRBM}–SNAP. Following the protocol from the manufacturer, we dissolved the dye in dimethylsulfoxide for a solution of 0.6 mM SNAP tag and stored it at −20 °C. Then, 1 ml of cell culture was collected and spun down at 600*g* for 5 min in a table-top centrifuge. We diluted the dye 1:200 into TP medium with 1% BSA and resuspended the pellet in 200 µl of the dye solution. We incubated the cell culture of WT, *key1-1;RBCS1-Venus, key1-1;RBCS1-Venus;KEY1-SNAP* and *key1-1;RBCS1-Venus;KEY1^{ΔRBM}-SNAP* with the dye for 30 min with shaking and constant light. Then the cells were spun down and washed three times with TP. On the final spin, the cells were resuspended in TP with 1% BSA and incubated with shaking for 1 h. Then the cells were spun down and resuspended in TP medium for imaging.

For imaging, 200 µl of cell samples was added to an ibidi eight-well plate and allowed to settle for 5 min. The liquid medium was then aspirated to leave a layer of cells on the glass. Next, TP medium with 2% low melting point agarose was added at 35–40 °C to the well to trap the cells on the glass surface. For timelapse imaging, after 5 min when the agar was solid, 200 µl of mineral oil was added to prevent evaporation. Ensuring the mineral oil wetted the entire well was essential for stable, long timelapse imaging.

With the following exceptions, all microscopy images were taken on a VT-iSIM super-resolution spinning disk confocal microscope on an Olympus iX83 body equipped with a Hamamatsu Orca Quest sCMOS camera and run using VisiView control software. Venus fluorescence was excited with a 514 nm laser and collected at 545/50 nm. Chlorophyll autofluorescence was excited with a 642 nm laser and collected at 700/75 nm. Images shown in Fig. 2 and Supplementary Video 2 were taken on a Zeiss 980 Laser Scanning Confocal Microscope with AiryScan. Images shown in Extended Data Fig. 2 and Supplementary Video 3 were taken on a Nikon A1R scanning confocal microscope.

For analysing condensate size and number, cell samples were imaged on a VT-iSIM with a ×60 1.42 NA oil immersion objective (Olympus: UPLXAPO60XO). The z-step was 0.3 µm. For each strain, the scan-slide function was used in VisiView to obtain images for a large number of cells. These images were subsequently stitched together in VisiView and saved as series of tif files. Analysis was performed using custom MATLAB scripts (https://github.com/linnealemma/KEY1_He-Lemma-etal). In brief, the chlorophyll channel was used to identify individual cells and measure their volume using intensity-based thresholding and MATLAB regionprops3 function.

Then, intensity-based thresholding and regionprops3 were applied to the Venus channel for each cell to identify the condensate size and number. These results for each strain were plotted as violin plots. *P* values were calculated using the *t*-test function in MATLAB.

For analysing the condensate behaviour at the start of the day, cells were imaged on a VT-iSIM microscope with ×100 1.3 NA silicone immersion objective (Olympus, UPLSAPO100XS). The z-step was 0.3 µm. For each strain, a field of view with many cells was defined and the stage positions were stored in VisiView. Every hour, a z-stack was obtained with the same laser intensities. To mimic light conditions from the chamber, the brightfield lamp intensity was adjusted to 150 µmol and turned on between hourly acquisitions. Fiji was used to process z-stack acquisition files. To quantify condensate size, the max-z-projection was created for the Venus channel and the condensates were manually outlined using the ImageJ polygon tool and ROI manager. To quantify condensate number, the look-up-tables were set equal for all time points and then condensates were counted manually by scrubbing through the z-stack for each cell.

For observing pyrenoid phase behaviours of WT and *key1-1* mutant cells with EPYC1–Venus during cell division, cell samples were imaged every 90 s on a Zeiss 980 AiryScan confocal microscope in AiryScan mode using a ×100 1.46 oil immersion objective (Zeiss, 420792-9800-000) with pinhole size 97 µm. Venus fluorescence and chlorophyll autofluorescence were excited with 508 nm and 640 nm lasers, and collected at 524 nm and 667 nm with a spectral detector respectively. The z-step was 1 µm. Zeiss Definite Focus system was used and cells were imaged for 4–8 h.

For observing pyrenoid phase behaviours of WT and *key1-1* mutant cells with RBCS1–Venus during cell division, cell samples were imaged at room temperature every 20 min for 2 h and then every 5 min for 6 h on a Nikon A1R scanning confocal microscope in resonant scanning mode using Nikon Elements software, a ×100 1.49 NA objective and pinhole size 49.81 µm. Venus fluorescence was excited with 514 nm laser and collected at 585/65 nm on a GaAsP detector. Chlorophyll autofluorescence was excited with a 640 nm laser and collected with a long pass 650 nm on a PMT HV detector. Images were collected with 4× line-averaging and 0.5-µm z-steps driven by Nikon A1 Piezo z-drive through the cell volume. The Nikon perfect focus system was enabled. Multiple positions were defined to capture both WT and *key1* mutant cell divisions on the same day. For visualization, the Nikon AI denoise algorithm was used. The unprocessed images were used for statistical analysis. Similar acquisition parameters on the same Nikon A1R were also used to observe pyrenoid phase behaviours of KEY1 rescued strain, with WT and *key1-1* mutant controls.

For analysing pyrenoid phase behaviours in all cell types, the four-dimensional tiff stacks were imported into Fiji. Cells were manually cropped and saved in separate folders as a series of tif files. Division times were manually determined as the frame when the cleavage furrow split the chloroplast(s). Analysis was performed on these images using custom MATLAB scripts (https://github.com/linnealemma/KEY1_He-Lemma-etal). In brief, for each parent cell, the chlorophyll channel of images throughout cell division was used to mask the Venus channel which limited analysis to signal within the chloroplast(s). The Venus images were corrected for photobleaching using a simple ratio, which assumes that protein concentration is constant throughout the acquisition. The Venus channel was then thresholded using a value that was constant for a single acquisition. The MATLAB function regionprops3 was used with the raw and thresholded images to identify the condensate(s) in each frame and measure the volume inside and outside the dense phase.

For analysing the subcellular localization of KEY1 or KEY1^{ΔRBM}, cell samples were imaged on a VT-iSIM microscope using excitation and emission settings for Venus and Chlorophyll as described above. Additionally, the TMR-SNAP dye labelling KEY1 was excited with a 561 nm

laser and collected at 595/40 nm. A ×60 1.46 NA oil immersion objective was used for imaging. The z-step was 0.3 µm. Analysis was performed on these images using custom MATLAB scripts (https://github.com/linnealemma/KEY1_He-Lemma-etal). In brief, the same process was used to identify chloroplasts and condensates as for analysing condensate size and number (above). The SNAP channel was then used to calculate the intensity in the whole chloroplast and the partitioning between the condensate and the non-condensate chloroplast for each strain.

### Spot test growth assays

WT, *key1-1*, *key1-2*, and complemented cell lines were grown in TAP under 200 µmol photons $m^{-2} s^{-1}$ white light until ~2 × 10^6 cells per ml, washed once with TP, resuspended in TP to a concentration of 2 × 10^5 cells per ml, then serially diluted 1:10 three times. Then, 10 µl of each dilution was spotted onto four TP plates or TAP plates and incubated in the air (0.04% $CO_2$), 3% $CO_2$ or 0.004% $CO_2$ under 200 µmol photons $m^{-2} s^{-1}$ white light or in the dark. The plates were imaged after 6 days (TP and TAP-dark, air), 8 days (TP, 3% $CO_2$) or 10 days (TP, 0.004% $CO_2$).

### Immunoblotting analysis

Cell lysates for direct immunoblot analysis were prepared as described by Meyer et al.[85], with adaptations. Specifically, 10 ml of unsynchronized TAP-grown cell cultures were pelleted at 1,000*g* at 4 °C for 5 min and resuspended in 300 µl of lysis buffer containing 5 mM HEPES-KOH (pH 7.5), 100 mM dithiothreitol (DTT), 100 mM $Na_2CO_3$, 2% SDS, 12% sucrose, 1 mM NaF, 0.3 mM $Na_3VO_4$, 2× Halt Protease Inhibitor Cocktail, EDTA-free (Thermo Fisher Scientific) and 2× Halt Phosphatase Inhibitor Single-Use Cocktail (Thermo Fisher Scientific). For the diurnal western blot, 10 ml of synchronized TAP-grown cell cultures were pelleted every hour. The pellets were then weighed and lysis buffer was added for a final concentration of 0.05 g per 300 µl.

Then, for both unsynchronized and diurnal samples, the lysates were transferred to 1.5-ml microcentrifuge tubes and heat-denatured in a thermomixer at 37 °C, 750 rpm for 10 min before being centrifuged at 16,000*g* at 4 °C for 5 min. The supernatant was aliquoted, flash-frozen in liquid nitrogen and stored at −80 °C until use.

Cell lysates for subsequent kinase or phosphatase treatments are prepared as described by Mackinder et al.[55], with adaptations. Specifically, cells were collected by centrifugation at 1,000*g* at 4 °C for 5 min, and then resuspended in ice-cold lysis buffer containing 50 mM HEPES (pH 6.8), 50 mM KOAc, 2 mM Mg(OAc)$_2$, 1 mM $CaCl_2$, 200 mM sorbitol, 1 mM NaF, 0.3 mM $Na_3VO_4$, 2× Halt Protease Inhibitor Cocktail, EDTA-free (Thermo Fisher Scientific) and 2× Halt Phosphatase Inhibitor Single-Use Cocktail (Thermo Fisher Scientific) before being lysed by sonication (6 × 30-s bursts of 20 µm amplitude, with 15 s on ice between bursts; Soniprep 150, MSE UK).

For the unsynchronized lysate (Fig. 3a), gel-loading was normalized by total chlorophyll a + b content as described by Meyer et al.[85].

For electrophoresis, 30 µl of lysate sample (with or without kinase or phosphatase treatments) was mixed with 10 µl 4× SDS−PAGE buffer (Bio-Rad) containing 100 mM DTT (Sigma-Aldrich) followed by denaturation by heating at 70 °C for 10 min. Then, a 12.5-µl denatured protein sample was loaded into a well of a 12.5% 17-well SuperSep Phos-tag Precast Gel (FUJIFILM) for electrophoresis at 100 V for 200 min.

After electrophoresis, the Phos-tag precast gels were washed four times in a washing buffer containing 25 mM Tris, 192 mM glycine, 20% (w/v) methanol and 10 mM EDTA for 10 min each time, and then washed three times with the transfer buffer without EDTA (containing 25 mM Tris, 192 mM glycine and 20% (w/v) methanol) for 10 min each time. Following the washing steps, proteins were transferred to Immobilon-P PVDF membranes (Millipore) using a semidry blotting system (Bio-Rad) at 15 V for 46 min. Membranes were blocked with 5% (w/v) non-fat dry milk (LabScientific) in TBST buffer which contained 0.1% (v/v) Tween 20 (Bio-Rad) at room temperature for 1 h. Blocked membranes were then washed with TBST and incubated

with the primary antibody (anti-EPYC1, obtained from YenZym) with a 1:5,000 dilution in TBST containing 2.5% milk at room temperature for 1 h or at 4 °C overnight. Membranes were washed in TBST four times before incubation with the secondary antibody, goat anti-rabbit IgG (H + L) (Invitrogen cat. no. P131466), with a 1:10,000 dilution at room temperature for 1 h. Immunoreactive proteins were detected using enhanced chemiluminescence (WesternBright ECL, Advansta) followed by X-ray film processing (CL-XPosure Film, Thermo Fisher Scientific; SRX-101A, Konica-Minolta) or imaged by an iBright FL1500 Imaging System (Thermo Fisher Scientific).

For Fig. 4g, cell fractionation and immunoblotting was performed as described by Hennacy et al.[75]. In brief, WT *Chlamydomonas* was grown in TAP medium at air levels of $CO_2$ until it reached 2 × 10^6 cells per ml density. Cells were spun down for 5 min at 1,000*g* and the pellet was weighed and resuspended in 2× volumes of lysis buffer (50 mM HEPES, 10 mM KOAc, 2 mM Mg(OAC)$_2$, 1 mM $CaCl_2$, pH 7.0 and protease inhibitors). Cells were sonicated on ice for 5 min, 3-s pulse and 60% amplitude. Lysates were spun for 10 min at 2,000*g* to remove any unlysed cells. Then, 50 µl of supernatant was collected as whole-cell lysate. The remaining lysate was spun for 30 min at 18,000*g*. The supernatant was collected and the pellet was washed in 5× volumes of lysis buffer, re-spun and resuspended. Laemmeli sample buffer was added to samples followed by boiling at 95 ˚°C for 10 min. Each sample was split in two, with half being separated on an SDS−PAGE (Bio-Rad) while the other half was separated on a Phos-tag precast gel. Gel was transferred to a PVDF membrane using a semidry transfer system (Bio-Rad). Primary antibody was added overnight at 4 ˚°C, followed by three 10-min washes in 1× TBS−0.1% Tween. The secondary antibody was added for 1 h at room temperature, followed by three additional washes in TBST. Blots were imaged on an iBright imaging system using enhanced chemiluminescence.

### Protein expression and purification

The KEY1 and KEY1$^{\Delta RBM}$ protein were expressed and purified by ProteoGenix. For the *E. coli* expression system, the cDNA coding for the 6×His-tagged KEY1 (6His−KEY1-EC) was chemically synthesized with optimization for expression in *E .coli*. After the starter growth of the cells at 37 °C, the protein expression was induced with 1 mM IPTG at 16 ˚°C for 16 h. The expressed 6×His-tagged KEY1 was purified on Nickel resin, with equilibration and binding with PBS buffer (pH 7.5) and washes and elution by imidazole shift. The final sample was buffer exchanged with 20 mM Tris and 50 mM NaCl, pH 8. For the expression with the baculovirus/insect cells expression system, the gene coding for the 6×His-tagged KEY1 (6His−KEY1-IC) was chemically synthesized with optimization for expression in insect cells and then was subcloned in ProteoGenix' proprietary expression vector for insect cells. The expression construct obtained was used to transform *E. coli* strain DH10Bac to produce recombinant Bacmids. Purified recombinant Bacmids were then prepared by a standard method, and transfected in *Spodoptera frugiperda* (Sf) cells to generate the P1 virus stock. Sf cells were infected with different quantities of P2 stock, and the best expression level was observed in Sf9 cells with infection during 72 h with 30 µl of virus. Cell lysis was obtained by sonication in PBS buffer (pH 7.5), and the 6×His-tagged KEY1 was purified with an affinity versus His-Tag purification using a standard protocol, which is an equilibration with PBS (pH 7.5) followed by three washes with PBS (pH 7.5) and 0 mM, 30 mM, 50 mM imidazole buffer, and then an elution with PBS, pH 7.5, 200 mM and 400 mM imidazole buffer. Elutions were pooled and the final sample was buffer exchanged versus 20 mM Tris and 50 mM NaCl, pH 8 by dialysis method and concentrated. Both the *E. coli* and insect cell-purified proteins were stored in 50% glycerol and flash-frozen in liquid nitrogen for shipment and long-term storage at −80 °C. The proteins used in experiments were subjected to an additional flash-freeze−thaw cycle after aliquoting.

EPYC1–GFP and EPYC1 (both with His tag) were expressed and purified as described by He et al.[83] and Wunder et al.[58]. In brief, the pHueEPYC1–GFP plasmid was transformed into BL21 DE3 *E. coli* cells. Cells were grown from frozen glycerol stocks in LB with carbenicillin at 37 ˚°C overnight. Then, 1.2 ml of starter culture were added to 250 ml of medium in 1-l flasks and grown at 37 ˚°C until OD 0.6. At OD 0.6, the flasks were placed on ice to cool for 30 min. Then 0.4 mM IPTG was added and the cultures were grown overnight at 18 ˚°C to induce expression and reduce toxicity. Cells were collected by centrifugation at 5,000$g$ for 30 min. The pellet was resuspended in 50 ml of medium, transferred to a 50-ml Falcon tube and centrifuged for 10 min at 3,200$g$. The pellet was then frozen at −80 °C for future use. On the day of purification, the cell pellet was thawed and resuspend in high-salt lysis buffer (20 mM Tris-HCl, 500 mM NaCl, 10 mM Imidazole, 0.3 mg ml$^{-1}$ lysozyme, 3 mM phenylmethylsulfonyl fluoride, 50 units of Benzonase (Sigma-Aldrich) and 2 mM MgCl$_2$) so that the total volume was 25 ml. The cells were lysed on ice by sonication using a tip sonicator (Q125 + CL-18 probe, QSonica) for six cycles of 10 s with 50 s rest at 50% amplitude, or with additional cycles until visually lysed. The lysate was clarified by centrifugation at 100,000$g$ in SW 41 rotor (Beckman Coulter) for 30 min. The supernatant was then filtered through a 0.2-μm filter before loading onto a HisTrap HP column using an AKTA protein purification system (Cytivia). The column was eluted into high imidazole buffer (20 mM Tris-HCl, pH 8.0, 500 mM NaCl and 300 mM Imidazole) using a 0% to 100% gradient over 5 min at a flow rate of 1 ml min$^{-1}$. Fractions that had a peak in absorbance 280 nm were pooled together. These fractions were loaded onto a size-exclusion column (Superdex Increase 200 10/300 GL, Cytivia) equilibrated in storage buffer (20 mM Tris-HCl, 50 mM NaCl, pH 8.0 and 5% glycerol). Fractions that came off the column at the correct molecular weight peak were pooled, and the concentration was measured using Qubit Fluorometric Quantitation (Thermo Fisher). We also measured the 260/280 nm absorbance using a nanodrop to ensure that DNA contamination was minimal. For the proteins used in the phase diagram (Fig. 4c,d), the A260/280 was 0.3. The protein was then aliquoted, flash-frozen in liquid nitrogen and stored at −80 °C for future use.

Rubisco was extracted and purified as described by He et al. and Meyer et al. with adaptations[59,85]. Specifically, cells were collected by centrifugation at 3,990$g$ for 15 min in an Avanti J-26X centrifuge with an 8.1000 rotor (Beckman) at 4 °C. The pellets were washed in pre-chilled TAP medium and then resuspended in a 1:1 (v/w) ratio of cold extraction buffer (10 mM MgCl$_2$, 50 mM Bicine, 10 mM NaHCO$_3$ and 1 mM DTT, pH 8.0) supplemented with Halt Protease Inhibitor Cocktail, EDTA-Free (Thermo Fisher Scientific). Cell slurry was immediately added to liquid nitrogen to form small popcorn pellets, which were stored at −80 °C until needed. Cells were lysed by cryogenic grinding using a Cryomill (Retsch) at a frequency of 25 oscillations per second for 20 min. The ground powder was defrosted on ice for 60–120 min. The soluble proteins were isolated by centrifugation (16,000$g$, 30 min, 4 °C) and 600 ml of the clarified lysate was loaded on top of a thin-wall ultracentrifugation tube (Ultra-Clear, Beckman Coulter) containing 12 ml of a 10–30% sucrose gradient prepared with the extraction buffer. Gradients were made with a gradient maker (BioComp Instruments) and left to equilibrate at 4 °C for at least 1 hour. After ultracentrifugation in a SW 41 Ti rotor (Beckman Coulter) at a speed of 210,000$g$ at r$_{max}$ (35,000 rpm) for 20 h at 4 °C, fractions (750 ml each) were collected with a piston gradient fractionator (BioComp Instruments). Rubisco-containing fractions were applied to an anion exchange column (MONO Q 5/50 GL, Cytivia) and eluted with a linear salt gradient from 30 to 500 mM NaCl in a buffer with 20 mM Tris-HCl, pH 8 and 2.5 mM DTT. Fractions enriched in Rubisco were confirmed by SDS–PAGE, and were pooled before being concentrated and buffer exchange into phase-separation buffer (20 mM Tris-HCl, pH 8.0 and 50 mM NaCl) with centrifugal filters (Amicon 100K, MilliporeSigma).

## Identification of EPYC1 phosphorylation sites regulated by KEY1

EPYC1–Venus–3×Flag and *key1-1*;EPYC1–Venus–3×Flag cells were prepared for immunoprecipitation as described by Wang et al. with the following modifications[94]. Halt protease and Halt phosphatase inhibitors (Thermo Fisher Scientific) were added at 1× to the lysis buffer and elution buffers. No digitonin was added to the wash or elution buffers. An additional high-salt wash was added, with 0.5 M NaCl in wash buffer to remove Rubisco and other protein interactors. After elution, samples were divided into 50-μl aliquots and immediately flash-frozen.

EPYC1–GFP and KEY1 were purified from *E. coli* as described above. Samples were prepared in 20 mM Tris-HCl 50 mM NaCl, pH 8, supplemented with 5 mM MgCl$_2$ and 5 mM MnCl$_2$, which are necessary for kinase and phosphatase activity, respectively. Each sample had a final volume of 50 μl. *E. coli*-purified EPYC1–GFP was added for a final concentration 17.5 μM. For the Lambda phosphatase-treated sample, 1 μl of Lambda Protein Phosphatase (400,000 U μl$^{-1}$, NEB) was added. For KEY1-treated samples, *E. coli*-purified KEY1 was added at 115 nM, 690 nM, 1.4 μM and 3 μM final concentration along with 1 mM ATP. All samples were incubated at room temperature for 30 min. Then, 2 μl of Halt protease and 2 μl of Halt phosphatase inhibitors (Thermo Fisher Scientific) were added and 2 μl of each sample was reserved for Phos-tag gel analysis. The remaining 48 μl was immediately processed for mass spectrometry analysis.

For the *Chlamydomonas*-purified EPYC1, in-gel digestion of protein bands using trypsin was performed as by Shevchenko et al.[99]. For the *E. coli*-purified EPYC1, the liquid samples were subjected to in-solution thiol reduction/alkylation and Trypsin Gold (Promega) digestion overnight according to the manufacturer's instructions. Trypsin-digested samples were dried completely in a SpeedVac and resuspended with 20 μl of 0.1% formic acid, pH 3, in water. Then, 2 μl (~360 ng) was injected per run using an Easy-nLC 1200 UPLC system. Samples were loaded directly onto a 45-cm long 75-μm inner diameter nanocapillary column packed with 1.9 μm C18-AQ resin (Dr. Maisch) mated to metal emitter in-line with an Orbitrap Fusion Lumos (Thermo Scientific). Column temperature was set at 45 °C and 2-h gradient method with 300 nl min$^{-1}$ flow was used. The mass spectrometer was operated in a data-dependent mode with the 120,000 resolution MS1 scan (positive mode, profile data type, AGC gain of 4e5, maximum injection time of 54 s and mass range of 375–1,500 m/z) in the Orbitrap followed by HCD fragmentation in the Orbitrap (30,000 resolution) with 35% collision energy. A dynamic exclusion list was invoked to exclude previously sequenced peptides for 60 s and a maximum cycle time of 3 s was used. Peptides were isolated for fragmentation using a quadrupole (1.2-m/z isolation window).

Raw files were searched using MSAmanda (v.2.0)[100] and Sequest HT algorithms[101] within the Proteome Discoverer v.2.5.0 suite (Thermo Scientific). Then, 10 ppm MS1 and 0.4 Da MS2 mass tolerances were specified. Carbamidomethylation of cysteine was used as fixed modification, oxidation of methionine and phosphorylation of serine, threonine and tyrosine were specified as dynamic modifications. Pyroglutamate conversion from glutamic acid and glutamine were set as dynamic modifications at the peptide N terminus. Acetylation was specified as dynamic modification at the protein N terminus. Trypsin digestion with a maximum of two missed cleavages was allowed. Files were searched against the UP000006906 *Chlamydomonas* database downloaded from UniProt.org.

Scaffold (v.Scaffold 5.1.0, Proteome Software) was used to validate MS/MS-based peptide identifications. Peptide identifications were accepted if they could be established at greater than 95.0% probability by the Scaffold Local false discovery rate algorithm. The data and subsequent analysis are often unable to distinguish the phosphorylation pattern on the polyS sequences in EPYC1: a shift of 80 Da was observed in the fragmentation, indicating the addition of a phosphate group, but which serine it fell on remained ambiguous. In these cases, the Scaffold

analysis software assigns the phosphorylation site to the first serine in the sequence. In some cases, a phosphorylation site was unambiguously identified as the second or third serine in the sequence.

## In vitro phase-separation experiments

EPYC1–GFP and KEY1 were purified from *E. coli* as described above. Rubisco was purified from *Chlamydomonas* as described above. Then, 10 μM EPYC1–GFP was mixed with 0 μM or 3 μM KEY1 and 0.8 mM ATP in protein kinase buffer (20 mM Tris-HCl, pH 8.0, 5 mM MgCl₂, 5 mM MnCl₂ and 1 mM DTT) and incubated at room temperature for 30 min to produce unphosphorylated and phosphorylated EPYC1–GFP, respectively.

We used microscopy to determine the phase diagram. A glass chamber with parafilm spacers between two no. 1.5 coverslips cleaned with Hellmanex soap was constructed[102]. We mapped the phase diagram by mixing stock Rubisco, EPYC1–GFP and phase-separation buffer at 0–2 μM Rubisco and 0–5 μM EPYC1–GFP in 3 μl total volume. Each sample was loaded immediately after mixing, and the lane was sealed using clear nail polish. After 5 min, the sample was assessed for droplets under the microscope. We used a Nikon Ti Eclipse widefield microscope with an EGFP filter cube (excitation 40/35 nm, dichroic mirror 505 nm and emission 535/40 nm) to assess the EPYC1–GFP signal. Phase separation was visually assessed by the presence of droplets in brightfield and GFP channels.

## Fluorescence correlation spectroscopy to measure EPYC1–Rubisco interactions

Phosphorylated and unphosphorylated EPYC1–GFP were prepared as in the in vitro phase-separation experiments. The phosphorylation status was confirmed through Phos-tag gel with Coomassie stain (Fig. 3c). Then, 20 nM EPYC1–GFP was mixed with 2 μM Rubisco, a concentration at which phase separation was not observed. Fluorescent correlation spectroscopy was performed on the mixtures using a custom-built setup with a 488 nm excitation laser. Diffusion coefficients of EPYC1–GFP were extracted from fitting the fluorescence intensity correlation curves using a single-species analytical model. Details of the method and analysis are described by He et al.[83].

## Surface plasmon resonance experiments

All surface preparation experiments were performed at 25 °C using a Biacore 8K+ SPR system (Cytiva Life Sciences). All binding assays were performed using Biacore PBS-P+ Buffer (20 mM phosphate buffer, 2.7 mM KCl, 137 mM NaCl and 0.05% Surfactant P20, pH 6.8) as a running buffer.

Purified Rubisco was immobilized on the experimental flow cell of each of the eight channels on a CM5 sensor chip using a Biacore Amine Coupling kit (Cytiva Life Sciences) according to the manufacturer's instructions. In brief, the chip surface was activated by an injection of 1:1 *N*-hydroxysuccinimide/1-ethyl-3-(3-dimethylaminopropyl)carbodiimide hydrochloride. Rubisco was diluted to ~50 μg ml⁻¹ in 10 mM acetate (pH 4.5; this pH had been previously optimized using the immobilization pH scouting wizard) and was injected over the chip surface. Excess-free amine groups were then capped with an injection of 1 M ethanolamine. The immobilization levels were ~2,200 resonance units on all channels. The reference flow cell of each of the eight channels on this sensor chip was prepared in exactly the same manner as the experimental surfaces, except that no Rubisco was injected.

For the binding assay, the analyte, which is the RBM[KEY1] peptide (LGFWRVDIEDQAAFI) synthesized by Genscript, was dissolved in the running buffer and diluted to 100 μM. The peptide was injected over the reference flow cell and experimental flow cell on each channel at a flow rate of 30 μl min⁻¹ for 2 min, followed by 10 min of the running buffer alone to allow for dissociation. Analysis of the results was performed with Biacore Evaluation Insight software.

The signal of the change in refractive index at the sensor chip surface is recorded real-time in resonance units, which are proportional to the total mass and number of molecules bound to the surface. To measure binding between the peptide + buffer (or buffer alone) and the Rubisco bound to the surface, we subtracted the signal measured for binding between the peptide + buffer (or buffer alone) and the uncoated surface from the signal measured for binding of the peptide + buffer (or buffer alone) and the Rubisco bound to the surface. The single-subtracted binding signals of the peptide + buffer or buffer alone at the end of the injection time (when the binding signal had reached a steady-state phase) were plotted in Fig. 5d.

To account for the binding between the buffer and the Rubisco, we subtracted the single-subtracted buffer signals from the single-subtracted peptide + buffer signals to obtain the double-subtracted peptide binding to Rubisco signals shown in the Extended Data Fig. 7.

## Mathematical model

To model the dynamics of the linker protein EPYC1 in the pyrenoid, we consider a continuum model in which the concentration of unphosphorylated (sticky) and phosphorylated (non-sticky) EPYC1 are denoted by $c_s(\mathbf{r}, t) = c_{max}\phi_s(\mathbf{r}, t)$ and $c_{ns}(\mathbf{r}, t) = c_{max}\phi_{ns}(\mathbf{r}, t)$, respectively, where $\mathbf{r}$ is position, $t$ is time, $c_{max}$ is the maximum possible molar concentration and $\phi$ are the spatially dependent volume fractions.

**EPYC1 free energies and chemical potentials.** We model EPYC1 and solvent interactions with the following Flory–Huggins free-energy density:

$$\frac{f}{k_B T} = \sum_i \left( \phi_i \ln \phi_i + \sum_j \frac{\chi_{i-j}}{2} \phi_i \phi_j + \sum_j \frac{\kappa_{i-j}}{2} \nabla \phi_i \cdot \nabla \phi_j \right) \quad (1)$$

where $i = \{s, ns, sol\}$ denotes sticky EPYC1s, non-sticky EPYC1s and solvent, respectively, $k_B$ is the Boltzmann constant and $T$ the absolute temperature. The first term in equation (1) is the entropy of mixing, which tends to keep the system well mixed. The second term represents the interaction energy between the components of the mixture, where $\chi_{i-j}$ are Flory–Huggins interaction parameters. The third term reflects a surface energy that determine the width of the interface $\sqrt{\kappa_{i-j}}$ between phases. In equation (1) we have assumed that the molecular volumes of EPYC1 and solvent molecules are the same, which is a reasonable approximation considering the crowded cytoplasm of the cell as solvent. In what follows, we also assume that non-sticky EPYC1s behave identically to solvent molecules, and therefore consider a single non-zero interaction parameter $\chi_{s-s} \equiv \chi < 0$, stemming from the attractive interaction between the sticky unphosphorylated EPYC1s.

**EPYC1 dynamics.** To model the dynamics of EPYC1, we consider the following kinetic equations:

$$\partial_t \phi_i = \nabla \cdot \left[ \sum_j L_{i-j} \nabla (\mu_j - \mu_{sol}) \right] - J_{i \to j} + J_{j \to i} \quad (2)$$

where $i, j = \{s, ns\}$. The first term describes diffusive fluxes driven by gradients of chemical potential $\mu_i$, and $J_{i \to j}$ and $J_{j \to i}$ are the switching fluxes between sticky and non-sticky states. The total EPYC1 concentration is constant $c_{tot} = c_{max}(\phi_s + \phi_{ns})$. To derive equation (2) we have assumed that the diffusive fluxes are linearly proportional to the gradients of the local chemical potentials $\mu$, that is, $\mathbf{j}_i = -\sum_j L_{i-j} \nabla \mu_j$, where $L_{i-j}$ is a matrix of mobility coefficients that satisfies the Onsager reciprocal relations, $L_{i-j} = L_{j-i}$. Imposing an incompressibility condition $\sum_i \phi_i = 1$ implies $\sum_i \mathbf{j}_i = \mathbf{0}$ and thus $\sum_j L_{i-j} = 0$, which yields equation (2). For simplicity, we assume that cross-terms are negligible, that is, $L_{i-j} = 0$ for $i \neq j$ so that gradients of sticky enzyme chemical potential do not drive fluxes of non-sticky enzymes and vice versa, and $L_{i-i} = A_{i-i}\phi_i$. We further assume for simplicity that the diffusivity of enzymes is independent of their configuration or density, that is, $A_{s-s} = A_{ns-ns} = A \equiv D/(k_B T)$, where $D$ is a constant diffusion coefficient.

The chemical potentials of sticky and non-sticky EPYC1 and solvent are obtained from the functional derivative of the free energy of the system with respect to each volume fraction. Thus, the chemical potential differences driving the diffusive fluxes read:

$$\frac{\mu_s - \mu_{sol}}{k_B T} = \ln\left(\frac{\phi_s}{1 - \phi_s - \phi_{ns}}\right) + 2\chi\phi_s - \kappa'_s\nabla^2\phi_s - \kappa'_{s-ns}\nabla^2\phi_{ns} \quad (3)$$

$$\frac{\mu_{ns} - \mu_{sol}}{k_B T} = \ln\left(\frac{\phi_{ns}}{1 - \phi_s - \phi_{ns}}\right) - \kappa'_{s-ns}\nabla^2\phi_s - \kappa'_{ns}\nabla^2\phi_{ns} \quad (4)$$

where $\kappa'_s = \kappa_{s-s} - 2\kappa_{s-sol} + \kappa_{sol-sol}$, $\kappa'_{ns} = \kappa_{ns-ns} - 2\kappa_{ns-sol} + \kappa_{sol-sol}$, and $\kappa'_{s-ns} = \kappa_{s-ns} - \kappa_{s-sol} - \kappa_{ns-sol} + \kappa_{sol-sol}$.

**Switching rates.** For simplicity, we assume that the kinase KEY1 is uniformly distributed throughout the pyrenoid and so we take the switching rates between sticky and non-sticky EPYC1 to be spatially uniform, and thus the switching fluxes are proportional to the volume fractions of EPYC1:

$$J_{i\to j} = k_{i\to j}\phi_i \quad (5)$$

where $k_{i\to j}$ are constant switching rates.

**Boundary conditions.** We impose periodic boundary conditions for the simulations in the square geometry, and no-flux boundary conditions in the geometry mimicking the cell's shape, that is, $\boldsymbol{j}_s \cdot \boldsymbol{n} = 0$ and $\boldsymbol{j}_{ns} \cdot \boldsymbol{n} = 0$. Furthermore, in the latter geometry, to avoid wetting of condensates on the boundaries, we impose the following boundary conditions[103,104]:

$$\nabla\phi_s \cdot \boldsymbol{n} = \frac{\sqrt{2}}{2\kappa'_s}\cos(\theta_s)(1 - \phi_s^2) \quad (6)$$

$$\nabla\phi_{ns} \cdot \boldsymbol{n} = \frac{\sqrt{2}}{2\kappa'_{ns}}\cos(\theta_{ns})(1 - \phi_{ns}^2) \quad (7)$$

where $\theta_i$ is the contact angle of the condensate at the boundary. To prevent wetting, we impose a large angle, $\theta_i = 5\pi/6$.

**Non-dimensionalization.** We reduce the number of parameters of our model by non-dimensionalizing it. To this end, we choose $\sqrt{\kappa'_{s-ns}}$ as characteristic length scale, and $k_{ns\to s}^{-1}$ as characteristic time scale. The resulting dimensionless parameters are:

$$\tilde{D} \equiv \frac{D}{\kappa'_{s-ns}k_{ns\to s}} = \frac{\text{EPYC1 diffusion rate}}{\text{EPYC1 switching rate}} \quad (8)$$

$$\tilde{k} \equiv \frac{k_{s\to ns}}{k_{ns\to s}} = \frac{\text{s} \to \text{ns switching rate}}{\text{ns} \to \text{s switching rate}} \quad (9)$$

Upon this choice of scales, the uniform solution of equations (2)–(5) is $\phi_{s,0} = \phi_{tot}/(1 + \tilde{k})$, and $\phi_{ns,0} = \phi_{tot}\tilde{k}/(1 + \tilde{k})$, where $\phi_{tot}$ is the total EPYC1 volume fraction. The other governing dimensionless parameters are $\phi_{tot}$, the EPYC1-EPYC1 attractive interaction strength $\chi$, the ratio between interfacial parameters $\tilde{\kappa}_s = \kappa'_s/\kappa'_{s-ns}$ and $\tilde{\kappa}_{ns} = \kappa'_{ns}/\kappa'_{s-ns}$, and the dimensionless size of the system $\tilde{L} \equiv L/\sqrt{\kappa'_{s-ns}}$.

**Linear stability analysis.** To determine when EPYC1 will form clusters and whether coarsening of EPYC1 clusters will continue indefinitely or be arrested, we perform a linear stability analysis. Specifically, starting from the uniform solution, the dynamical equations obtained above are linearized by introducing small-amplitude perturbations, which are decomposed into normal modes to obtain the relation between the growth rate $\tilde{\omega}$ and wavenumber $\tilde{q}$ of the perturbations, $L(\tilde{\omega}, \tilde{q}) = 0$, where tildes denote dimensionless quantities:

$$\frac{\tilde{\omega}(\tilde{\omega}+1+\tilde{k})}{\tilde{q}^4\tilde{D}} - \tilde{q}^2\tilde{D}\frac{\begin{array}{c}\phi_{ns,o}[\tilde{\kappa}_{ns}(1-\phi_{ns,o})-\phi_{s,o}] + \phi_{s,o}[\tilde{\kappa}_s(1-\phi_{s,o})-\phi_{ns,o}] \\ +2\chi\phi_{s,o}\phi_{ns,o}\tilde{\kappa}_{ns}(\phi_{tot}-1)\end{array}}{\phi_{tot}-1}$$
$$-\frac{\tilde{\omega}[2-\phi_{tot}+2\chi\phi_{s,o}(\phi_{tot}-1)]+1+\tilde{k}+2\chi\phi_{s,o}(\phi_{tot}-1)}{\tilde{q}^2(\phi_{tot}-1)}$$
$$+\phi_{ns,0}[1+\tilde{\kappa}_{ns}(\tilde{\omega}+\tilde{k})] + \phi_{s,o}[\tilde{k}+\tilde{\kappa}_s(\tilde{\omega}+1)]$$
$$-\frac{\tilde{D}[1+2\chi\phi_{s,o}(\phi_{s,o}-1)]}{\phi_{tot}-1} + \tilde{q}^4\tilde{D}\phi_{s,o}\phi_{ns,o}(\tilde{\kappa}_s\tilde{\kappa}_{ns}-1) = 0 \quad (10)$$

By solving equation (10), we can obtain the most unstable perturbation with maximum growth rate $\tilde{\omega}_{max}$, and corresponding wavenumber $\tilde{q}_{max} = 2\pi/\tilde{\lambda}_{max}$, where $\tilde{\lambda}_{max}$ is the most unstable wavelength of perturbations. To infer whether the system exhibits continued or arrested coarsening for a given set of parameter values ($\phi_{tot}, \tilde{\kappa}_s, \tilde{\kappa}_{ns}, \chi, \tilde{D}, \tilde{k}$), we analyse the amplification curve $\tilde{\omega}(\tilde{q})$. Specifically, when small-wavenumber modes are suppressed ($\tilde{\omega} < 0$), coarsening might be arrested in the nonlinear regime. In the limit $\tilde{q} \to 0$, equation (10) yields $\tilde{\omega} \to 0$ and $d\tilde{\omega}/d\tilde{q} \to 0$. Therefore, to determine the parameter values under which coarsening might be arrested, we identify cases where $\tilde{\omega}(\tilde{q})$ is negatively convex, that is, $d^2\tilde{\omega}/d\tilde{q}^2 < 0$ at $\tilde{q} = 0$, while the system still exhibits at least one unstable mode, $\tilde{\omega}_{max} > 0$, which implies phase separation at a finite wavelength.

**Model parameter values.** In all simulations shown in Fig. 6a–f, we use $\tilde{D} = 10$, $\phi_{tot} = 0.1$, $\tilde{\kappa}_s = 5$, $\tilde{\kappa}_{ns} = 1$, and $\chi = 7$. The dimensionless minor and major semiaxes of the ellipse corresponding to the outer boundary are 29.17 and 35, respectively, and the radius of the inner circle is 16.67. These dimensionless parameter values correspond to: $D = 60~\mu m^2 s^{-1}$, $k_{ns\to s} \simeq 294~s^{-1}$, $\kappa'_{s-ns} \simeq 0.02~\mu m^2$, and a major semiaxes of the ellipse of $5~\mu m$. Additionally, $\tilde{k} = 0.14$ in Fig. 6d,f, and $\tilde{k} = 0.01$ in Fig. 6e. In Fig. 6g, we use $\tilde{k} = 10^{-3}$ and $\chi = 5.5$ to model low KEY1 activity and EPYC1 normally sticky (top snapshot), and $\tilde{k} = 0$ and $\chi = 8$ to model no KEY1 activity and EPYC1 overly sticky.

**Simplified droplet model.** To better understand the selection of finite-size clusters when EPYC1 switches between the sticky and non-sticky state, we simplify our model. First, we assume that the system is a two-component system composed of unphosphorylated, sticky EPYC1, with volume fraction $\phi$, and a bath of solvent plus phosphorylated EPYC1, with total volume fraction $1 - \phi$ due to incompressibility. Neglecting terms related to interfacial tension, the conservation equation of sticky EPYC1 reads:

$$\partial_t\phi = D\nabla \cdot \left(\nabla\phi + \frac{\phi}{1-\phi}\nabla\phi + 2\chi\phi\nabla\phi\right) - k_{s\to ns}\phi + k_{ns\to s}(1-\phi) \quad (11)$$

We consider the case where a dense condensate, primarily composed of sticky EPYC1, forms with an equilibrium concentration $\phi_+$ within the condensate, and a dilute bath with an equilibrium concentration $\phi_- < \phi_+$. We linearize equation (11) inside and outside the condensate around these equilibrium concentrations. For simplicity, we assume that the effective diffusivities of sticky EPYC1 are identical both inside and outside the condensate. Furthermore, we assume the cluster is in a quasi-steady state, meaning its growth or shrinkage occurs much more slowly than the time it takes for EPYC1 to diffuse or switch between states. These assumptions lead to the following simplified equation:

$$0 = \nabla^2\phi_i - \phi_i + \tilde{k}(1 - \phi_i) \quad (12)$$

where $i = (\text{in}, \text{out})$, and $\tilde{k} = k_{ns\to s}/k_{s\to ns}$. We have non-dimensionalized equation (12) by choosing $\ell = \sqrt{D/k_{s\to ns}}$ as characteristic length scale,

which stems from a balance between EPYC1 diffusion and the sticky to non-sticky switching rate. We solve equation (12) considering a spherically symmetric cluster of dimensionless radius $\tilde{R} = R/\ell$:

$$\phi_{\text{in}}(r) = \frac{\tilde{k}}{1+\tilde{k}} + \frac{\tilde{R}\left[\tilde{k}(\phi_+ - 1) + \phi_+\right]\text{csch}\left(\sqrt{1+\tilde{k}}\tilde{R}\right)\sinh\left(\sqrt{1+\tilde{k}}\tilde{r}\right)}{(1+\tilde{k})\tilde{r}} \quad (13)$$

$$\phi_{\text{out}}(r) = \frac{\tilde{k}}{1+\tilde{k}} + \frac{\tilde{R}\left[\tilde{k}(\phi_- - 1) + \phi_-\right]\exp\left[\sqrt{1+\tilde{k}}(\tilde{R}-\tilde{r})\right]}{(1+\tilde{k})\tilde{r}} \quad (14)$$

where we have imposed regularity of the solution at $\tilde{r} = 0$, zero EPYC1 flux far from the cluster at $\tilde{r} \to \infty$, and fixed equilibrium concentrations, $\phi_{\text{in}} = \phi_+$ and $\phi_{\text{out}} = \phi_-$, at the cluster interface $\tilde{r} = \tilde{R}$, disregarding any contribution from Laplace pressure. The evolution of the cluster is determined by the following kinematic equation:

$$\frac{d\tilde{R}}{d\tilde{t}} = \boldsymbol{n} \cdot \frac{\tilde{\boldsymbol{j}}_{\text{in}} - \tilde{\boldsymbol{j}}_{\text{out}}}{\phi_{\text{in}} - \phi_{\text{out}}}$$
$$= \frac{1}{\tilde{R}} - \frac{\tilde{k}(\phi_- - 1) + \phi_- + [\tilde{k}(\phi_+ - 1) + \phi_+]\coth(\sqrt{1+\tilde{k}}\tilde{R})}{\sqrt{1+\tilde{k}}(\phi_+ - \phi_-)} \quad (15)$$

where we have chosen $t_c = \ell^2/D$ as characteristic time scale, and $\tilde{\boldsymbol{j}}_i = -\nabla\phi_i$ are dimensionless diffusive fluxes, where $\tilde{\boldsymbol{j}}_{\text{in}}$ is the flux density just inside the surface of the cluster and $\tilde{\boldsymbol{j}}_{\text{out}}$ is the flux density just outside the surface of the cluster. In the absence of activity, that is without switching rates, $\tilde{\boldsymbol{j}}_{\text{in}} = 0$ and $\tilde{\boldsymbol{j}}_{\text{out}} = -4\pi(\phi_\infty - \phi_-)\tilde{R}$, implying that the condensate grows unboundedly in a diffusive manner by adsorbing material. This leads to the following growth law: $\tilde{R} = \tilde{R}_{\text{init}} + \sqrt{2(\phi_\infty - \phi_-)/(\phi_+ - \phi_-)t}$, where $\phi_\infty$ is the far-field volume fraction of sticky EPYC1. When reactions are turned on, there is a critical ratio of switching rates $\tilde{k}_c = f(\phi_+, \phi_-)$, below which equation (15) has a stable fixed point (that is, $d\tilde{R}/d\tilde{t} = 0$), implying an equilibrium radius $\tilde{R}_{\text{eq}}$. Above this critical point, no stable fixed points exist, and the cluster grows unboundedly. Close to the transition region, $\tilde{R} \gg 1$. Thus, taking this limit in equation (15) and solving $d\tilde{R}/d\tilde{t} = 0$ for $\tilde{k}$ yields the critical ratio of switching rates, $\tilde{k}_c \simeq \frac{\phi_+ + \phi_-}{2 - \phi_+ - \phi_-}$, in excellent agreement with numerical solution of equation (15).

Below the critical switching ratio, $\tilde{k} < \tilde{k}_c$, EPYC1 diffusion and reactions result in the internal and external fluxes of sticky EPYC1 across the condensate surface scaling differently with cluster radius: $\tilde{\boldsymbol{j}}_{\text{in}} \sim \frac{4\pi\tilde{R}^3[\tilde{k}(1-\phi_+)-\phi_+]}{3}$ and $\tilde{\boldsymbol{j}}_{\text{out}} \sim \frac{4\pi\tilde{R}^2[\tilde{k}(\phi_- - 1)+\phi_-]}{\sqrt{1+\tilde{k}}}(1 + \frac{1}{\sqrt{1+\tilde{k}}\tilde{R}})$, leading to a stable cluster size, where we have expanded $\tilde{\boldsymbol{j}}_{\text{in}}$ and $\tilde{\boldsymbol{j}}_{\text{out}}$ in powers of $\tilde{R}$. Above the critical switching ratio, $\tilde{k} > \tilde{k}_c$, most EPYC1 are sticky, and the external diffusive flux, scaling as $\sim 4\pi\tilde{R}^2$, dominates, causing the condensate to grow unboundedly. In the former case, well below the critical switching ratio, equating $\tilde{\boldsymbol{j}}_{\text{in}} = \tilde{\boldsymbol{j}}_{\text{out}}$ (which implies $d\tilde{R}/d\tilde{t} = 0$), yields a reasonable approximation for the equilibrium radius $\tilde{R}_{\text{eq}}$. In particular, in the limit $\phi_+ \gg \phi_-$, the equilibrium radius $R_{\text{eq}} \simeq \sqrt{\frac{D}{k_{\text{s}\to\text{ns}}}}\frac{3\tilde{k}+\sqrt{3\tilde{k}(4+3\tilde{k})}}{2\sqrt{1+\tilde{k}}}$.

## Statistics and reproducibility
No statistical method was used to predetermine sample size, but our sample sizes are similar to those reported in previous publications[52,58,59,75,79,83,105]. Dead cells were excluded from analysis of confocal images of fluorescently tagged strains. Images with substantial offset between fluorescent channels were excluded from analysis of confocal images of fluorescently tagged strains. No other data were excluded from the analyses. The experiments were not randomized. The investigators were not blinded to allocation during experiments and outcome assessment. For all $P$ values calculated in the paper, the data distribution was assumed to be normal but this was not formally tested.

## Reporting summary
Further information on research design is available in the Nature Portfolio Reporting Summary linked to this article.

## Data availability
Proteomics data that support the findings of this study have been deposited in the ProteomeXchange Consortium Repository under accession code PXD072834 (refs. 106,107). Previously published proteomics data that were referenced here are available under accession code PXD007664. The underlying tif data for all confocal microscopy images shown in the figures can be found on GitHub at https://github.com/linnealemma/KEY1_He-Lemma-etal. All other data supporting the findings of this study are available from the corresponding author on reasonable request. Source data are provided with this paper.

## Code availability
Code is available on GitHub at https://github.com/linnealemma/KEY1_He-Lemma-etal and https://github.com/amcalv/2025-Simulations-Kinase-KEY1-paper.

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

## Acknowledgements

We thank S. Kyin and H. Shwe for their help with mass spectrometry through the Princeton Molecular Biology Core Facility; M. Wühr and F. Keber for useful discussions on mass spectrometry data; E. Gatzogiannis for microscopy support and discussions; X. Li for technical support of SPR experiments; M. Bao, as part of Life Science Editors, L. Mackinder, A. McCormick, M. Meyer, A. Garde, V. Crans, S.e Skanchy, A. Strom, N. Jaberi-Lashkari, A. Donlic and J. Botello for manuscript editing help; H. Griffiths, members of the Jonikas laboratory, Brangwynne laboratory and Wingreen group for helpful discussions. We are grateful to the *Chlamydomonas* Resource Center for maintaining the strains described here. This project was supported by funding from the Howard Hughes Medical Institute to C.P.B. and M.C.J.; grants from the US National Science Foundation (MCB-1935444, MCB-2410354 and MCB-1914989) and US Department of Energy (DE-SC0020195) to M.C.J. and N.S.W; and US National Institutes of Health (1R01GM140032-01 and T32GM007388) to N.S.W., M.C.J., G.H. and J.H.H. L.M.L. was supported by the Omenn-Darling Bioengineering Institute Innovators (ODBI2) Postdoctoral Fellowship and the HHMI Hanna H. Gray Postdoctoral Fellowship. A.M.C. was supported by a Princeton Center for Theoretical Science (PCTS) fellowship and a Human Frontier Science (HFSP) fellowship (LT000035/2021). A.M.C. also acknowledges support from the Center for the Physics of Biological Function. The content is solely the responsibility of the authors and does not necessarily represent the official views of the funders.

## Author contributions

S.H., L.M.L. and M.C.J. conceived the project. A.M.C. and N.S.W. conceived the mathematical model. L.W. performed immunoprecipitation experiments and mass spectrometry. S.H. performed analysis. S.H. performed genetic analysis of *key1-1* and *key1-2* mutants. A.K.R. performed RT–qPCR. J.H.H. performed the transmission electron microscopy. S.H. performed or designed plasmid construction and cloning, and generated *Chlamydomonas* strains by mating or transformation. L.M.L., C.W., A.K.B. and S.H. performed confocal microscopy and analysis. L.M.L. and S.H. performed spot test experiment. L.M.L. expressed and purified EPYC1–GFP and extracted Rubisco. G.H. expressed and purified 6×His-EPYC1. S.H. and L.M.L supervised the expression and purification project for KEY1 and KEY1$^{\Delta RBM}$. S.H. and S.L.E. performed immunoblotting analyses for EPYC1 phosphorylation status. L.M.L. and S.H. prepared samples and performed analysis to identify the phosphorylation sites on EPYC1 mediated by KEY1. L.M.L., S.H., C.W. and L.B. collected diurnal cell samples for western blot experiments. L.M.L. performed in vitro phase-separation experiments. G.H., Q.W. and L.M.L. performed fluorescence correlation spectroscopy experiments and analysis. S.H. analysed the sequence of KEY1 and predicted its structure with AlphaFold 2. S.H. performed surface plasmon resonance experiment. A.M.C. performed simulations and theoretical analyses. S.H., L.M.L., A.M.C., C.P.B., N.S.W. and M.C.J. guided the research and wrote the manuscript with input from all authors.

## Competing interests

C.P.B. is a scientific founder, Scientific Advisory Board member, shareholder and consultant for Nereid Therapeutics. All other authors declare no competing interests.

## Additional information

**Extended data** is available for this paper at https://doi.org/10.1038/s41556-026-01908-w.

**Correspondence and requests for materials** should be addressed to Ned S. Wingreen or Martin C. Jonikas.

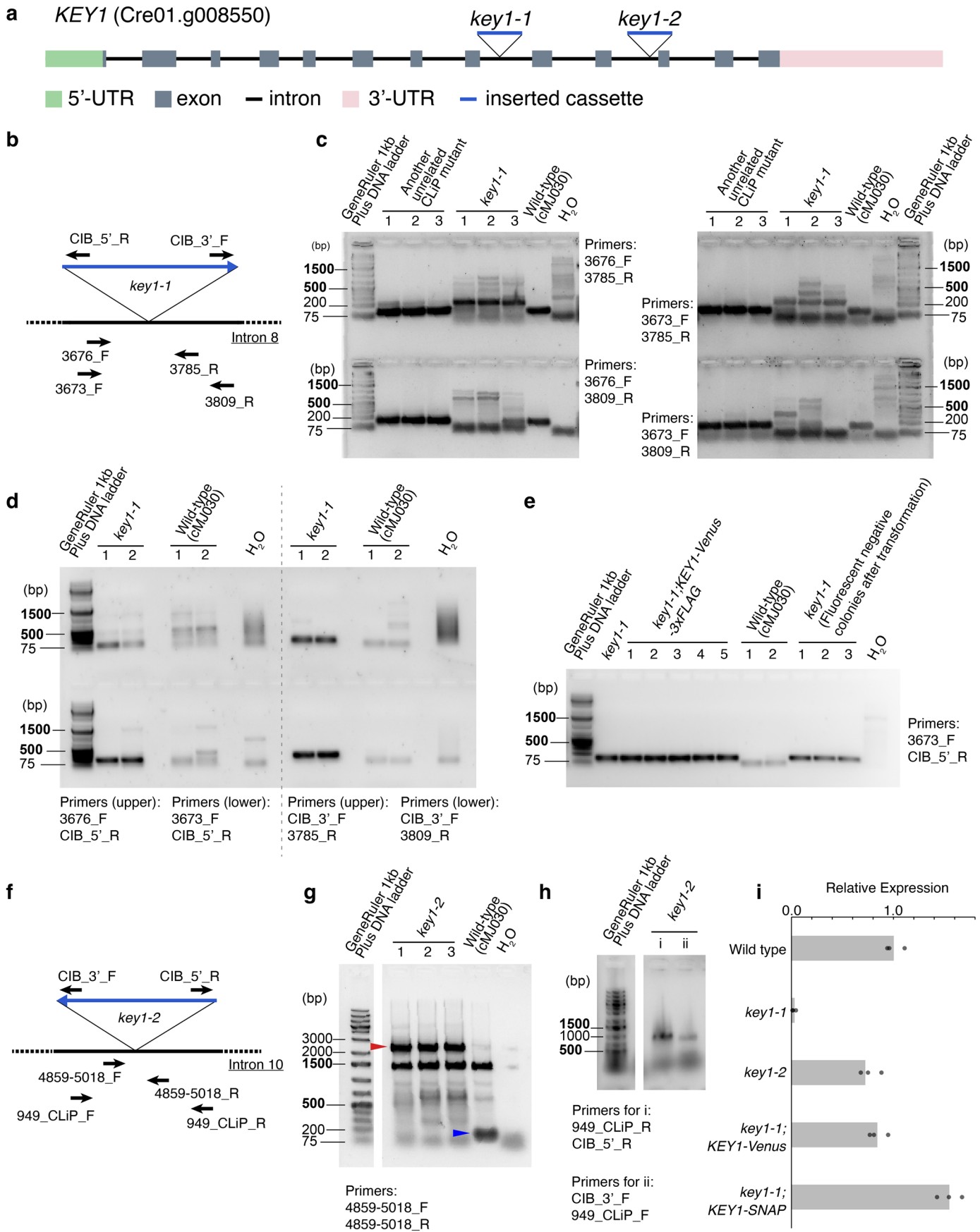

**Extended Data Fig. 1 | See next page for caption.**

**Extended Data Fig. 1 | Genetic characterization and complementation of *key1* mutants, related to Fig. 1. a**, Gene structure of *KEY1* and the cassette insertion sites of *key1-1* and *key1-2* mutants obtained from the CLiP mutant library collection[71] (see also Methods). The insertion cassettes in the CLiP mutants contain transcription terminators, which typically lead to the knockdown of the transcript due to nonsense-mediated decay. **b**, Cartoon of the *key1-1* mutant cassette insertion site, showing the approximate positions of the primers used to characterize *key1-1*. **c**, Agarose gel electrophoresis of the results of PCR amplification across the cassette insertion sites of *key1-1*. In each of the four primer-pair combinations shown, a specific fragment (ranging from 110 bp to 137 bp) was amplified from wild-type genomic DNA, but not from the genomic DNA of any of the three single colonies of *key1-1*, suggesting a DNA insertion at this locus in the *key1-1* genome. **d**, Agarose gel electrophoresis of the results of PCR amplification across the junction regions of the insertion cassette and the KEY1 encoding sequence on both sides of the insertion site in *key1-1*. In each of the four primer-pair combinations shown, a specific fragment (ranging from ~160 bp to ~230 bp) was amplified from *key1-1* genomic DNA in both replicates, but not from wild-type genomic DNA in either replicate, indicating a cassette insertion at this locus in the *key1-1* genome. **e**, Agarose gel electrophoresis of the results of PCR amplification across the junction regions of the insertion cassette and the KEY1 encoding sequence in the complemented strain of

*key1-1* (key1-1;KEY1-Venus-3×Flag). A specific fragment was amplified from the genomic DNA of both *key1-1* and the complemented strains, but not from wild-type genomic DNA in either replicate, indicating a cassette insertion in the complemented strain at the same locus as in the *key1-1* genome. **f**, Cartoon of the *key1-2* mutant cassette insertion site, showing the approximate positions of the primers used to characterize *key1-2*. **g**, Agarose gel electrophoresis of the results of PCR amplification across the cassette insertion sites of *key1-2*. A specific fragment (160 bp, indicated by the blue arrowhead) was amplified from wild-type genomic DNA, but not from the genomic DNA of any of the three single colonies of *key1-2*. However, a specific band of ~2.4 kb (indicated by the red arrowhead) was amplified from the genomic DNA of each of the three single colonies of *key1-2*, indicating a cassette insertion at this locus in the *key1-2* genome. **h**, Agarose gel electrophoresis of the results of PCR amplification across the junction regions of the insertion cassette and the KEY1 encoding sequence on both sides of the insertion site in *key1-2*. In each of the primer-pair combinations shown, a specific fragment (~1 kb) was amplified from *key1-2* genomic DNA, indicating a cassette insertion at this locus in the *key1-2* genome. **i**, RT–qPCR analysis of KEY1 expression in lines *key1-1*, *key1-2*, *key1-1;KEY1-Venus*, *key1-1;KEY1-SNAP*, and wild type. The height of the bar represents the average of n = 3 biological replicates. Each point represents a single biological replicate.

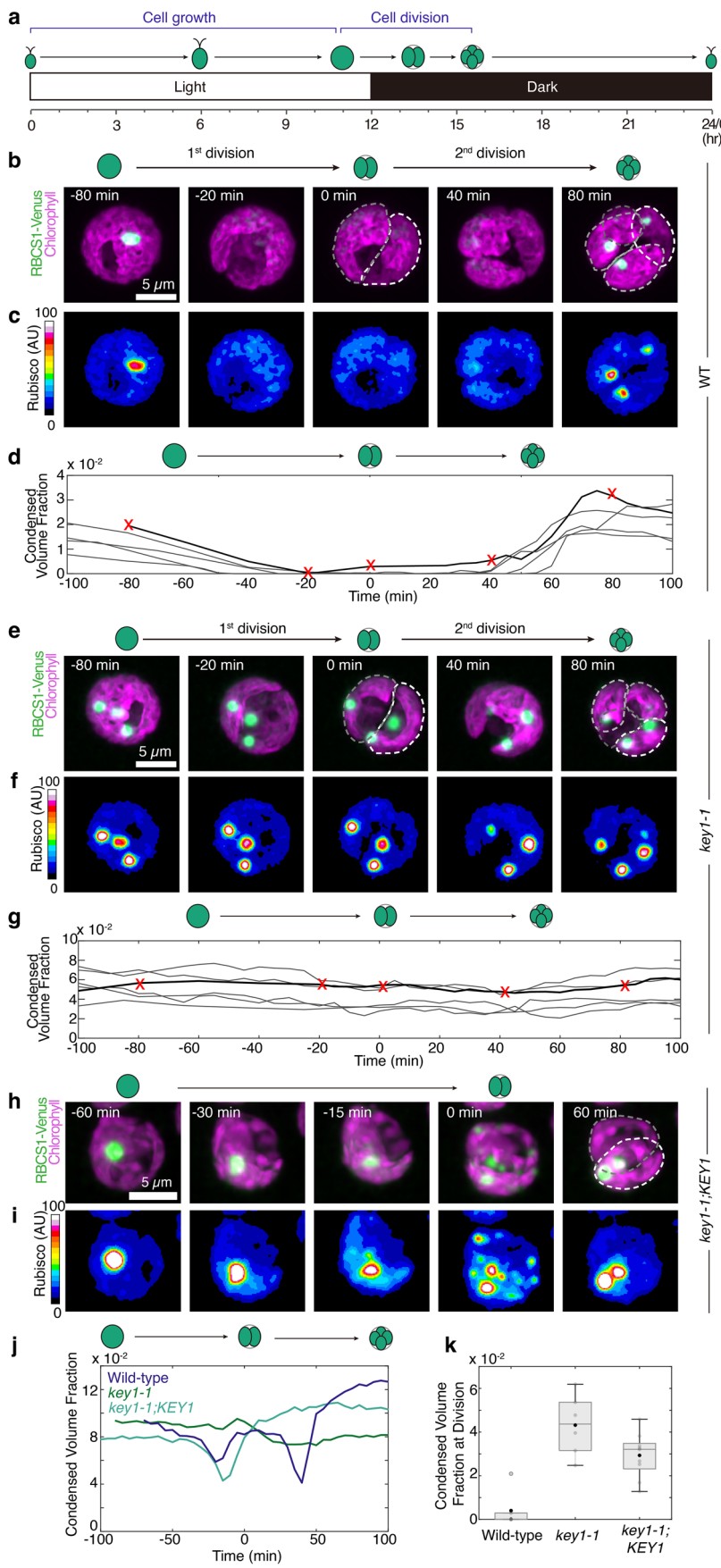

**Extended Data Fig. 2 | See next page for caption.**

**Extended Data Fig. 2 | Rubisco-Venus behaves similarly to EPYC1–Venus in WT and *key1-1*, and the dissolution of the pyrenoid condensate is observed in the *key1-1* rescued strain, related to Fig. 2. a**, Chlamydomonas cells were synchronized in a diurnal cycle where they divided twice in rapid succession upon the shift from light to dark each day. **b-c**, Timelapse microscopy of a dividing wild-type cell where the pyrenoid condensate is labelled by RBCS1–Venus (green, maximum z-projection), and the chloroplast is visualized through chlorophyll autofluorescence (magenta, maximum z-projection) (**b**). The first cell division was completed at 0 min, and the second division was completed at 50 min, ending with four descendant cells. A heatmap allows visualization of RBCS1–Venus dissolution during cell division (**c**). **d**, The condensed volume fraction ($V_{\text{dense phase}}/V_{\text{chloroplast(s)}}$) of RBCS1–Venus in wild-type cells throughout cell division. The grey curves represent the individual mother cell profiles. The black curve shows the cell in (**b, c**) with the time points marked in red. Protein concentration was assumed to be constant across the acquisition. **e-f**, Timelapse microscopy of a dividing *key1-1* mutant cell with the pyrenoid condensate labelled by RBCS1–Venus (green, maximum z-projection) and the chloroplast visualized through chlorophyll autofluorescence (magenta, maximum z-projection) (**e**). The first cell division was completed at 0 min, and the second

division was completed at 55 min, ending with four descendant cells. The heatmap shows RBCS1–Venus concentration (**f**). **g**, Condensed volume fraction of RBCS1–Venus in *key1-1* mutant cells throughout cell division. The grey curves represent the individual mother cell profiles. The black curve shows the cell in (**e-f**) with the time points marked in red. Protein concentration was assumed to be constant across the acquisition. **h-i**, Timelapse microscopy of a dividing *key1-1* cell rescued with KEY1 (*key1-1;RBCS1-Venus;KEY1-SNAP*) with the pyrenoid condensate labelled by RBCS1–Venus (green, maximum z-projection) and chloroplast visualized through chlorophyll autofluorescence (magenta, maximum z-projection) (**h**). The cell divided once at 0 min. The heatmap shows RBCS1–Venus concentration (**i**). **j**, The condensed volume fraction of RBCS1–Venus in wild type, *key1-1* mutant, and the rescued strain (*key1-1;RBCS1-Venus;KEY1-SNAP*) during cell division. Protein concentration was assumed to be constant across the acquisition. **k**, Condensed volume fraction of RBCS1–Venus in wild type (n = 6 cells), *key1-1* (n = 6 cells), and the rescued strain (*key1-1;RBCS1-Venus;KEY1-SNAP*, n = 9 cells) at cell division. The black circle indicates the median, the grey box indicates the first quartile, and the line indicates the third quartile of each distribution. The grey circles indicate individual cells. Protein concentration was assumed to be constant across the acquisition.

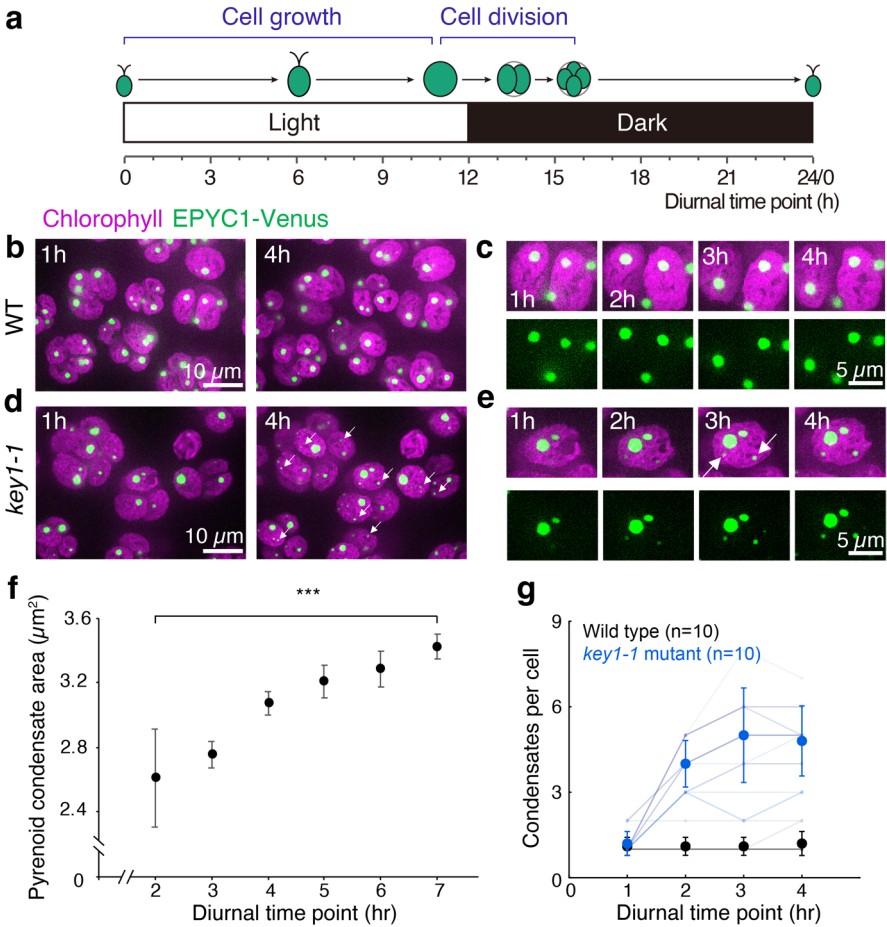

**Extended Data Fig. 3 | The pyrenoid condensate grows over time in the wild type, while ectopic condensates nucleate over time in the *key1-1* mutant during cell growth, related to Fig. 2. a**, Chlamydomonas cells were synchronized in a diurnal cycle, during which they grow in the light cycle, and divide twice in rapid succession upon the shift from light to dark each day. **b**, Timelapse confocal images of wild-type cells during the light cycle. The pyrenoid condensate is labelled by EPYC1–Venus (green, maximum z-projection), and the chloroplast is visualized through chlorophyll autofluorescence (magenta, maximum z-projection). **c**, Zoom-in of a single wild-type cell from **b** showing hourly time points. **d**, Timelapse confocal images of a *key1-1* cell during the light cycle. The pyrenoid condensate is labelled by EPYC1–Venus (green, maximum z-projection),

and the chloroplast is visualized through chlorophyll autofluorescence (magenta, maximum z-projection). The white arrows indicate several of the ectopic pyrenoid condensates that appeared during the timelapse. **e**, Zoom-in of a single *key1-1* cell from **d** showing hourly time points. The white arrows indicate the ectopic pyrenoid condensates that appeared during the timelapse. **f**, Mean pyrenoid condensate area in wild-type cells expressing EPYC1–Venus ($n = 11$ cells). Error bars, SEM. *** indicates a statistically-significant ($p = 1.4 \times 10^{-6}$, two-sided t-test) difference in area from hour 2 to hour 7. **g**, Mean condensate number per cell in wild-type (black) and *key1-1* (blue) cells expressing EPYC1–Venus during cell growth (n = 10 cells). Error bars represent standard deviation. Light grey and blue indicate individual cell traces.

**a**

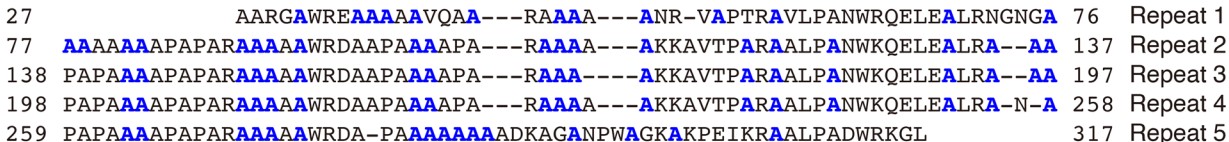

EPYC1<sup>phosphonull</sup>

| | | | |
|---|---|---|---|
| 27 | AARGAWREAAAAAVQAA---RAAAA---ANR-VAPTRAVLPANWRQELEALRNGNGA | 76 | Repeat 1 |
| 77 | AAAAAAAPAPARAAAAAWRDAAPAAAAPA---RAAAA---AKKAVTPARAALPANWKQELEALRA--AA | 137 | Repeat 2 |
| 138 | PAPAAAAPAPARAAAAAWRDAAPAAAAPA---RAAA----AKKAVTPARAALPANWKQELEALRA--AA | 197 | Repeat 3 |
| 198 | PAPAAAAPAPARAAAAAWRDAAPAAAAPA---RAAAA---AKKAVTPARAALPANWKQELEALRA-N-A | 258 | Repeat 4 |
| 259 | PAPAAAAPAPARAAAAAWRDA-PAAAAAAAADKAGANPWAGKAKPEIKRAALPADWRKGL | | 317 | Repeat 5 |

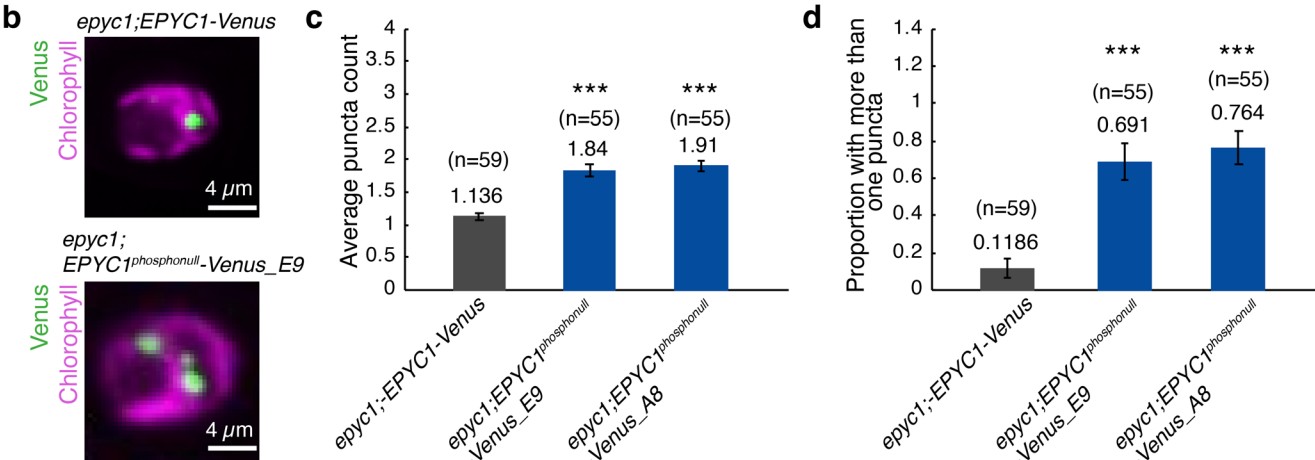

**Extended Data Fig. 4 | Phosphonull mutants of EPYC1 have multiple condensates, related to Fig. 3. a**, Amino acid sequence of the phosphonull mutant of EPYC1. All the Serines and Threonines in the wild-type EPYC1 sequence were substituted for Alanines, which are shown in blue. **b**, Confocal fluorescence images of *epyc1* rescued by EPYC1 (*epyc1;EPYC1-Venus*) (top) and *epyc1* expressing the phosphonull mutant of EPYC1 (*epyc1;EPYC1*<sup>phosphonull</sup>*-Venus*) (bottom). Magenta shows the mid-plane of chlorophyll autofluorescence. Green shows the maximum intensity z-projection of Venus. **c-d**, The average number of condensates (**c**) and the proportion of cells in the sample with greater than one condensate (**d**) were quantified in a representative sample of cells. Sample sizes are n = 59, n = 55, and n = 55 cells for *epyc1;EPYC1–Venus, epyc1;EPYC1*<sup>phosphonull</sup>*-Venus* E9, and *epyc1;EPYC1*<sup>phosphonull</sup>*-Venus* A8 respectively, from an independent experiment for each strain. Additional replicates showed qualitatively similar results. Values and error bars depict mean and SEM, respectively. *** indicates a phenotype is different from the *epyc1;EPYC1-Venus* phenotype with statistical significance $p < 0.001$ ($p = 1.4 \times 10^{-6}$ and $p = 1.8 \times 10^{-9}$ for puncta count, $p = 7.6 \times 10^{-5}$ and $p = 2.0 \times 10^{-5}$ for proportion with more than one puncta for phosphonull strains E9 and A8 respectively), two-sided t-test.

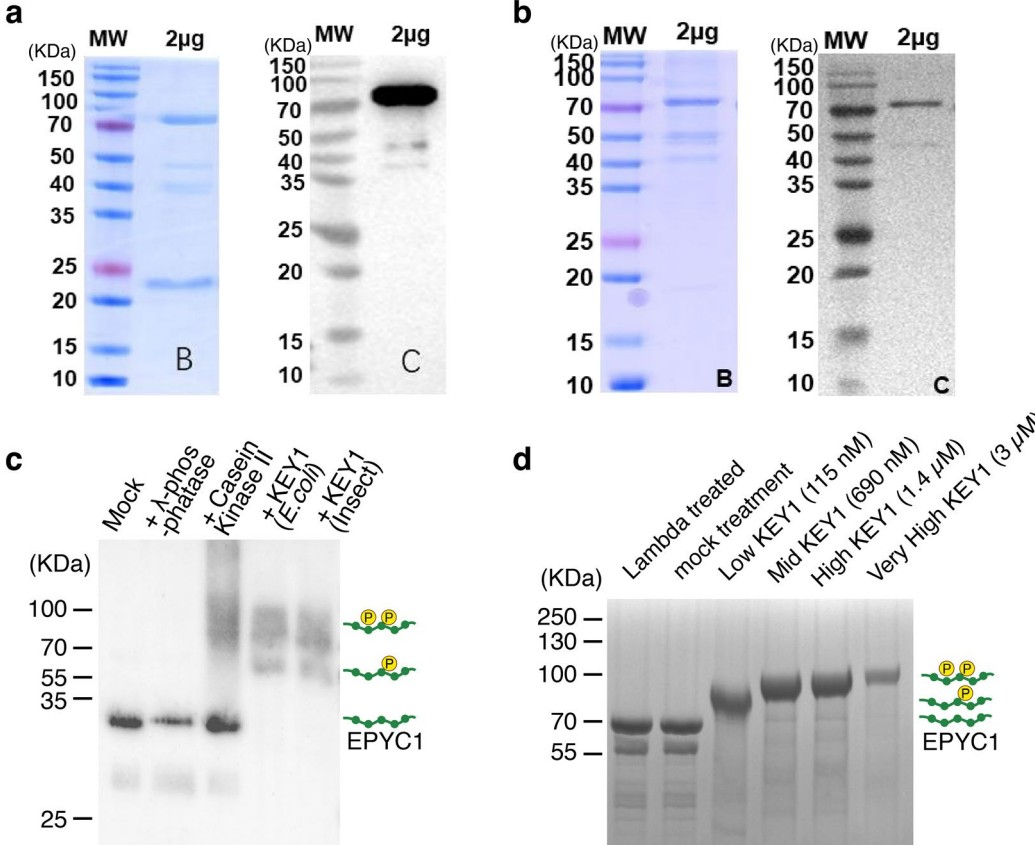

**Extended Data Fig. 5 | Purified KEY1 from *E. coli* and insect cells is active on *E. coli*-expressed EPYC1, related to Fig. 3. a,** Coomassie blue staining (left) and western blot with anti-His-tag antibody (right) for KEY1 protein purified from *E. coli* expression system (data provided by Proteogenix). The experiment was performed 1 time during purification. Purity was assessed 2 times later through Coomassie stain with similar results. **b,** Coomassie blue staining (left) and western blot with anti-His-tag antibody (right) for KEY1 protein purified from baculovirus/insect cell expression system (data provided by Proteogenix).

The experiment was performed 1 time during purification. **c,** Anti-EPYC1 western blot of Phos-tag gel of purified His-tagged EPYC1 expressed in *E. coli* with or without treatment with Lambda phosphatase, Casein Kinase II, *E. coli*-expressed KEY1, or insect cell expressed KEY1. The experiment was performed 1 time and was consistent with results in Fig. 3a,c and with the same experiment on EPYC1–GFP. **d,** Coomassie stain of Phos-tag gels of *E. coli*-expressed EPYC1–GFP samples prepared for phosphoproteomics. The experiment was performed 2 times, once for each mass spectrometry replicate.

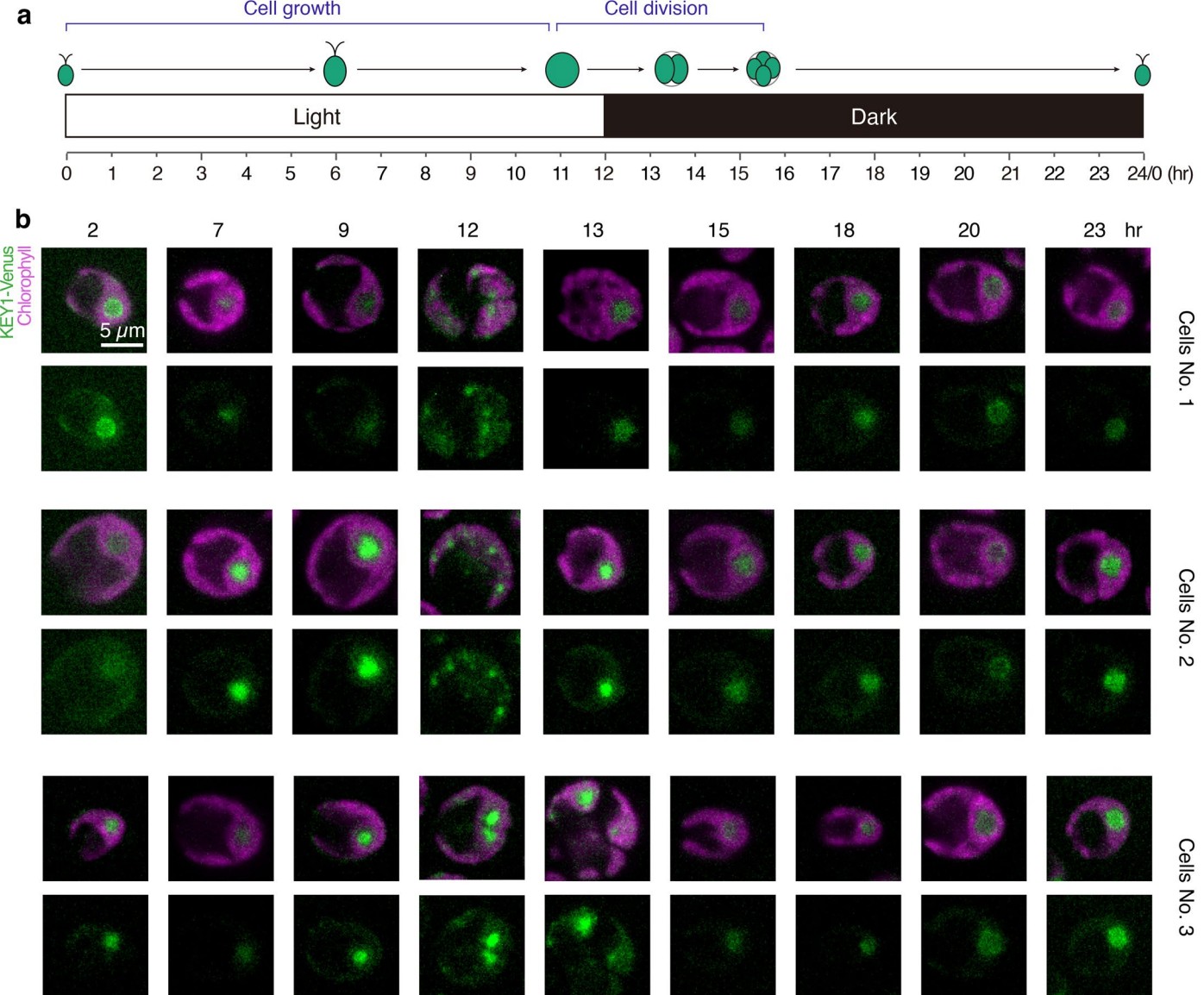

**Extended Data Fig. 6 | KEY1 localizes to the condensates throughout the cell, related to Fig. 5. a**, A cartoon shows Chlamydomonas cells synchronized in a diurnal cycle, where they grow in the light cycle and divide twice in rapid succession upon the shift from light to dark each day. **b**, Confocal images of *key1-1;KEY1-Venus* cells throughout the cell cycle. At each time point, three representative cells are shown. Magenta shows chlorophyll autofluorescence mid-plane, and green shows KEY1–Venus mid-plane. All image acquisition and display parameters for the Venus channel are constant across images. The experiment was performed 1 time; results were consistent with 3 independent measurements at t = 9 h, 3 independent measurements of mixed cell cycle, and *key1-1;KEY1-SNAP* diurnal localization at all time points.

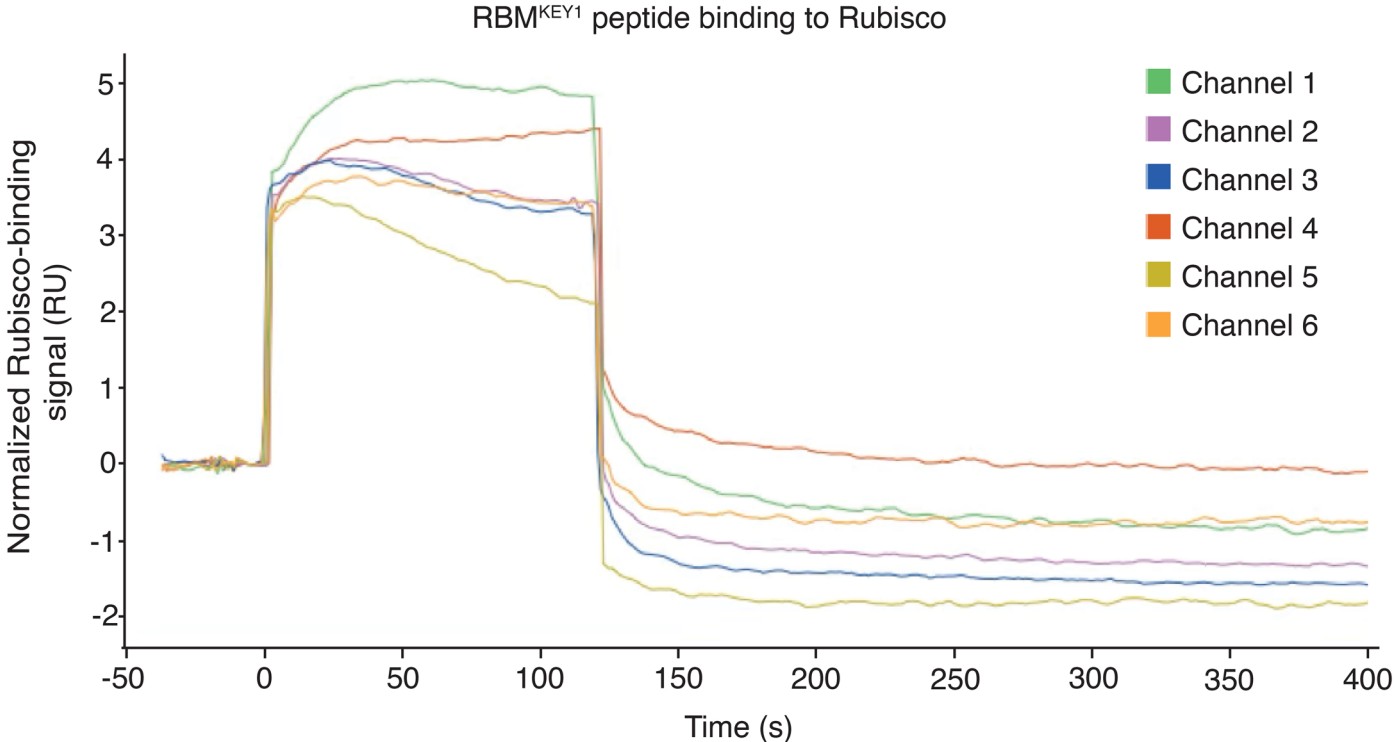

**Extended Data Fig. 7 | The Rubisco-binding motif of KEY1 directly binds to Rubisco holoenzyme, related to Fig. 5.** Double-subtracted sensorgrams showing the normalized binding signal of the RBM^KEY1 peptide to Rubisco on each of the six channels (see also Methods). RU, response unit.

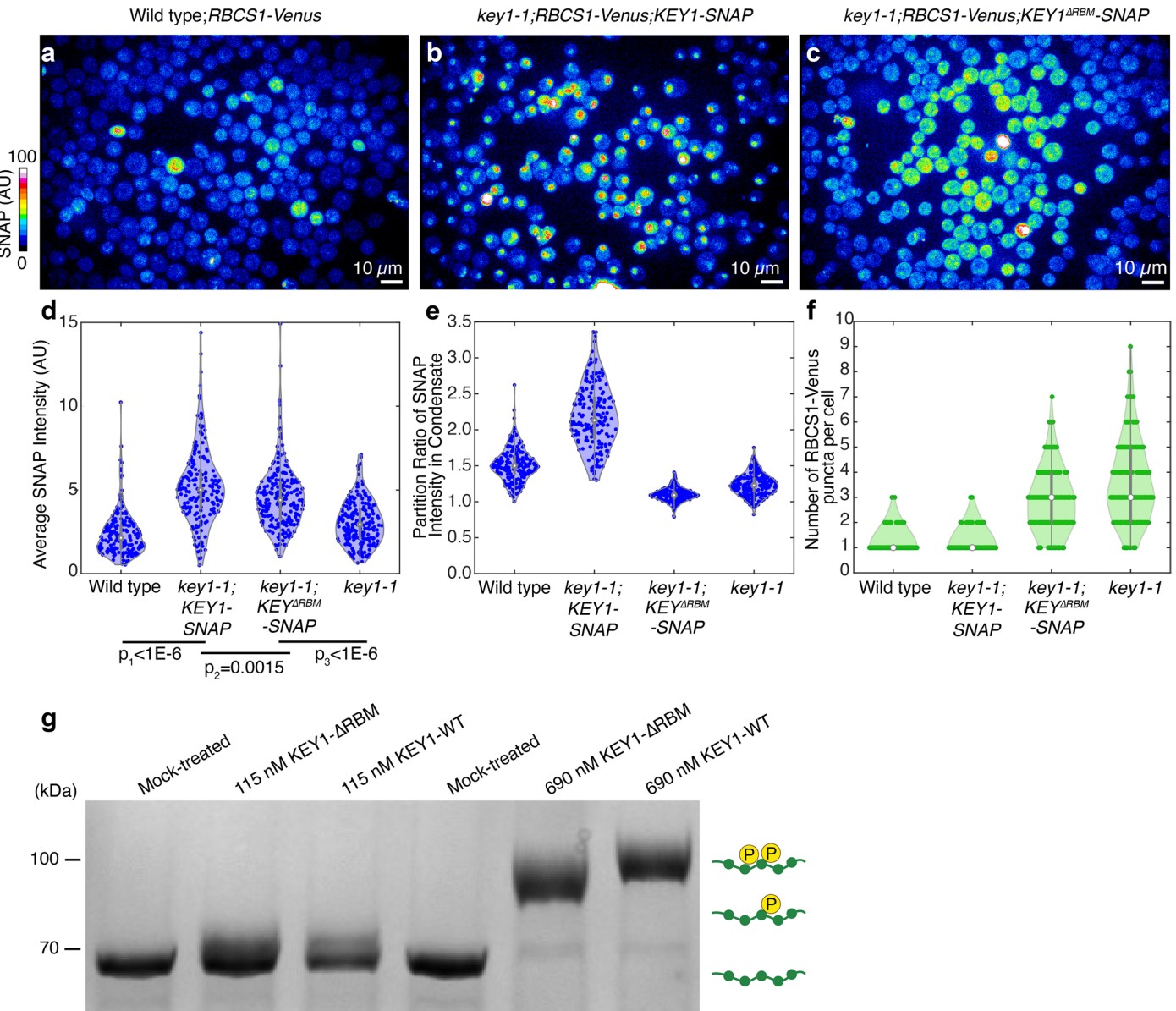

**Extended Data Fig. 8 | A Rubisco-binding motif promotes targeting of KEY1 to the condensate, related to Fig. 5. a-c,** Full fields of view of maximum z-projection of SNAP-stained cells from which representative individual cells were shown in Fig. 5a. Shown are wild type (**a**), *key1-1;KEY1-SNAP* (**b**), and *key1-1; KEY1^ΔRBM-SNAP* (**c**) all expressing RBCS1–Venus. The colour bar indicates SNAP intensity. All image acquisition and display parameters are identical between the three images. **d-e,** Violin plots showing average SNAP intensity in the cell (**d**) and partition ratio of SNAP intensity into pyrenoid condensate (**e**) in the cells expressing RBCS1–Venus with the background of wild type (n = 188 cells), *key1-1;KEY1-SNAP* (n = 188 cells), *key1-1;KEY1^ΔRBM-SNAP* (n = 215 cells), and *key1-1* mutant (n = 208 cells) from one experiment. Additional replicates showed qualitatively similar results. Partition ratio of SNAP intensity is calculated as the mean intensity inside the condensate divided by the mean intensity outside the condensate; a partition ratio of 1 indicates no enrichment. White circles indicate median of the distributions. Grey bars show first (thick) and third (thin) quartiles of the distributions. *P* values $p_1 = 10^{-53}$, $p_2 = 0.0015$, $p_3 = 10^{-31}$ were calculated using a two-sided t-test. **f,** Violin plot showing the number of pyrenoid condensates (visualized by RBCS1–Venus) per cell in the background of wild type (n = 188 cells), *key1-1;KEY1-SNAP* (n = 188 cells), *key1-1;KEY1^ΔRBM-SNAP* (n = 215 cells) and *key1-1* mutant (n = 208 cells) from one experiment. Additional replicates showed qualitatively similar results. White circles indicate median of the distributions. Grey bars show first (thick) and third (thin) quartiles of the distributions. **g,** A Phos-tag gel stained with Coomassie of in vitro activity assay with KEY1-WT and KEY1-ΔRBM on EPYC1–GFP.

**a**

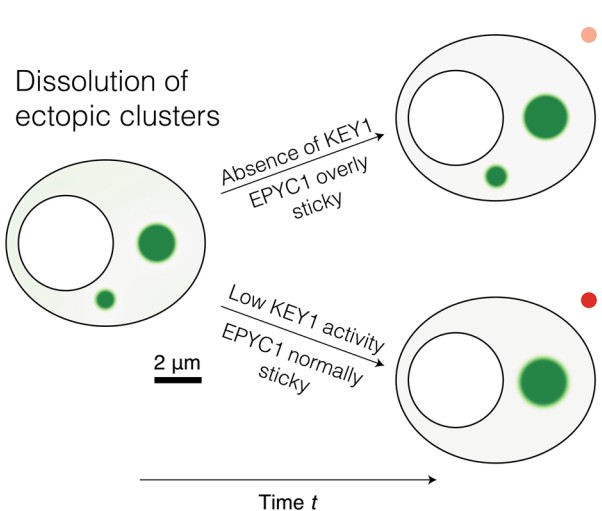

**b**

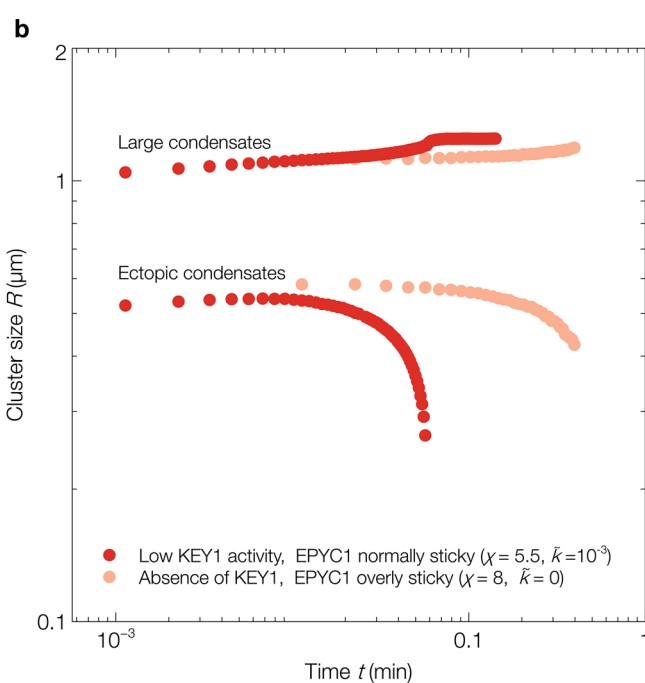

**Extended Data Fig. 9 | Ectopic clusters dissolve more slowly in the absence of KEY1. a**, Dynamical simulations of our model starting from an initial condition with a small ectopic condensate and a large condensate located in the canonical pyrenoid position. Top: without KEY1 activity ($\bar{k} = 0$) EPYC1 is overly sticky ($\chi = 8$), and the small condensate does not dissolve over time. Bottom: at low KEY1 activity ($\bar{k} = 10^{-3}$), EPYC1 has normal stickiness ($\chi = 5.5$) and the small ectopic condensate dissolves over time (Fig. 6g). **b**, Large and ectopic cluster size as a function of time for the two cases shown in panel a.

# Reporting Summary

## Statistics

For all statistical analyses, confirm that the following items are present in the figure legend, table legend, main text, or Methods section.

| n/a | Confirmed | |
|---|---|---|
| ☐ | ☒ | The exact sample size (*n*) for each experimental group/condition, given as a discrete number and unit of measurement |
| ☐ | ☒ | A statement on whether measurements were taken from distinct samples or whether the same sample was measured repeatedly |
| ☐ | ☒ | The statistical test(s) used AND whether they are one- or two-sided *Only common tests should be described solely by name; describe more complex techniques in the Methods section.* |
| ☒ | ☐ | A description of all covariates tested |
| ☒ | ☐ | A description of any assumptions or corrections, such as tests of normality and adjustment for multiple comparisons |
| ☐ | ☒ | A full description of the statistical parameters including central tendency (e.g. means) or other basic estimates (e.g. regression coefficient) AND variation (e.g. standard deviation) or associated estimates of uncertainty (e.g. confidence intervals) |
| ☐ | ☒ | For null hypothesis testing, the test statistic (e.g. *F*, *t*, *r*) with confidence intervals, effect sizes, degrees of freedom and *P* value noted *Give P values as exact values whenever suitable.* |
| ☒ | ☐ | For Bayesian analysis, information on the choice of priors and Markov chain Monte Carlo settings |
| ☒ | ☐ | For hierarchical and complex designs, identification of the appropriate level for tests and full reporting of outcomes |
| ☒ | ☐ | Estimates of effect sizes (e.g. Cohen's *d*, Pearson's *r*), indicating how they were calculated |

*Our web collection on statistics for biologists contains articles on many of the points above.*

## Software and code

Policy information about availability of computer code

| Data collection | Microscopy data was collected using Zeiss Zen Blue, Nikon Elements, VisiView. and MicroManager. SPR data was collected Biacore Software. |
|---|---|
| Data analysis | Mass Spec data was analyzed using Proteome Discoverer 2.5 and Scaffold 5 software. Microscopy data was analyzed in FIJI and with custom MATLAB scripts available on GitHub (https://github.com/linnealemma/KEY1_He-Lemma-etal). SPR data was analyzed using Biacore Evaluation Insight Software. Simulations were performed in COMSOL (https://github.com/amcalv/2025-Simulations-Kinase-KEY1-paper). |

For manuscripts utilizing custom algorithms or software that are central to the research but not yet described in published literature, software must be made available to editors and reviewers. We strongly encourage code deposition in a community repository (e.g. GitHub). See the Nature Portfolio guidelines for submitting code & software for further information.

## Data

Policy information about availability of data

All manuscripts must include a data availability statement. This statement should provide the following information, where applicable:
- Accession codes, unique identifiers, or web links for publicly available datasets
- A description of any restrictions on data availability
- For clinical datasets or third party data, please ensure that the statement adheres to our policy

Data available in the supplemental information and additional supporting data is available on request from M.C. Jonikas.

# Research involving human participants, their data, or biological material

Policy information about studies with [human participants or human data](). See also policy information about [sex, gender (identity/presentation), and sexual orientation]() and [race, ethnicity and racism]().

| Reporting on sex and gender | NA |
| --- | --- |
| Reporting on race, ethnicity, or other socially relevant groupings | NA |
| Population characteristics | NA |
| Recruitment | NA |
| Ethics oversight | NA |

Note that full information on the approval of the study protocol must also be provided in the manuscript.

# Field-specific reporting

Please select the one below that is the best fit for your research. If you are not sure, read the appropriate sections before making your selection.

☒ Life sciences ☐ Behavioural & social sciences ☐ Ecological, evolutionary & environmental sciences

For a reference copy of the document with all sections, see [nature.com/documents/nr-reporting-summary-flat.pdf]()

# Life sciences study design

All studies must disclose on these points even when the disclosure is negative.

| Sample size | No statistical methods were used to pre-determine sample sizes. Sample sizes were determined by convention in the literature and quality of the data to distinguish wild-type behaviors from mutant cell behaviors. |
| --- | --- |
| Data exclusions | For live cell microscopy of fluorescently-tagged Chlamydomonas strains, cells were excluded from analysis if they were dead, as assessed by their autofluorescence at 560 nm excitation. Images were excluded from analysis if the offset between the fluorescent channels caused the outline of the cell to shift between colors. |
| Replication | All experiments were independently repeated as described below.<br><br>Co-immunoprecipitation mass spectrometry experiment was performed 2 times and is consistent with previous study (Mackinder et al., 2017; PMID: 28938113). (Fig. 1b-c)<br>TEMs were performed 2 times and results were consistent with fluorescence microscopy data. (Fig. 1d-e)<br>Imaging of wild-type, key1 mutants and KEY1 rescue pyrenoids was performed 3 times with similar results. (Fig.1f-h)<br>Spot test assay was performed 3 times with similar results. (Fig. 1k)<br>The confocal imaging of pyrenoid dynamics during cell division was performed independently 3 times (all EPYC1-Venus strains and key1-1;RBCS1-Venus;KEY1-SNAP) and 5 times (Wild type and key1-1 with RBCS1-Venus). Results between replicates were similar (Fig. 2a,b,d,e; Extended Data Fig. 2b,c,e,f,h,i).<br>Phos-tag gel-based western blot on cell lysate was performed 3 times with similar results (Fig. 3a).<br>Phos-tag gel-based western blot on cell lysate treated with KEY1 or phosphatase was performed 1 time and was consistent with Fig. 3a and Fig. 3c (Fig. 3b, Extended Data 5c).<br>The Coomassie Phos-tag gel for in vitro KEY1 activity on EPYC1 was performed 3 independent times. Results between replicates were similar (Fig. 3c).<br>Mass spec on in vivo EPYC1 was not replicated, but is consistent with previous studies (Wang et al., 2014; doi.:10.1074/mcp.M114.038281) (Fig. 3d).<br>Mass spec on in vitro EPYC1 protein was performed 2 independent times (Fig. 3e-g).<br>The diurnal Phos-tag gel-based western blot was performed 3 independent times with similar results (Fig. 3k).<br>The in vitro phase diagram was measured 3 times (Fig. 4a-d).<br>The FCS experiment to measure the binding between Rubisco and phosphorylated EPYC1 was performed 2 times (Fig. 4f).<br>The pyrenoid Phos-tag gel-based western blot was performed 3 independent times with similar results (Fig. 4g).<br>The SNAP labeling and confocal imaging of KEY1-SNAP and KEY1ΔRBM-SNAP were performed 3 independent times with n>50 cells for each strain in each replicate. Results between replicates were similar (Fig. 5a, Extended Data Fig. 8a-c).<br>SPR measurements of KEY1's Rubisco binding motif interaction with Rubisco was performed 8 times (Fig. 5d, Extended Data Fig. 7).<br>The Phos-tag gel-based western blot on KEY1-SNAP and KEY1ΔRBM-SNAP was performed 2 times with similar result (Fig. 5e).<br>RT-qPCR was performed 2 times (Extended Data Fig. 1i).<br>The time course of EPYC1-Venus was performed 4 independent times with n=10 cells for each experiment with similar results (Extended Data Fig. 3).<br>The confocal imaging to the phosphonull mutant with EPYC1-Venus was performed independently 3 times with similar results (Extended Data Fig. 4).<br>The Coomassie and anti-His western for KEY1 protein purifications was performed once at the time of purification. Purity was assessed two times later for E. coli-purified KEY1 through Coomassie stain with similar results (Extended Data Fig 5a,b). |

The Coomassie Phos-tag gel of the samples sent for mass spec was performed 2 times, once for each mass spec replicate. The results between replicates were similar (Extended Data Fig. 5d).
Diurnal imaging of KEY1-Venus localization was performed once. Results were consistent with 3 independent measurements at t=9 hours, 3 independent measurements of mixed cell cycle, and KEY1-SNAP diurnal localization (Extended Data Fig. 6b).
The in vitro assay for KEY1-ΔRBM activity on EPYC1 was performed independently 3 times with similar results (Extended Data Fig. 8g).

Randomization — No randomization was used in this study.

Blinding — No blinding was used in this study.

# Reporting for specific materials, systems and methods

We require information from authors about some types of materials, experimental systems and methods used in many studies. Here, indicate whether each material, system or method listed is relevant to your study. If you are not sure if a list item applies to your research, read the appropriate section before selecting a response.

## Materials & experimental systems

| n/a | Involved in the study |
|---|---|
| ☐ | ☒ Antibodies |
| ☐ | ☒ Eukaryotic cell lines |
| ☒ | ☐ Palaeontology and archaeology |
| ☒ | ☐ Animals and other organisms |
| ☒ | ☐ Clinical data |
| ☒ | ☐ Dual use research of concern |
| ☒ | ☐ Plants |

## Methods

| n/a | Involved in the study |
|---|---|
| ☒ | ☐ ChIP-seq |
| ☒ | ☐ Flow cytometry |
| ☒ | ☐ MRI-based neuroimaging |

## Antibodies

Antibodies used — Rabbit IgG in Goat, polyclonal secondary HRP from Invitrogen Cat#: P131466. Rabbit polyclonal anti-EPYC1 from Mackinder et al., 2016.

Validation — Validated in previous study Mackinder et al., 2016.

## Eukaryotic cell lines

Policy information about cell lines and Sex and Gender in Research

Cell line source(s) — C. reinhardtii wild-type cells were CC-4453 , mating type (+).
C. reinhardtii key1 mutant cells were from CLiP mutant library. They can be found at the Chlamydomonas Resource Center: key1-1 LMJ.RY0402.107748 and key1-2 LMJ.RY0402.168929.
Remaining strains were generated for the study and are described in Supplementary Table 4. They are available through the Chlamydomonas Resource Center.

Authentication — key1 mutant cassette insertion sites were validated by PCR as described in the Methods.

Mycoplasma contamination — NA

Commonly misidentified lines (See ICLAC register) — NA

## Plants

Seed stocks — NA

Novel plant genotypes — NA

Authentication — NA

