## [Peer Review File · Nature Cell Biology]

Kinase KEY1 controls pyrenoid condensate size throughout the cell cycle by disrupting phase separation interactions

Corresponding Author: Dr Martin Jonikas

Version 0:

Decision Letter:

*Please delete the link to your author homepage if you wish to forward this email to co-authors.

Dear Dr Jonikas,

Your manuscript, "Kinase KEY1 controls pyrenoid condensate size throughout the cell cycle by disrupting phase separation interactions", has now been seen by 3 referees, who are experts in biomolecular condensation (referees 1 and 2); and CO₂ concentrating mechanisms (referee 3). As you will see from their comments (attached below) they find this work of potential interest, but have raised substantial concerns, which in our view would need to be addressed with considerable revisions before we can consider publication in Nature Cell Biology.

Nature Cell Biology editors discuss the referee reports in detail within the editorial team, including the chief editor, to identify key referee points that should be addressed with priority, and requests that are overruled as being beyond the scope of the current study. To guide the scope of the revisions, I have listed these points below. We are committed to providing a fair and constructive peer-review process, so please feel free to contact me if you would like to discuss any of the referee comments further.

In particular, it would be essential to:

- A.) Experimentally test functional effects of KEY1 kinase activity (all Reviewers)
- B.) Further characterize condensate behaviour, as well as test the effects of condensate behaviour on Rubisco activity (all Reviewers)
- C.) Justify and experimentally test the conditions and predictions from the simulations (Reviewers #1 and #2).
- D.) All other referee concerns pertaining to strengthening existing data, providing controls, methodological details, clarifications and textual changes, should also be addressed.
- E.) Finally please pay close attention to our guidelines on statistical and methodological reporting (listed below) as failure to do so may delay the reconsideration of the revised manuscript. In particular please provide:
 - a Supplementary Figure including unprocessed images of all gels/blots in the form of a multi-page pdf file. Please ensure that blots/gels are labeled and the sections presented in the figures are clearly indicated.
 - a Supplementary Table including all numerical source data in Excel format, with data for different figures provided as different sheets within a single Excel file. The file should include source data giving rise to graphical representations and statistical descriptions in the paper and for all instances where the figures present representative experiments of multiple independent repeats, the source data of all repeats should be provided.

We would be happy to consider a revised manuscript that would satisfactorily address these points, unless a similar paper is published elsewhere, or is accepted for publication in Nature Cell Biology in the meantime.

- ensure that it conforms to our format instructions and publication policies (see below and <https://www.nature.com/nature/for-authors>).

- provide a point-by-point rebuttal to the full referee reports verbatim, as provided at the end of this letter.

- provide the completed Reporting Summary (found here <https://www.nature.com/documents/nr-reporting-summary.pdf>). This is essential for reconsideration of the manuscript will be available to editors and referees in the event of peer review. For more information see <http://www.nature.com/authors/policies/availability.html> or contact me.

Nature Cell Biology is committed to improving transparency in authorship. As part of our efforts in this direction, we are now requesting that all authors identified as 'corresponding author' on published papers create and link their Open Researcher and Contributor Identifier (ORCID) with their account on the Manuscript Tracking System (MTS), prior to acceptance. ORCID helps the scientific community achieve unambiguous attribution of all scholarly contributions. You can create and link your ORCID from the home page of the MTS by clicking on 'Modify my Springer Nature account'. For more information please visit www.springernature.com/orcid.

This journal strongly supports public availability of data. Please place the data used in your paper into a public data repository, or alternatively, present the data as Supplementary Information. If data can only be shared on request, please explain why in your Data Availability Statement, and also in the correspondence with your editor. Please note that for some data types, deposition in a public repository is mandatory - more information on our data deposition policies and available repositories appears below.

Link Redacted

We would like to receive a revised submission within six months.

We hope that you will find our referees' comments, and editorial guidance helpful. Please do not hesitate to contact me if there is anything you would like to discuss.

Best wishes,

Daryl

Daryl Jason Verzosa David, PhD

Senior Editor, Nature Cell Biology
Advisory Editor, npj Biological Physics and Mechanics
Nature Portfolio

Heidelberger Platz 3, 14197 Berlin, Germany
Email: daryl.david@nature.com
ORCID: <https://orcid.org/0000-0002-9253-4805>

Reviewers' Comments:

Reviewer #1 (Remarks to the Author):

The manuscript "Kinase KEY1 controls pyrenoid condensate size throughout the cell cycle by disrupting phase separation

interactions” by He, Lemma, Martinez-Calvo, et al. examines the condensation and dissolution dynamics of the pyrenoid membraneless organelle of the alga *Chlamydomonas reinhardtii*. They find a previously un-annotated kinase “KEY1” that acts on the Rubisco linker protein EPYC1 regulating its interactions with Rubisco and disrupting their co-phase separation. They demonstrate that the upregulation of KEY1’s activity is important to dissolve the pyrenoid at key steps in the cell cycle. They develop a minimal continuum mathematical model that demonstrates that the relative activity of a kinase like KEY1, and a countering phosphatase can control condensate dissolution, reformation, number, size, and even affect the position of the condensate. In the case of the pyrenoid, high KEY1 activity results in dissolution of the condensate, while low activity results in a single large condensate that would center itself within its available volume, much like the pyrenoid does. This is a well written paper with well-designed experiments, and mostly clean data. The paper advances the understanding of the regulation of a condensate that is necessary to ensure proper cell cycle dynamics and equal inheritance into daughter cells. The modeling helps generalize the findings and advance the field’s understanding of how condensate sizes and behaviors may be controlled within cells. We recommend it for publication once our points below are addressed satisfactorily.

Major comments

1. The authors ask the question how KEY1 mediated phosphorylation of EPYC1 reduces EPYC1-Rubisco co-condensation, and they come up with two proposals: “(1) that EPYC1 phosphorylation could cause EPYC1 dissociation from Rubisco, or (2) EPYC1 phosphorylation could promote the formation of small stable complexes of 2-3 molecules of EPYC1 and single Rubisco holoenzyme in the dilute phase”. This basically amounts to (1) the disruption of the network or (2) solubilization, i.e. counteracting of the density transition.

The authors go on to show that EPYC1 is phosphorylated on its Rubisco binding motif, which suggests that the network is destroyed. However, their phase diagram of EPYC1 and Rubisco (Figure 4C) suggests that networking with Rubisco is not needed and is in fact counterproductive towards EPYC1 phase separation. This phase diagram needs to be more finely mapped to get a better understanding of the driving forces for phase separation. As it stands, the shape of the two-phase regime that is implied by drawing is dependent on a single data point. Furthermore, it is at odds with the interpretation in the text as we point out above. In the text, starting at line 278, they discuss that the “EPYC1-mediated inter-Rubisco interactions” are required for condensation, however this phase diagram indicates otherwise. It shows EPYC1 phase separating more readily on its own than it does with Rubisco, indicating that Rubisco solubilizes EPYC1 and reduces the condensation. It would of course be possible that condensation is weakened by solubilization even if the Rubisco binding sites are affected. The wording of possibility (2) should be clarified. It is unclear why the authors propose this specific stoichiometry.

2. Figure 5: The analysis of how the Rubisco binding motif might act as an autoinhibitory sequence and localizes KEY1 to the condensate seems problematic. If the Rubisco binding motif (RBM) acts as an autoinhibitory sequence, then shouldn’t the RBM mutant of KEY1 be a hyperactive version of the kinase? The fraction of EPYC1 in the supernatant in Figure 4f would be phosphorylated by hyperactive KEY1, preventing it from returning to the condensate. According to the model in Figure 6, this would result in multiple small condensates, as perhaps suggested in Figure 5a. However, no phosphorylated EPYC1 is observed in Figure 5e in the presence of the mutant. This makes us wonder whether KEY1 RBM is a functional kinase. If it is not functional, then the question is whether the condensates are still size controlled or not. It is difficult to tell from Figure 5e. Is there any evidence that the mutant still functions as a kinase?

3. It would be very interesting to test the mathematical model. This could e.g. be achieved by overexpressing KEY1 to test whether this results indeed in small but uniformly sized and spaced condensates.

4. The mathematical model nicely explains how the driving force for phase separation varies with KEY1 activity, and the centering effect. It is clear that KEY activity disfavors the formation of extra small condensates. But why does it favor coarsening into a single large condensate?

Minor comments

Figure 1: It would be helpful to clarify what the purpose of the TP vs. TAP media is for a broader audience.

Figure 3: Are the phosphorylation sites indicated correctly? 3e indicates multiple sites on the *E. coli*-expressed protein without the addition of the KEY1 Kinase. This seems like an error.

Figure 5d is a non-standardized way of displaying SPR data. Please provide the standard SPR traces.

Reviewer #2 (Remarks to the Author):

Final review

Note: I collaborated on this review with an exceptional predoctoral colleague, hence the use of “we” throughout the enclosed comments.

Summary of insights

In this paper, He, Lemma, and colleagues investigate how the size and number of a biological condensate organelle called pyrenoid is regulated for function in *Chlamy*. First, they identify a kinase KEY1 as a phenotypic (size, number) and functional (reduced growth under low CO₂ condition) regulator of the pyrenoid through a genetic mutant allele *key1-1*. Through in vitro experiments, they show that KEY1 phosphorylates EPYC1, which directly disrupts EPYC1-Rubisco phase separation. In cells, they show that *key1-1* mutant cells lack number/size regulation and that KEY1 expression (or activity) and downstream EPYC1 phosphorylation are both correlated across the cell cycle. They also perform multiple orthogonal experiments to postulate a mechanism by which KEY1 is recruited into the pyrenoid, binds to Rubisco to “activate” its kinase (through auto-inhibition), and in turn, phosphorylate EPYC1 to evacuate it from the pyrenoid. Finally, the authors develop a simple physics-based model to synthesize these distinct observations, which in turn, recapitulates experimentally observed size regulation

as well as predicting other observed features, including centering and inter-pyrenoid spacing. Overall, the use of many distinct techniques, experimental and modeling, provides a solid foundation for their proposed model.

Below are a list of concerns (further stratified into major/minor) and confusion we had while reviewing the paper - addressing which we believe would improve the manuscript and strengthen claims made in the text.

Major concerns

1. Clarification of Key1 Mutant Phenotype and Behavior (mostly Fig 1, S1-2, and text)

The authors rely on a mutant cell type of key1 to compare to wild-type cells for if and how KEY1 regulated pyrenoid number and size.

A. However, the authors do not directly describe what the mutation was to key1 and its resulting phenotype in KEY1 behavior. Can the authors describe more clearly?

B. Is KEY1 expressed at all in the key1-1 line? If the mutant has no expression, then that should be clearly stated up front to contrast. As such, this was hard for the reviewers to understand how key1-1 was different from the wild-type.

C. If no key1 is expressed - is the rate of coarsening just vastly slower not leading to 1 single pyrenoid or do the authors think there are other mechanisms? This was hard to understand from the paper or model presented.

By contrast, if it is expressed at all in the mutant lines, then the following points are unclear:

D. Does it still localize to the pyrenoid, albeit at lower levels?

E. Is there a similar spike in mRNA levels of KEY1 in the mutant lines? What are the mechanisms that prevent the multiple condensates from dissolving but can still cause them to be smaller?

F. As a corollary, if the key1 mutant has aberrant but lower activity throughout, shouldn't there be only 1 large condensate? Are there compensatory mechanisms that are present?

2. Quantification of Condensed Fraction and Protein Behavior (mostly Figs 2-3, S2-3)

A. Is condensed fraction reported per cell post division or total volume?

B. More generally, are there any statements made about intensities or protein concentrations and how they change? What is the implication about effective concentrations per cell?

C. In S1, the Venus-tag-Rubisco shows much lower cellular condensed fractions at WT condition as opposed to mutants but not for Fig 2 which measures EPYC1 instead? Is this a statement of these two proteins behaving differently? Can they be treated interchangeably as specified?

D. An "ectopic" condensate is shown in Fig S3C for a single representative cell but it would be important to verify this is a statistically reproducible behavior - perhaps by reporting on the number or nucleation rate of condensates in more images?

3. Simulation Assumptions and Kinase Activity (Fig 6)

A. Is total EPYC1 concentration held constant across simulation?

B. Are the authors making any statements about total rates of coarsening when kinases are present versus not present? (Fig 6g) if so, perhaps, this is worth reporting additionally as a net change in rate of dissolution besides particular snapshots.

C. In this model, there is no nucleation (unless I am missing a description of thermal fluctuations in this continuum model) so it is worth explicitly stating that in the main text and briefly explaining how the "condensates" are seeded under the various conditions and how they reform (presumably through spontaneous phase separation?)

D. We also felt that the figure presentation could have been improved as Fig 6. Almost read like a panel of schematics instead of data. It would help to know that these were simulated conditions (for e.g. by adding colorbars, cleanly labeling plots). We say this only to suggest that it took a while for us to dig into the figure and really understand it.

E. The model simplifies activity into 1 effective parameter that is not spatially varying and reports broad correlation in size/number with data. Do the other predictions made (repulsion, centering) also hold true if experiment-motivated connections to models of KEY1 - i.e. explicitly modeling KEY1 species concentrations, effective recruitment to pyrenoid, and pyrenoid-dependent activity? We ask this because experiments suggest that loss of recruitment of KEY1 to pyrenoid does show a functional phenotype in experiments but perhaps in the model this would not reflect directly (or would it only be lumped once again into an effective activity?)

4. Conceptual questions with the overall proposed logic of pyrenoid regulation

A. The authors demonstrate that mutations to key1 will disrupt the regulation of pyrenoid size by showing smaller and more condensates compared to the WT. Additionally, the authors experiment state that higher phosphorylation of EPYC1 dissolves the pyrenoid.

A. If KEY1 activity is disrupted, can the authors explain or clarify why the pyrenoid would not stay as one condensate since this decreases the ability for KEY1 to phosphorylate EPYC1?

B. Do the authors believe there are other constitutive kinases of EPYC1?

5. Text clarity

By and large, the text is well written. The intro specifies "active size regulation is largely unexplored" - this is a subjective statement, obviously. Here we point to the authors for missing citations to functional mechanisms of size control - both passive and active (We are confident we are missing many more) to:

A. <https://doi.org/10.1016/j.devcel.2020.06.003>

B. <https://www.nature.com/articles/s41556-022-00882-3>

C. <https://pubmed.ncbi.nlm.nih.gov/29973724/>

I would recommend adding these and suitably revising text if needed!

Minor concerns

1. Figure 5: Can you report on a partition ratio from SNAP dye for better characterization?
2. The authors reference key1-1 mutants in the main text, but Figure 1 shows data for a key1-2 mutant. Please clarify the difference between key1-1 and key1-2 mutants and why mutants result in similar phenotypes in figure 1k.
3. Under 'Phosphorylation of EPYC1 by KEY1 disrupts EPYC1 binding to Rubisco', the conclusion is that EPYC1 phosphorylated disrupts binding to RUBISCO. The topic sentence hypothesizes this happens by EPYC1 dissociating or making smaller clusters. Please clarify how readers can distinguish between the two based on the data in figure 4 and what was tested to define a difference.
4. Strongly encourage the authors to deposit their simulation code publicly on the github.

Reviewer #3 (Remarks to the Author):

The pyrenoid is a liquid-liquid phase separated organelle present in many microalgae. It is part of the algal CO₂ concentrating mechanism (CCM) which is a key player for global primary productivity. Here, He and coworkers report a new kinase (KEY1) which is critical to control the size of the pyrenoid, they further describe the mechanism involved in how KEY1 regulates the pyrenoid size and develop a simple mathematical model that recapitulates what is shown *in vivo*. The new kinase reported is the first mechanism reported in *Chlamydomonas* to directly control the pyrenoid size, it will be of interest to a broad audience. The work is done carefully and very well described. The manuscript is easy to read and provides controls and multiple independent ways to characterize KEY1's function *in vivo* and *in vitro*, which makes the work very solid. Overall this manuscript is excellent and will have a big impact on the fields of microalgae physiology, CCM, liquid-liquid phase separation and beyond.

I just have one main comment which should be clarified: While the authors claim that KEY1 activity is important for its function, this does not seem to be the case for the pyrenoid's main function of being involved in the CCM. Although the authors do not mention this point directly, they imply it very strongly lines 119-124. However, while the key1 mutant has an impaired growth under low CO₂ and very low CO₂ (Fig. 1F), this growth defect is not specific to CO₂ limiting condition and is also seen at 3% CO₂ or with TAP. Additionally, while the complementing lines show a fully rescued pyrenoid sizes, they do not really show a complemented growth phenotype, suggesting that any growth phenotype is not directly linked to the pyrenoid size control. The authors should clarify what is the "pyrenoid function" they find associated with KEY1 activity and if the CCM is one of them, they should measure it more directly, for example by measuring the cell's affinity for CO₂. As a side note, the spot test 3% CO₂-TP is overgrown and cannot be used to assess a growth difference.

Minor comments:

- I could not find the manuscript on BioRxiv or a similar public preprint repository. If not already out there, please upload it to favor openness of data as early as possible.
- Line 58-59 "most eukaryotic algae", please change "most" for "many" as none of the cited articles have done a thorough analysis of all eukaryotic microalgae
- Line 93 "Kinase of EPYC 1 (KEY1)", very cool naming, good job!
- Line 240-241, although KEY1 is able to phosphorylate EPYC1 at the same residues observed *in vivo*, I don't see what would make it being the "primary kinase" involved as some EPYC 1 residues seem to not be phosphorylated directly by KEY1 *in vitro*, suggesting the existence of other kinases and secondary effects in the key1 mutant.
- Lines 338-345 It seems that the KEY1- Δ RBM is excluded from the pyrenoid. Is that the case? Also, is there some indication that KEY1- Δ RBM can be active?
- Line 359: why do you propose that there would be only one unidentified phosphatase and not multiple ones?

-

READABILITY OF MANUSCRIPTS – Nature Cell Biology is read by cell biologists from diverse backgrounds, many of whom are not native English speakers. Authors should aim to communicate their findings clearly, explaining technical jargon that might be unfamiliar to non-specialists, and avoiding non-standard abbreviations. Titles and abstracts should concisely communicate the main findings of the study, and the background, rationale, results and conclusions should be clearly

explained in the manuscript in a manner accessible to a broad cell biology audience. Nature Cell Biology uses British spelling.

ABSTRACT AND MAIN TEXT – please follow the guidelines that are specific to the format of your manuscript, as listed in our Guide to Authors (http://www.nature.com/ncb/pdf/nbc_gta.pdf) Briefly, Nature Cell Biology Articles, Resources and Technical Reports have 3500 words, including a 150 word abstract, and the main text is subdivided in Introduction, Results, and Discussion sections. Nature Cell Biology Letters have up to 2500 words, including a 180 word introductory paragraph (abstract), and the text is not subdivided in sections.

Methods should be written concisely, but should contain all elements necessary to allow interpretation and replication of the results. As a guideline, Methods sections typically do not exceed 3,000 words. The Methods should be divided into subsections listing reagents and techniques. When citing previous methods, accurate references should be provided and any alterations should be noted. Information must be provided about: antibody dilutions, company names, catalogue numbers and clone numbers for monoclonal antibodies; sequences of RNAi and cDNA probes/primers or company names and catalogue numbers if reagents are commercial; cell line names, sources and information on cell line identity and authentication. Animal studies and experiments involving human subjects must be reported in detail, identifying the committees approving the protocols. For studies involving human subjects/samples, a statement must be included confirming that informed consent was obtained. Statistical analyses and information on the reproducibility of experimental results should be provided in a section titled "Statistics and Reproducibility".

All Nature Cell Biology manuscripts submitted on or after March 21 2016 must include a Data availability statement as a separate section after Methods but before references, under the heading "Data Availability". For Springer Nature policies on data availability see <http://www.nature.com/authors/policies/availability.html>; for more information on this particular policy see <http://www.nature.com/authors/policies/data/data-availability-statements-data-citations.pdf>. The Data availability statement should include:

- Accession codes for primary datasets (generated during the study under consideration and designated as "primary accessions") and secondary datasets (published datasets reanalysed during the study under consideration, designated as "referenced accessions"). For primary accessions data should be made public to coincide with publication of the manuscript. A list of data types for which submission to community-endorsed public repositories is mandated (including sequence, structure, microarray, deep sequencing data) can be found here

<http://www.nature.com/authors/policies/availability.html#data>.

- Unique identifiers (accession codes, DOIs or other unique persistent identifier) and hyperlinks for datasets deposited in an approved repository, but for which data deposition is not mandated (see here for details <http://www.nature.com/sdata/data-policies/repositories>).
- At a minimum, please include a statement confirming that all relevant data are available from the authors, and/or are included with the manuscript (e.g. as source data or supplementary information), listing which data are included (e.g. by figure panels and data types) and mentioning any restrictions on availability.
- If a dataset has a Digital Object Identifier (DOI) as its unique identifier, we strongly encourage including this in the Reference list and citing the dataset in the Methods.

We recommend that you upload the step-by-step protocols used in this manuscript to [protocols.io](https://www.protocols.io). More details can be found at <https://www.protocols.io/help/publish-articles>.

All imaging data should be accompanied by scale bars, which should be defined in the legend. Cropped images of gels/blots are acceptable, but need to be accompanied by size markers, and to retain visible background signal within the linear range (i.e. should not be saturated). The boundaries of panels with low background have to be demarcated with black lines. Splicing of panels should only be considered if unavoidable, and must be clearly marked on the figure, and noted in the legend with a statement on whether the samples were obtained and processed simultaneously. Quantitative comparisons between samples on different gels/blots are discouraged; if this is unavoidable, it should only be performed for samples derived from the same experiment with gels/blots were processed in parallel, which needs to be stated in the legend.

The total number of Supplementary Figures (not including the “unprocessed scans” Supplementary Figure) should not exceed the number of main display items (figures and/or tables (see our Guide to Authors and March 2012 editorial <http://www.nature.com/ncb/authors/submit/index.html#suppinfo>; <http://www.nature.com/ncb/journal/v14/n3/index.html#ed>). No restrictions apply to Supplementary Tables or Videos, but we advise authors to be selective in including supplemental data.

GUIDELINES FOR EXPERIMENTAL AND STATISTICAL REPORTING

REPORTING REQUIREMENTS – We are trying to improve the quality of methods and statistics reporting in our papers. To that end, we are now asking authors to complete a reporting summary that collects information on experimental design and reagents. The Reporting Summary can be found here <https://www.nature.com/documents/nr-reporting-summary.pdf> If you would like to reference the guidance text as you complete the template, please access these flattened versions at <http://www.nature.com/authors/policies/availability.html>.

We strongly recommend the presentation of source data for graphical and statistical analyses as a separate Supplementary Table, and request that source data for all independent repeats are provided when representative experiments of multiple independent repeats, or averages of two independent experiments are presented. This supplementary table should be in Excel format, with data for different figures provided as different sheets within a single Excel file. It should be labelled and numbered as one of the supplementary tables, titled “Statistics Source Data”, and mentioned in all relevant figure legends.

Version 1:

Decision Letter:

Our ref: NCB-A57235A

20th November 2025

Dear Dr. Jonikas,

Thank you for submitting your revised manuscript "Kinase KEY1 controls pyrenoid condensate size throughout the cell cycle by disrupting phase separation interactions" (NCB-A57235A). It has now been seen by the original referees and their comments are below. The reviewers find that the paper has improved in revision, and therefore we'll be happy in principle to publish it in Nature Cell Biology, pending minor revisions to satisfy the referees' final requests and to comply with our editorial and formatting guidelines.

Thank you again for your interest in Nature Cell Biology Please do not hesitate to contact me if you have any questions.

Sincerely,
Daryl

Daryl Jason Verzosa David, PhD

Senior Editor, Nature Cell Biology
Advisory Editor, npj Biological Physics and Mechanics
Nature Portfolio

Heidelberger Platz 3, 14197 Berlin, Germany
Email: daryl.david@nature.com
ORCID: <https://orcid.org/0000-0002-9253-4805>

Reviewer #1 (Remarks to the Author):

The manuscript "Kinase KEY1 controls pyrenoid condensate size throughout the cell cycle by disrupting phase separation interactions" by He, Lemma, Martinez-Calvo, et al. examines the condensation and dissolution dynamics of the pyrenoid membraneless organelle of the alga *Chlamydomonas reinhardtii*. They find a previously un-annotated kinase "KEY1" that acts on the Rubisco linker protein EPYC1 regulating its interactions with Rubisco and disrupting their co-phase separation. They demonstrate that the upregulation of KEY1's activity is important to dissolve the pyrenoid at key steps in the cell cycle. They develop a minimal continuum mathematical model that demonstrates that the relative activity of a kinase like KEY1, and a counteracting phosphatase can control condensate dissolution, reformation, number, size, and even affect the position of the condensate. In the case of the pyrenoid, high KEY1 activity results in dissolution of the condensate, while low activity results in a single large condensate that would center itself within its available volume, much like the pyrenoid does. This is a well-written manuscript with well-designed experiments. The manuscript advances the understanding of the regulation of condensates that are necessary to ensure proper cell cycle dynamics and equal inheritance into daughter cells. The modeling helps generalize the findings and advance the field's understanding of how condensate size and behavior may be controlled within cells. The authors have satisfactorily address our comments, and we recommend the manuscript for publication.

Reviewer #2 (Remarks to the Author):

In this revised manuscript, the authors present a combination of new experiments/data, simulations, and textual edits. We believe they have done a good job of successfully addressing all of our concerns raised during the original submission.

Minor suggestion:

We saw that the github link exists but the repository itself only comprised 2 files, a README and a code file that we couldn't even open or get to run (2 of us, both computational scientists, at least gave it a fair shot each). While we don't think you need to make production quality software, the following are fairly typical features of academic code papers:

- README that clearly documents the steps or provides dependencies for installing the code and associated libraries
- Instructions to run the code to generate outcomes
- Code that has basic documentation - docstrings at the least - for the functions/classes/modules involved
- Typically an example script to generate data - of analysis in the paper or a representative simulation.

Additionally, if the authors feel so inclined - while this is not necessary, they could add:

- adding simulation data used in figures (csv, npy or other files)
- adding clean scripts to simulate model in the parameter/mesh regime studied in the paper

All these are suggestions in the service of helping the authors work become more reproducible and easier to access for scientists interested to understand, extend, or test your model.

Reviewer #2 (Remarks on code availability):

Please see the review response.

Reviewer #3 (Remarks to the Author):

The authors have adequately answered my comments, except I still disagree with their interpretation that KEY1 is required for the "function" of the pyrenoid.

The new data, now shown by the authors in Fig. 1k, rather suggest that KEY1 is required for the pyrenoid's role in the CCM. It would be great if the authors acknowledged this nuance (which I am sure they are aware of).

This is an important nuance as the "function" of the pyrenoid is likely more diverse than just the CCM: the pyrenoid is present in high CO₂ conditions when the CCM is not active (<https://doi.org/10.1073/pnas.1522866113>), and it is also highly increased in size in response to O₂ and H₂O₂ (<https://doi.org/10.7554/eLife.67565>), which led to the suggestion that the pyrenoid might be a micro-oxic niche for O₂-sensitive reactions (<https://www.biorxiv.org/content/10.1101/2023.12.21.572672v1.full>).

This nuance is even more critical, given that the field has drawn significant attention to the role of the pyrenoid in the CCM, risking overlooking other (potentially important) roles it might play. These other roles might actually be quite important for the fundamental understanding of this compartment, as well as what it might mean to engineer it into a different system.

Version 2:

Decision Letter:

Dear Dr Jonikas,

I am pleased to inform you that your manuscript, "Kinase KEY1 controls pyrenoid condensate size throughout the cell cycle by disrupting phase separation interactions", has now been accepted for publication in Nature Cell Biology.

Due to the importance of these deadlines, we ask that you please let us know now whether you will be difficult to contact over the next month. If this is the case, we ask you provide us with the contact information (email, phone and fax) of someone

who will be able to check the proofs on your behalf, and who will be available to address any last-minute problems.

Please note that *Nature Cell Biology* is a Transformative Journal (TJ). Authors may publish their research with us through the traditional subscription access route or make their paper immediately open access through payment of an article-processing charge (APC). Authors will not be required to make a final decision about access to their article until it has been accepted. [Find out more about Transformative Journals](https://www.springernature.com/gp/open-research/transformative-journals)

Authors may need to take specific actions to achieve compliance with funder and institutional open access mandates. If your research is supported by a funder that requires immediate open access (e.g. according to [Plan S principles](https://www.springernature.com/gp/open-science/plan-s-compliance) or the [NIH public access policy](https://www.springernature.com/gp/open-science/us-federal-agency-compliance)) then you should select the gold OA route, and we will direct you to the compliant route where possible. Because authors warrant under our subscription licensing terms that they haven't committed to licensing any version of their article under a licence inconsistent with the terms of our agreement – including the applicable embargo period – publication under the subscription model isn't suitable for authors whose funders require no embargo.

If you have not already done so, we strongly recommend that you upload the step-by-step protocols used in this manuscript to protocols.io (<https://protocols.io>), an open online resource that allows researchers to share their detailed experimental know-how. All uploaded protocols are made freely available and are assigned DOIs for ease of citation. Protocols and Nature Portfolio journal papers in which they are used can be linked to one another, and this link is clearly and prominently visible in the online versions of both. Authors who performed the specific experiments can act as primary authors for the Protocol as they will be best placed to share the methodology details, but the Corresponding Author of the present research paper should be included as one of the authors. By uploading your Protocols onto protocols.io, you are enabling researchers to more readily reproduce or adapt the methodology you use, as well as increasing the visibility of your protocols and papers. You can also establish a dedicated workspace to collect your lab Protocols. Further information can be found at <https://www.protocols.io/help/publish-articles>.

Nature Cell Biology encourages authors presenting evidence for cell, biological, molecular, and genetic interactions to consider communicating these findings using Biofactoid (<https://biofactoid.org/>). This tool helps users share a searchable representation of interactions (e.g. binding, gene expression, post-translational modification) between genes, gene products, or chemicals. Information added to Biofactoid, with author attribution, is shared on social media and public databases, such as Pathway Commons, where it can be discovered and analyzed in the context of a large and growing corpus of knowledge.

With kind regards,
Daryl

Daryl Jason Verzosa David, PhD

Senior Editor, Nature Cell Biology
Advisory Editor, npj Biological Physics and Mechanics
Nature Portfolio

Heidelberger Platz 3, 14197 Berlin, Germany
Email: daryl.david@nature.com
ORCID: <https://orcid.org/0000-0002-9253-4805>

** Visit the Springer Nature Editorial and Publishing website at http://editorial-jobs.springernature.com?utm_source=ejp_NCB_email&utm_medium=ejp_NCB_email&utm_campaign=ejp_NCB for more information about our career opportunities. If you have any questions please click [here](mailto:editorial.publishing.jobs@springernature.com).

Point-by-Point Response to Reviewer input on the He, Lemma et al. manuscript

We are grateful to all the Reviewers for their helpful input, which we feel has resulted in substantial improvements to the manuscript. Below, we provide a point-by-point response to the Reviewers' comments.

We note that Word numbers lines differently with tracked changes shown vs. not shown. The line numbers we indicate here are for the version with tracked changes shown.

Reviewers' Comments:

Reviewer #1 (Remarks to the Author):

The manuscript "Kinase KEY1 controls pyrenoid condensate size throughout the cell cycle by disrupting phase separation interactions" by He, Lemma, Martinez-Calvo, et al. examines the condensation and dissolution dynamics of the pyrenoid membraneless organelle of the alga *Chlamydomonas reinhardtii*. They find a previously un-annotated kinase "KEY1" that acts on the Rubisco linker protein EPYC1 regulating its interactions with Rubisco and disrupting their co-phase separation. They demonstrate that the upregulation of KEY1's activity is important to dissolve the pyrenoid at key steps in the cell cycle. They develop a minimal continuum mathematical model that demonstrates that the relative activity of a kinase like KEY1, and a countering phosphatase can control condensate dissolution, reformation, number, size, and even affect the position of the condensate. In the case of the pyrenoid, high KEY1 activity results in dissolution of the condensate, while low activity results in a single large condensate that would center itself within its available volume, much like the pyrenoid does.

This is a well written paper with well-designed experiments, and mostly clean data. The paper advances the understanding of the regulation of a condensate that is necessary to ensure proper cell cycle dynamics and equal inheritance into daughter cells. The modeling helps generalize the findings and advance the field's understanding of how condensate sizes and behaviors may be controlled within cells. We recommend it for publication once our points below are addressed satisfactorily.

We are grateful for the Reviewer's positive comments.

Major comments

1. The authors ask the question how KEY1 mediated phosphorylation of EPYC1 reduces EPYC1-Rubisco co-condensation, and they come up with two proposals: "(1) that EPYC1 phosphorylation could cause EPYC1 dissociation from Rubisco, or (2) EPYC1 phosphorylation could promote the formation of small stable complexes of 2-3 molecules of EPYC1 and single Rubisco holoenzyme in the dilute phase". This basically amounts to (1) the disruption of the network or (2) solubilization, i.e. counteracting of the density transition.

The authors go on to show that EPYC1 is phosphorylated on its Rubisco binding motif, which suggests that the network is destroyed. However, their phase diagram of EPYC1 and Rubisco (Figure 4C) suggests that networking with Rubisco is not needed and is in fact counterproductive towards EPYC1 phase separation. This phase diagram needs to be more finely mapped to get a better understanding of the driving forces for phase separation. As it stands, the shape of the two-phase regime that is implied by drawing is dependent on a single data point. Furthermore, it is at odds with the interpretation in the text as we point out above. In the text, starting at line 278, they discuss that the “EPYC1-mediated inter-Rubisco interactions” are required for condensation, however this phase diagram indicates otherwise. It shows EPYC1 phase separating more readily on its own than it does with Rubisco, indicating that Rubisco solubilizes EPYC1 and reduces the condensation. It would of course be possible that condensation is weakened by solubilization even if the Rubisco binding sites are affected.

We are very grateful to the Reviewer for pointing out a problem with our previous Figure 4c. Upon investigation, we think the phase separation we observed for EPYC1 in the absence of Rubisco was likely due to contamination of the EPYC1 with *E. coli* DNA. **We have repeated our EPYC1 purification with nuclease to limit DNA contamination, which we confirmed by measuring the 260/280 absorbance (now explained in Methods lines 1753-1755).** The new phase diagram shows a dependence of phase separation on the presence of both Rubisco and EPYC1, which agrees with previously published work showing that both factors are required for condensates^{1,2}. **We have updated Fig. 4a-d with the new results.**

The wording of possibility (2) should be clarified. It is unclear why the authors propose this specific stoichiometry.

This was a common comment among the Reviewers, and **we have rewritten the section (lines 324-370) and created a new figure panel (Fig. 4e)** to more clearly convey the two hypotheses for how KEY1 phosphorylation disrupts phase separation. The small-complex hypothesis (previously Hypothesis 2, now Hypothesis 1 in the revised text) does not rely on a specific stoichiometry. Instead, the hypothesis is that the phosphorylation of EPYC1 would enhance the formation of small complexes with one or more EPYC1 molecules occupying the binding sites of one Rubisco molecule, preventing its interaction with other Rubisco molecules and thus disfavoring phase separation. Such small complexes were previously identified in the dilute phase of Rubisco-EPYC1 mixtures *in vitro*¹.

2. Figure 5: The analysis of how the Rubisco binding motif might act as an autoinhibitory sequence and localizes KEY1 to the condensate seems problematic. If the Rubisco binding motif (RBM) acts as an autoinhibitory sequence, then shouldn't the RBM mutant of KEY1 be a hyperactive version of the kinase? The fraction of EPYC1 in the supernatant in Figure 4f would be phosphorylated by hyperactive KEY1, preventing it from returning to the condensate. According to the model in Figure 6, this would result in multiple small condensates, as perhaps suggested in Figure 5a. However, no phosphorylated EPYC1 is observed in Figure 5e in the presence of the mutant. This makes us wonder whether KEY1 RBM is a functional kinase. If it is not functional,

then the question is whether the condensates are still size controlled or not. It is difficult to tell from Figure 5e. Is there any evidence that the mutant still functions as a kinase?

We thank the Reviewer for thinking about the proposed autoinhibitory model. To address this point, we further explored the impact of the Δ RBM mutation on KEY1 activity.

To test the activity of the Δ RBM mutant kinase, we compared the *in vitro* activity of *E. coli*-purified KEY1- Δ RBM and KEY1-WT (Extended Data Fig. 8g). If the auto-inhibition hypothesis were true, we would expect that the KEY1- Δ RBM mutant would be more active than the wild-type. However, our results indicate that the activity of KEY1- Δ RBM was not substantially different from that of KEY1-WT.

This experiment and analysis have led us to conclude that it is unlikely that autoinhibition through the RBM is acting to control kinase activity. Rather, it seems that it is the localization through the RBM to the condensate that allows KEY1 to significantly phosphorylate EPYC1. Without the RBM, KEY1 is not able to localize to the proximity of the majority of EPYC1, which is in the condensate. This model explains the hypo-phosphorylation of EPYC1 observed in the Δ RBM mutant (Fig. 5e). **In light of these results, we have removed the discussion of the autoinhibition hypothesis from the manuscript (lines 427-434).**

Since KEY1- Δ RBM cannot localize to the condensate, we speculate that the condensates of the KEY1- Δ RBM mutant are similar to those observed in the *key1-1* mutant. We propose that the multiple condensates in both strains do not arise from an active mechanism, rather that the unphosphorylated EPYC1 are very sticky, which inhibits passive ripening (Fig. 6g, Extended Data Fig. 3). **We have updated the text to clearly explain this passive mechanism (lines 555-560).**

3. It would be very interesting to test the mathematical model. This could e.g. be achieved by overexpressing KEY1 to test whether this results indeed in small but uniformly sized and spaced condensates.

We agree that this is a very interesting avenue. **To address this point, we attempted multiple times to overexpress KEY1 but obtained no transformants.** Overexpression may be toxic to cells for reasons that we do not currently understand. Future efforts may attempt to address this potential limitation by using inducible promoters or drug-inhibitable KEY1 variants, but we feel strongly that these are beyond the scope of the present work.

4. The mathematical model nicely explains how the driving force for phase separation varies with KEY1 activity, and the centering effect. It is clear that KEY activity disfavors the formation of extra small condensates. But why does it favor coarsening into a single large condensate?

We thank the Reviewer for asking this question, as it was not clearly explained in the text or modeling results. The effect is indeed rather subtle: the coarsening rate due to Ostwald ripening depends in part on the concentration of molecules in the dilute phase. KEY1 activity increases the number of EPYC1 molecules present in the dilute phase, accelerating coarsening. We

speculate that in the *key1-1* mutant, EPYC1 is so sticky that there is very little of it in the dilute phase, and thus, coarsening is very slow compared to the timescale of the cell cycle.

We model this effect by having KEY1 not only affect the EPYC1 switching rates between sticky and non-sticky states, but also the overall effective sticky self-interaction strength χ . Accordingly, in Fig. 6g we consider two scenarios: (i) absence of KEY1 (*key1-1* mutant), where $k_{s \rightarrow ns}/k_{ns \rightarrow s} = 0$ and EPYC1 is overly sticky with $\chi = 8$; and (ii) low KEY1 activity, where $k_{s \rightarrow ns}/k_{ns \rightarrow s} = 0.001$ and EPYC1 is normally sticky with $\chi = 5.5$. Thus, in our mathematical model, it is the combined effect of a relatively low value of χ and a finite value of $k_{s \rightarrow ns}/k_{ns \rightarrow s}$ that leads to accelerated ripening and dissolution of small condensates.

To clarify the role of KEY1 in promoting the formation of a single condensate in wild-type cells, **we have revised the main text (lines 223-227, lines 555-560) and Methods (lines 2016-2019) in our manuscript. We also added a modeling figure showing the increased ripening in the presence of KEY1 activity (Extended Data Fig. 9).**

Minor comments

Figure 1: It would be helpful to clarify what the purpose of the TP vs. TAP media is for a broader audience.

We have updated the main text to improve the description of the role of TP and TAP media for the spot test growth assay (lines 154-157). TAP media contains acetate, which provides a source of carbon for the heterotrophic growth of the cells. TP media lacks acetate, which forces the cells to grow photoautotrophically.

Under either 3% CO₂ TP with light or 0.04% CO₂ TAP in the dark, a functional pyrenoid is not required for *Chlamydomonas* cell growth (because there is no need to activate the CO₂-concentrating mechanism under 3% CO₂, and cells can grow heterotrophically and cannot photosynthesize on TAP in the dark). These two conditions serve as controls to verify whether the growth phenotypes observed under 0.04% or lower CO₂ on the TP medium are caused by defects in other photosynthetic mechanisms or general metabolic pathways.

Figure 3: Are the phosphorylation sites indicated correctly? 3e indicates multiple sites on the *E. coli*-expressed protein without the addition of the KEY1 Kinase. This seems like an error.

Thank you for the careful reading of the data and for pointing out an opportunity to improve the clarity of the manuscript. These markings are not errors; they represent rare phosphorylation sites we detected in the *E. coli*-expressed protein. To improve the clarity of this part, **we updated Figure 3** by reordering the panels e-g to help the readers compare the EPYC1 phosphorylation sites *in vivo* and *in vitro*, as well as identify the background phosphorylation of EPYC1 expressed in *E. coli* more easily. **We also added a sentence in the figure caption to clarify that untreated EPYC1-GFP from *E. coli* contained several phosphorylation sites (lines 721-722).**

Figure 5d is a non-standardized way of displaying SPR data. Please provide the standard SPR traces.

We thank the reviewer for this request. **We now provide the SPR sensorgrams (Extended Data Fig. 7). Additionally, we have updated the Methods (lines 1903-1921) to clearly explain the analysis done to obtain both Extended Data Fig. 7 and Fig. 5d.**

Reviewer #2 (Remarks to the Author):

Final review

Note: I collaborated on this review with an exceptional predoctoral colleague, hence the use of “we” throughout the enclosed comments.

Summary of insights

In this paper, He, Lemma, and colleagues investigate how the size and number of a biological condensate organelle called pyrenoid is regulated for function in *Chlamy*. First, they identify a kinase KEY1 as a phenotypic (size, number) and functional (reduced growth under low CO₂ condition) regulator of the pyrenoid through a genetic mutant allele *key1-1*. Through in vitro experiments, they show that KEY1 phosphorylates EPYC1, which directly disrupts EPYC1-Rubisco phase separation. In cells, they show that *key1-1* mutant cells lack number/size regulation and that KEY1 expression (or activity) and downstream EPYC1 phosphorylation are both correlated across the cell cycle. They also perform multiple orthogonal experiments to postulate a mechanism by which KEY1 is recruited into the pyrenoid, binds to Rubisco to “activate” its kinase (through auto-inhibition), and in turn, phosphorylate EPYC1 to evacuate it from the pyrenoid. Finally, the authors develop a simple physics-based model to synthesize these distinct observations, which in turn, recapitulates experimentally observed size regulation as well as predicting other observed features, including centering and inter-pyrenoid spacing. Overall, the use of many distinct techniques, experimental and modeling, provides a solid foundation for their proposed model.

We are grateful to the Reviewer for their close reading of the manuscript and their positive comments.

Below are a list of concerns (further stratified into major/minor) and confusion we had while reviewing the paper - addressing which we believe would improve the manuscript and strengthen claims made in the text.

Major concerns

1. Clarification of Key1 Mutant Phenotype and Behavior (mostly Fig 1, S1-2, and text)

The authors rely on a mutant cell type of *key1* to compare to wild-type cells for if and how KEY1 regulated pyrenoid number and size.

A. However, the authors do not directly describe what the mutation was to *key1* and its resulting phenotype in KEY1 behavior. Can the authors describe more clearly?

We thank the Reviewer for this request. The molecular characterizations of the two *key1* mutant alleles (*key1-1* and *key1-2*) are shown in the Extended Data Fig.1 and its figure legend, Supplementary Table 2, and the Methods section. The mutants were both generated by random

insertion of a DNA cassette into introns of the *KEY1* gene. To improve the clarity of the description, **we updated the main text by adding the Chlamydomonas Resource Center IDs of the two *key1* mutant alleles and their accession numbers (lines 127-131), and directly referring to the Methods for further information. We also updated the figure legend of the Extended Data Fig.1 to improve its accuracy and clarity (lines 833-836).**

B. Is *KEY1* expressed at all in the *key1-1* line? If the mutant has no expression, then that should be clearly stated up front to contrast. As such, this was hard for the reviewers to understand how *key1-1* was different from the wild-type.

The Reviewer raises an important point. In an effort to measure *KEY1* expression, we obtained several antibodies, but none could specifically detect *KEY1*, possibly in part due to its low expression levels. Therefore, **we have addressed this request through RT-qPCR on the mutants, wild-type, and rescued strains.** Both *key1-1* and *key1-2* mutants have insertions in introns, which we think get spliced out at low frequency to yield decreased abundance of normal transcript. Our RT-qPCR results indicate that the *key1-1* mutant has ~3% of the *KEY1* transcript abundance compared to the wild-type strain (now shown in Extended Data Figure 1i). It is also worth noting that *key1-1* exhibited no detectable phosphorylation of EPYC1 on a Phos-tag gel-based western blot (Fig. 3a) and mass spectrometry analysis (not shown), indicating that there is not sufficient *KEY1* expressed to phosphorylate EPYC1. We detected higher levels of *KEY1* transcript in the *key1-2* line (extended Data Figure 1i), consistent with what appears to be a low level of phosphorylation of EPYC1 in this line (Fig. 3a). **We have added text to indicate the expression of the *KEY1* transcript in the *key1-1* and *key1-2* mutants (lines 131-133) and provide the RT-qPCR data in Extended Data (Extended Data Fig. 1i).**

C. If no *key1* is expressed - is the rate of coarsening just vastly slower not leading to 1 single pyrenoid or do the authors think there are other mechanisms? This was hard to understand from the paper or model presented.

Yes, precisely as the Reviewer describes, we speculate that the rate of coarsening is vastly slower in the *key1-1* mutant compared to wild type. Our data (Extended Data Fig. 3) and model (Fig. 6 and Extended Data Fig. 9) support this idea. In the revised manuscript, **we have added a plot of the number of condensates over time at the start of the G1 phase of the cell cycle when cells are growing (Extended Data Fig. 3g) to show the nucleation and persistence of condensates in the *key1-1* mutant compared to the wild type. We have also added Extended Data Fig. 9, which shows the rate of coarsening for the simulations in Fig. 6g. Finally, we have updated the text to more clearly explain this passive mechanism (lines 555-560).**

By contrast, if it is expressed at all in the mutant lines, then the following points are unclear:

D. Does it still localize to the pyrenoid, albeit at lower levels?

E. Is there a similar spike in mRNA levels of *KEY1* in the mutant lines? What are the mechanisms that prevent the multiple condensates from dissolving but can still cause them to be smaller?

F. As a corollary, if the *key1* mutant has aberrant but lower activity throughout, shouldn't there be only 1 large condensate? Are there compensatory mechanisms that are present?

Since RT-qPCR detected a very low abundance of *KEY1* mRNA in the *key1-1* mutant, we do not think that *KEY1* is meaningfully expressed in the *key1-1* mutant. Thus, we did not address these questions.

2. Quantification of Condensed Fraction and Protein Behavior (mostly Figs 2-3, S2-3)

A. Is condensed fraction reported per cell post division or total volume?

Thank you for pointing out the ambiguity. The condensed fraction is reported for the total volume of the chloroplasts from a single parent cell, as determined by the chlorophyll channel in imaging. **We have updated the Figure caption (Fig. 2c, line 690; Extended Data Fig. 2d, lines 886) and Methods (lines 1595-1597) to clarify the analysis.**

B. More generally, are there any statements made about intensities or protein concentrations and how they change? What is the implication about effective concentrations per cell?

Reported proteomics and mRNA data across the cell cycle suggest that Rubisco and EPYC1 concentrations are relatively constant across cell division³. Our diurnal western blot on EPYC1 in wild-type cells, in which equal weight of cell lysate were loaded, also shows relatively constant protein levels across cell division (Fig. 3k).

In our confocal timelapse movies during cell division (Fig. 2; Extended Data Fig. 2; Supplementary Videos 2,3), we observed some photobleaching of the Venus fluorescence, especially at the beginning of the acquisition:

Total intensity (AU) versus frame number for all the cells shown in Fig. 2 and Extended Data Fig. 2.

Given these data, in our analysis, we renormalize the intensities such that total fluorescence intensity is constant over time so that the segmentation of the condensates is consistent across the acquisition. This is described in the Methods. **We have updated the figure captions to explicitly state that the protein concentration is assumed to be constant throughout the divisions (Fig. 2c, lines 693-702; Extended Data Fig. 2d, lines 888-911)**

C. In SI, the Venus-tag-Rubisco shows much lower cellular condensed fractions at WT condition as opposed to mutants but not for Fig 2 which measures EPYC1 instead? Is this a statement of these two proteins behaving differently? Can they be treated interchangeably as specified?

We attributed the lower condensed volume fraction of RBCS1-Venus in wild type cells to the timing of that specific experiment. For this experiment, it seems that the pyrenoid of the wild-type cells was already dissolving at -80 min, which can be seen by the recovery to above the initial volume fraction at the end of the recondensation (Extended Data Fig. 2d). While it still does not recover to the full *key1-1*;RBCS1-Venus volume fraction, it is within the range of variability observed in our experiments, and the remaining differences could be due to differences in expression level between the strains or imaging depth for the two samples.

To definitively test whether EPYC1 and RBCS1 can be treated interchangeably, **we performed an experiment in a dual-tagged strain of EPYC1-Venus and RBCS1-mCherry in a wild-type background.** This experiment showed that during cell division, the localizations of the two proteins were highly correlated. Thus, we believe that the proteins can be treated interchangeably. **We have now added a supplementary movie showing cell division in the dual-tagged strain**

and a line in the manuscript describing these additional results (Supplementary Video 4, lines 183-185).

D. An “ectopic” condensate is shown in Fig S3C for a single representative cell but it would be important to verify this is a statistically reproducible behavior - perhaps by reporting on the number or nucleation rate of condensates in more images?

We thank the Reviewer for this request. **We now provide a field-of-view of *key1-1*;EPYC1-Venus cells nucleating new condensates (new Extended Data Fig. 3b-e). We quantified this result as the number of condensates over each hour for *key1-1*;EPYC1-Venus and WT;EPYC1-Venus (new Extended Data Fig. 3g).**

3. Simulation Assumptions and Kinase Activity (Fig 6)

A. Is total EPYC1 concentration held constant across simulation?

Yes, the total concentration of EPYC1, $C_{EPYC1} = C_{max}(\varphi_s + \varphi_{ns})$, is held constant throughout the entire simulation. This can be inferred from Eq. (2) in the Methods section, which includes only switching rates between the two states. **To clarify this point, we have updated both the main text (lines 470-471) and the Methods section (lines 1949-1950).**

B. Are the authors making any statements about total rates of coarsening when kinases are present versus not present? (Fig 6g) if so, perhaps, this is worth reporting additionally as a net change in rate of dissolution besides particular snapshots.

We thank the Reviewer for raising this point. Following the reviewers' suggestion, **we have computed the characteristic cluster size as a function of time with different levels of kinase activity** (i.e., switching rates ratios) to compare the rates of coarsening. In our minimal model, the expected effect of increased ripening rate with changing kinase activity alone is quite small within ranges of $\tilde{k} = k_{s \rightarrow ns} / k_{ns \rightarrow s}$ where coarsening is expected rather than condensate size control, as illustrated in the plot below:

Coarsening behavior in simulations with varying kinase activity ($\chi = 7, \bar{D} = D / (k_{s \rightarrow ns} k_{ns \rightarrow s}) = 10$)

Because of this measurement, we do not make statements about the total rates of coarsening when changing the kinase activity alone.

Instead, to better understand and recapitulate our experimental observations of ectopic condensates in the *key1-1* mutant strain—where the coarsening rate dramatically increased—we hypothesized that KEY1 not only affects the EPYC1 switching rates between sticky and non-sticky states, but also the overall sticky self-interaction strength χ . Accordingly, in Fig. 6g we considered two scenarios: (i) absence of KEY1 (*key1-1* mutant), where $k_{s \rightarrow ns} / k_{ns \rightarrow s} = 0$ and EPYC1 is overly sticky with $\chi = 8$; and (ii) low KEY1 activity, where $k_{s \rightarrow ns} / k_{ns \rightarrow s} = 0.001$ and EPYC1 is normally sticky with $\chi = 5.5$. Thus, in our mathematical model, it is the combined effect of a relatively low value of χ and a finite value of $k_{s \rightarrow ns} / k_{ns \rightarrow s}$ that leads to accelerated ripening and dissolution of small condensates. To clarify this point, **we have revised the main text (lines 547-548, 555-560) and methods section (lines 2016-2019) in our manuscript and added a figure showing the dissolution of the ectopic condensate in simulations of wild-type and *key1* mutant cells (Extended Data Fig. 9).**

C. In this model, there is no nucleation (unless I am missing a description of thermal fluctuations in this continuum model) so it is worth explicitly stating that in the main text and briefly explaining how the “condensates” are seeded under the various conditions and how they reform (presumably through spontaneous phase separation?)?

The Reviewer is correct—there is no nucleation in this continuum model, which is entirely deterministic. Condensates emerge spontaneously via spinodal decomposition; that is, the

system is unstable to small spatial fluctuations in EPYC1 concentration. To clarify this point, **we have updated the main text (lines 510-514) of the manuscript.**

D. We also felt that the figure presentation could have been improved as Fig 6. Almost read like a panel of schematics instead of data. It would help to know that these were simulated conditions (for e.g. by adding colorbars, cleanly labeling plots). We say this only to suggest that it took a while for us to dig into the figure and really understand it.

We thank the reviewer for these suggestions. To better distinguish data from schematics (e.g., in panel h), **we have replaced the simulated data panel h with cartoons. We have also increased the size of the colorbar in panel c, which is the color scale for all the data in the figure, and updated the figure caption to clarify that the scale applies to all the data (line 802).**

E. The model simplifies activity into 1 effective parameter that is not spatially varying and reports broad correlation in size/number with data. Do the other predictions made (repulsion, centering) also hold true if experiment-motivated connections to models of KEY1 - i.e. explicitly modeling KEY1 species concentrations, effective recruitment to pyrenoid, and pyrenoid-dependent activity? We ask this because experiments suggest that loss of recruitment of KEY1 to pyrenoid does show a functional phenotype in experiments but perhaps in the model this would not reflect directly (or would it only be lumped once again into an effective activity?)

We thank the Reviewer for this thoughtful comment. While our model simplifies activity into a single effective parameter, it nonetheless captures a wide range of the rich dynamics observed experimentally. Allowing KEY1 to vary in space would in principle produce qualitatively similar results (see, e.g., our recent work: Martinez-Calvo et al., PRX Life, 2025⁴), reinforcing that the phenomena we report are not an artifact of our minimal formulation. Explicitly modeling KEY1 concentrations, recruitment to the pyrenoid, and pyrenoid-dependent activity is indeed a very interesting avenue for future work, which could help link our framework more directly to the underlying molecular biology. At the same time, such an extension comes with challenges: the biological processes are more complex than our current minimal model can capture, and additional assumptions would be required to describe recruitment dynamics and activity regulation in a realistic way. For this reason, in the present work, we have chosen to keep our minimal model, which already recapitulates the key experimental observations.

4. Conceptual questions with the overall proposed logic of pyrenoid regulation

A. The authors demonstrate that mutations to key1 will disrupt the regulation of pyrenoid size by showing smaller and more condensates compared to the WT. Additionally, the authors experiment state that higher phosphorylation of EPYC1 dissolves the pyrenoid.

A. If KEY1 activity is disrupted, can the authors explain or clarify why the pyrenoid would not stay as one condensate since this decreases the ability for KEY1 to phosphorylate EPYC1?

We thank the Reviewer for pointing out that we did not clearly explain our proposed model for why the multiple condensates do not ripen in the *key1-1* mutant strain. Our data indicate that the multiple condensates arise during pyrenoid growth at the start of the day (Extended Data Fig. 3). In the mutant, these newly emerged ectopic condensates do not ripen into a single condensate. We speculate that the unphosphorylated EPYC1 in the *key1-1* mutant is overly sticky, which inhibits ripening - and we support this idea with modeling in Fig. 6g and Extended Data Fig. 9b. **We have clarified this point as described in the reply to Reviewer 2's comment 1C above.**

B. Do the authors believe there are other constitutive kinases of EPYC1?

Our data suggest that essentially all phosphorylation of EPYC1 is dependent on the KEY1 kinase. From the Phos-tag gel-based western blots, EPYC1 is unphosphorylated in the *key1-1* and *key1-2* mutants, running at the same level as lysate treated with a generic phosphatase (Fig. 3b). While there are minor differences in the EPYC1 phosphorylation patterns observed on EPYC1 *in vivo* vs. on KEY1-treated EPYC1 *in vitro*, it is possible that these differences are due to an impact of Rubisco and/or the EPYC1-Rubisco condensate on the specificity of KEY1 (Rubisco is absent in the *in vitro* assays). Alternatively, it is also possible that *in vivo*, a second kinase acts downstream of KEY1 to phosphorylate additional sites on EPYC1 only after KEY1 initiates phosphorylation. **After revisiting the text, we believe that the current description of the phos-tag gel-based western blot data communicates this interpretation: “Together, these observations indicate that KEY1 is necessary for EPYC1 phosphorylation *in vivo*.” (lines 234-246).**

5. Text clarity

By and large, the text is well written. The intro specifies “active size regulation is largely unexplored” - this is a subjective statement, obviously. Here we point to the authors for missing citations to functional mechanisms of size control - both passive and active (We are confident we are missing many more) to:

A. <https://doi.org/10.1016/j.devcel.2020.06.003>

B. <https://www.nature.com/articles/s41556-022-00882-3>

C. <https://pubmed.ncbi.nlm.nih.gov/29973724/>

I would recommend adding these and suitably revising text if needed!

We thank the Reviewer for the positive comment and for pointing out the ambiguous language and missing references. **We have cited the three papers suggested by the Reviewer (reference numbers A-36, B-35, C-39) and added several others (reference numbers 33-34, 37,38,40, 42-50). Correspondingly, we have revised the text to better capture what is known about size regulation mechanisms within cells and where our work adds to this literature (lines 60-81).**

Minor concerns

1. Figure 5: Can you report on a partition ratio from SNAP dye for better characterization?

We had previously reported “Enrichment” in the Extended Data Fig. 6g as the total intensity inside the condensate divided by the total intensity outside the condensate. As suggested by the reviewer, **we have replaced this metric with the more standard “Partition Ratio”** calculated by the mean intensity inside the condensate divided by the mean intensity outside the condensate. **We have modified the plot (now Extended Data Fig. 8e), modified the figure caption (lines 1008-1013), and the main text (line 411).**

2. The authors reference key1-1 mutants in the main text, but Figure 1 shows data for a key1-2 mutant. Please clarify the difference between key1-1 and key1-2 mutants and why mutants result in similar phenotypes in figure 1k.

We have characterized two *key1* mutant alleles, *key1-1* and *key1-2*, obtained from the CLIP mutant library. The characterizations of these two mutant alleles were mainly described in the Extended Data Fig. 1 and its figure legend, Supplementary Table 2, and the Methods section. To improve the clarity of the description, **we updated the main text by adding the naming of the two *key1* mutant alleles and their accession numbers and cited the Methods section (lines 127-131). We also updated the figure legend of the Extended Data Fig.1 to improve its accuracy and clarity.**

In short, *key1-1* and *key1-2* are two different mutant alleles of the *KEY1* gene, each carrying a random insertion of a DNA cassette into different introns, both obtained from the same *Chlamydomonas* mutant library. As they have mutations in the same gene (*KEY1*), they result in similar phenotypes shown in Fig. 1k and Fig. 3a. The similarity of their phenotypes provides additional support to our conclusion that the phenotypes are due to disruption of the *KEY1* gene rather than due to second-site mutations. (The primary support for the genotype-phenotype connection is provided by genetic rescue of the *key1-1* mutant with a tagged wild-type copy of the gene in the *key1-1;KEY1-SNAP* and *key1-1;KEY1-Venus* lines.)

3. Under ‘Phosphorylation of EPYC1 by KEY1 disrupts EPYC1 binding to Rubisco’, the conclusion is that EPYC1 phosphorylated disrupts binding to RUBISCO. The topic sentence hypothesizes this happens by EPYC1 dissociating or making smaller clusters. Please clarify how readers can distinguish between the two based on the data in figure 4 and what was tested to define a difference.

This was a common comment among the Reviewers. **We have rewritten the section (lines 324-370) and created a new figure panel (Fig. 4e)** to convey the two hypotheses for how KEY1 phosphorylation disrupts phase separation and to explain how the experiment distinguishes the hypotheses. Briefly, we used Fluorescence Correlation Spectroscopy to probe the binding between individual EPYC1 and Rubisco molecules by measuring the diffusion coefficient of EPYC1-GFP: when EPYC1-GFP (60 kDa) binds to Rubisco (550 kDa), the diffusion coefficient of EPYC1-GFP will decrease. If the phosphorylated version of EPYC1-GFP were measured to bind to Rubisco, this would support the small-complex hypothesis (previous Hypothesis 2, now Hypothesis 1 in the revised text). If the phosphorylated version of EPYC1-GFP were found not to bind to Rubisco, this would support the dissociation hypothesis (now Hypothesis 2 in the revised

text). We found that phosphorylated EPYC1-GFP did not bind Rubisco, in support of the dissociation hypothesis.

4. Strongly encourage the authors to deposit their simulation code publicly on the github.

We have uploaded the simulation code to Github (<https://github.com/amcalv/2025-Simulations-Kinase-KEY1-paper>) and cited it in the text (line 2077-2079).

Reviewer #3 (Remarks to the Author):

The pyrenoid is a liquid-liquid phase separated organelle present in many microalgae. It is part of the algal CO₂ concentrating mechanism (CCM) which is a key player for global primary productivity. Here, He and coworkers report a new kinase (KEY1) which is critical to control the size of the pyrenoid, they further describe the mechanism involved in how KEY1 regulates the pyrenoid size and develop a simple mathematical model that recapitulates what is shown in vivo. The new kinase reported is the first mechanism reported in *Chlamydomonas* to directly control the pyrenoid size, it will be of interest to a broad audience. The work is done carefully and very well described. The manuscript is easy to read and provides controls and multiple independent ways to characterize KEY1's function in vivo and in vitro, which makes the work very solid. Overall this manuscript is excellent and will have a big impact on the fields of microalgae physiology, CCM, liquid-liquid phase separation and beyond.

We are grateful for the positive comments of the Reviewer.

I just have one main comment which should be clarified: While the authors claim that KEY1 activity is important for its function, this does not seem to be the case for the pyrenoid's main function of being involved in the CCM. Although the authors do not mention this point directly, they imply it very strongly lines 119-124. However, while the *key1* mutant has an impaired growth under low CO₂ and very low CO₂ (Fig. 1F), this growth defect is not specific to CO₂ limiting condition and is also seen at 3% CO₂ or with TAP. Additionally, while the complementing lines show a fully rescued pyrenoid sizes, they do not really show a complemented growth phenotype, suggesting that any growth phenotype is not directly linked to the pyrenoid size control. The authors should clarify what is the "pyrenoid function" they find associated with KEY1 activity and if the CCM is one of them, they should measure it more directly, for example by measuring the cell's affinity for CO₂. As a side note, the spot test 3% CO₂-TP is overgrown and cannot be used to assess a growth difference.

We thank the Reviewer for pointing out the low quality of the previous spot test. In the previous spot test, it appears that slightly more cells of the wild type were spotted than of the other strains, leading to darker spots of WT than other strains under all conditions, making comparisons challenging. To address this issue, **we repeated this experiment with better normalized cell concentrations (revised Fig. 1k)**, as can be seen by similar growth of all strains under the control conditions – TP-high CO₂ and TAP-dark-air. The results show defective growth of the independent mutants *key1-1* and *key1-2* under low CO₂ and very low CO₂. We see full recovery of growth of the genetically rescued mutant lines *key1-1*;KEY1-SNAP and *key1-1*;KEY1-Venus under low CO₂, and enhanced growth of these lines relative to the parental mutant line *key1-1* and the independent mutant line *key1-2* under very low CO₂. These results establish that KEY1 contributes to fitness under low CO₂ and very low CO₂.

As observed by the Reviewer, the genetically rescued lines do not fully match the growth of the wild type under very low CO₂. This suggests that the rescue construct is not fully functional, potentially due to the addition of the C-terminal tag or the insertion of these genetic rescue

constructs at random sites in the genome, which may lack some native, longer-range genomic regulation.

We agree that it is interesting that the tagged lines rescue a single pyrenoid at low CO₂ but do not fully rescue growth under very low CO₂. It appears that the *key1-1;KEY1-SNAP;RBCS-Venus* line has a slight defect in the timing and extent of its dissolution and recondensation, as can be seen in Extended Data Fig. 2h-k. We speculate that this defect may reflect abnormalities in pyrenoid regulation in the rescued strain, which could contribute to a growth defect in the rescued strains at very low CO₂. **We have added a comment in the main text to highlight this possibility (lines 204-205).**

Minor comments:

- I could not find the manuscript on BioRxiv or a similar public preprint repository. If not already out there, please upload it to favor open access of data as early as possible.

We have now uploaded the manuscript on BioRxiv. Please see:

<https://www.biorxiv.org/content/10.1101/2025.10.09.681382v1> .

- Line 58-59 “most eukaryotic algae”, please change “most” for “many” as none of the cited articles have done a thorough analysis of all eukaryotic microalgae

We thank the Reviewer for pointing out the issue with this statement and references. **To address this issue, we have removed "most" (line 82).** The sentence now reads "The pyrenoid is a large, singular biomolecular condensate⁵² found in the chloroplast of eukaryotic algae^{53,54}, where it mediates approximately one-third of global CO₂ assimilation⁵⁵."

-Line 93 “Kinase of EPYC 1 (KEY1)”, very cool naming, good job!

Thank you!

- Line 240-241, although KEY1 is able to phosphorylate EPYC1 at the same residues observed *in vivo*, I don't see what would make it being the “primary kinase” involved as some EPYC 1 residues seem to not be phosphorylated direct by KEY1 *in vitro*, suggesting the existence of other kinases and secondary effects in the key1 mutant.

We believe it is appropriate to call KEY1 the primary kinase of EPYC1 considering the observations that KEY1 is necessary for EPYC1 phosphorylation *in vivo* (Fig. 3a, b) and that *in vitro* treatment of EPYC1 with KEY1 can reproduce most of the observed EPYC1 phosphorylation sites (Fig 3d, e).

Regarding the presence of additional phosphorylation sites detected *in vivo* but not *in vitro*, we note that these modifications were detected at low frequency in our data, which may indicate that those sites are phosphorylated at low frequency—it is difficult to be confident in this because our mass spectrometry data are not quantitative. Since our *in vitro* phosphorylation experiments were

done in the absence of Rubisco, it is possible that KEY1's specificity expands to include the additional sites when it is in the context of an EPYC1-Rubisco condensate. Alternatively, it is possible that one or more other kinases act to add the additional phosphatases in a manner dependent on KEY1's activity: their activity could be dependent on KEY1-mediated EPYC1 phosphorylation sites, or KEY1 could activate the other kinase(s), e.g., by phosphorylating them. Given the above considerations, we leave the characterization of these additional phosphorylation sites to future studies.

- Lines 338-345 It seems that the KEY1-deltaRBM is excluded from the pyrenoid. Is that the case? Also, is there some indication that KEY1-deltaRBM can be active?

Our data do not show an exclusion from the pyrenoid(s), rather KEY1- Δ RBM-SNAP showed uniform signal throughout the chloroplast, including within the pyrenoids. This is consistent with previous findings which show that deleting the RBM on a pyrenoid-localized protein disrupted its enrichment into the pyrenoid but did not exclude it from the pyrenoid condensate (Fig. 2D of Meyer et al, Science Advances, 2024)⁵. **We have clarified this point by plotting the Partition Ratio for the SNAP fluorescent signal which is tightly distributed around Partition Ratio = 1 in the Δ RBM strain, indicating that there is no enrichment or depletion of SNAP signal from the condensates (Extended Data Fig. 8e).**

To determine whether KEY1- Δ RBM can be active, **we performed an *in vitro* assay with *E. coli*-purified KEY1-WT and KEY1- Δ RBM.** We found that KEY1- Δ RBM was able to phosphorylate EPYC1 at similar levels as KEY1-WT (**Extended Data Fig. 8g**).

- Line 359: why do you propose that there would be only one unidentified phosphatase and not multiple ones?

We thank the Reviewer for pointing out that there could be multiple unidentified phosphatases. **We have changed the text here from “an unidentified phosphatase” to “at least one unidentified phosphatase” (line 440).**

References

1. He, G. et al. Phase-separating pyrenoid proteins form complexes in the dilute phase. *Commun. Biol.* 6, 19 (2023).
2. Wunder, T., Cheng, S. L. H., Lai, S.-K., Li, H.-Y. & Mueller-Cajar, O. The phase separation underlying the pyrenoid-based microalgal Rubisco supercharger. *Nat Commun* 9, 5076 (2018).
3. Strenkert, D. et al. Multiomics resolution of molecular events during a day in the life of *Chlamydomonas*. *Proc. Natl. Acad. Sci.* 116, 2374–2383 (2019).
4. Martínez-Calvo, A., Zhou, J., Zhang, Y. & Wingreen, N. S. Sticky Enzymes: Increased Metabolic Efficiency via Substrate-Dependent Enzyme Clustering. *PRX Life* 3, 033011 (2025).

5. Meyer, M. T. et al. Assembly of the algal CO₂-fixing organelle, the pyrenoid, is guided by a Rubisco-binding motif. *Sci Adv* 6, eabd2408 (2020).

Reviewer #1 (Remarks to the Author):

The manuscript “Kinase KEY1 controls pyrenoid condensate size throughout the cell cycle by disrupting phase separation interactions” by He, Lemma, Martinez-Calvo, et al. examines the condensation and dissolution dynamics of the pyrenoid membraneless organelle of the alga *Chlamydomonas reinhardtii*. They find a previously un-annotated kinase “KEY1” that acts on the Rubisco linker protein EPYC1 regulating its interactions with Rubisco and disrupting their co-phase separation. They demonstrate that the upregulation of KEY1’s activity is important to dissolve the pyrenoid at key steps in the cell cycle. They develop a minimal continuum mathematical model that demonstrates that the relative activity of a kinase like KEY1, and a counteracting phosphatase can control condensate dissolution, reformation, number, size, and even affect the position of the condensate. In the case of the pyrenoid, high KEY1 activity results in dissolution of the condensate, while low activity results in a single large condensate that would center itself within its available volume, much like the pyrenoid does.

This is a well-written manuscript with well-designed experiments. The manuscript advances the understanding of the regulation of condensates that are necessary to ensure proper cell cycle dynamics and equal inheritance into daughter cells. The modeling helps generalize the findings and advance the field’s understanding of how condensate size and behavior may be controlled within cells. The authors have satisfactorily address our comments, and we recommend the manuscript for publication.

We thank the reviewer for their close reading and positive comments.

Reviewer #2 (Remarks to the Author):

In this revised manuscript, the authors present a combination of new experiments/data, simulations, and textual edits. We believe they have done a good job of successfully addressing all of our concerns raised during the original submission.

We thank the reviewer for their close reading and positive comments.

Minor suggestion:

We saw that the github link exists but the repository itself only comprised 2 files, a README and a code file that we couldn't even open or get to run (2 of us, both computational scientists, atleast gave it a fair shot each). While we don't think you need to make production quality software, the following are fairly typical features of academic code papers:

- README that clearly documents the steps or provides dependencies for installing the code and associate libraries
- Instructions to run the code to generate outcomes
- Code that has basic documentation - docstrings at the least - for the functions/classes/modules involved
- Typically an example script to generate data - of analysis in the paper or a representative simulation.

Additionally, if the authors feel so inclined - while this is not necessary, they could add:

- adding simulation data used in figures (csv, npy or other files)
- adding clean scripts to simulate model in the parameter/mesh regime studied in the paper

All these are suggestions in the service of helping the authors work become more reproducible and easier to access for scientists interested to understand, extend, or test your model.

We thank the reviewer for the suggestion and interest in the underlying code. We have updated the GitHub to include a complete README and basic documentation as described.

Reviewer #2 (Remarks on code availability):

Please see the review response.

Reviewer #3 (Remarks to the Author):

The authors have adequately answered my comments, except I still disagree with their interpretation that KEY1 is required for the "function" of the pyrenoid.

The new data, now shown by the authors in Fig. 1k, rather suggest that KEY1 is required for the pyrenoid's role in the CCM. It would be great if the authors acknowledged this nuance (which I am sure they are aware of).

This is an important nuance as the "function" of the pyrenoid is likely more diverse than just the CCM: the pyrenoid is present in high CO₂ conditions when the CCM is not active (<https://doi.org/10.1073/pnas.1522866113>), and it is also highly increased in size in response to O₂ and H₂O₂ (<https://doi.org/10.7554/eLife.67565>), which led to the suggestion that the pyrenoid might be a micro-oxic niche for O₂-sensitive reactions (<https://www.biorxiv.org/content/10.1101/2023.12.21.572672v1.full>).

This nuance is even more critical, given that the field has drawn significant attention to the role of the pyrenoid in the CCM, risking overlooking other (potentially important) roles it might play. These other roles might actually be quite important for the fundamental understanding of this compartment, as well as what it might mean to engineer it into a different system.

We thank the reviewer for pointing out the important nuance of pyrenoid function. We have adjusted the text to highlight that we assess only the pyrenoid's function in the CCM (lines 142-143).